



# Linking sea ice deformation to ice thickness redistribution using high-resolution satellite and airborne observations

Luisa von Albedyll[1], Christian Haas[1,2], and Wolfgang Dierking[1,3]

[1]Alfred Wegener Institute, Helmholtz Centre for Polar and Marine Research, 27570 Bremerhaven, Germany
[2]University of Bremen, 28359 Bremen, Germany
[3]Arctic University of Norway, 9019 Tromsø, Norway

**Correspondence:** Luisa von Albedyll (luisa.von.albedyll@awi.de)

**Abstract.**

An unusual, large polynya opened and then closed by freezing and convergence north of the coast of Greenland in late winter 2018. The closing corresponded to a natural, but well-constrained, full-scale ice deformation experiment. We have observed the closing of and deformation within the polynya with satellite synthetic-aperture radar (SAR) imagery, and measured the accumulated effects of dynamic and thermodynamic ice growth with an airborne electromagnetic (AEM) ice thickness survey one month after the closing began. During that time strong ice convergence decreased the area of the former polynya by a factor of 2.5. The AEM survey showed mean and modal thicknesses of the one-month old ice of 1.96 ±1.5 m and 0.95 m, respectively. We show that this is in close agreement with the modeled thermodynamic growth and with the dynamic thickening expected from the polynya area decrease during that time. In addition, we found characteristic differences in the shapes of ice thickness distributions in different regions of the closing polynya. These closely corresponded to different deformation histories of the surveyed ice that were derived from the high-resolution SAR imagery by drift tracking along Lagrangian backward trajectories. Results show a linear proportionality between convergence and thickness change that agrees well with ice thickness redistribution theory. In addition, the *e*-folding of the tails of the different ice thickness distributions is proportional to the magnitude of the total deformation experienced by the ice. Lastly, we developed a simple volume-conserving model to derive dynamic ice thickness change from high-resolution SAR deformation tracking. The model has a spatial resolution of 1.4 km and reconstructs thickness profiles in reasonable agreement with the AEM observations. The computed ice thickness distribution resembles main characteristics like mode, *e*-folding, and width of the observed distribution. This demonstrates that high-resolution SAR deformation observations are capable of producing realistic ice thickness distributions. The MYI surrounding the polynya had a mean and modal total thickness (snow + ice) of 2.1 ±1.4 m and 2.0 m, respectively. The similar first- and multi-year ice mean thicknesses elude to the large amount of deformation experienced by the closing polynya.

## 1 Introduction

Sea ice thickness is a key climate variable because it governs the mass, heat and momentum exchange between the ocean and the atmosphere (e.g. Maykut, 1986; Vihma, 2014). Sea ice thickness is controlled by a superposition of thermodynamic processes,





i.e. growth or melt, and ice dynamics, i.e. advection and deformation of ice. Both, thermodynamics and mechanics alter, but
also depend on ice thickness.

Thermodynamic processes modify ice thickness slowly depending on the surface energy balance (Maykut, 1986). In contrast,
deformation caused by differential motion of the ice leads to abrupt changes in ice thickness. Driven by winds, ocean currents
and tides, and constrained by coasts and the internal stress of the ice pack, divergent motion creates areas of open water (e.g.
leads) and reduces thickness to zero. Convergent motion results in closing of leads and then rafting and ridging of young and old
ice whereby the latter forms pressure ridges that are many times thicker than the initial thickness. For example, from a survey
of over 300 first-year ice (FYI) ridges, Strub-Klein and Sudom (2012) reported sail heights up to 7.8 m with a mean peak height
of 2.1 m and keel depths up to 26.8 m with a mean peak depth of 8.2 m. The interplay of dynamics and thermodynamics results
in the presence of very variable thicknesses, that can be well described by an ice thickness distribution (ITD). The ITD is a
key parameter in parameterizations of many climate and weather relevant processes, e.g. the heat transfer between ocean and
atmosphere that takes place only in very thin ice. Hence, knowledge of the ITD is crucial for realistic short- and long-term model
predictions of the sea ice thickness and volume (Kwok and Cunningham, 2016; Lipscomb et al., 2007). Since thermodynamic
growth is slow and limited by the equilibrium thickness, deformation dominates in shaping the ITD by re-distributing thin ice
to thicker ice during ridging events or by increasing the open water fraction during divergent motion (e.g. Wadhams, 1994;
Rabenstein et al., 2010).

Submarine and satellite-based observations show a substantial decline of sea ice thickness in the Arctic Ocean within the last
six decades (Lindsay and Schweiger, 2015; Kwok, 2018). At the same time, sea ice drift speed increased significantly, indicating
enhanced ice deformation (Spreen et al., 2011; Rampal et al., 2009). In the context of those changing Arctic conditions, in which
net thermodynamic growth and ice thickness are reduced, the contribution of dynamic processes to sea ice thickness gains more
importance (Itkin et al., 2018).

However, the interdependency between sea ice thickness and enhanced sea ice dynamics is not well understood yet. Most
apparently, the reduction in the material strength of the ice associated with its thinning is suspected to allow more deformation.
As a more fractured ice cover is easier to move, this may explain the substantial increase in sea ice drift speed (Rampal et al.,
2009). This effect is positively reinforced in the Transpolar Drift where enhanced drift speed accelerates the loss of thicker,
multi-year ice (MYI) through Fram Strait (Nghiem et al., 2007). On the other hand, the reduced ice strength and higher drift
speed lead to an increase in deformation that are of great importance in producing a thick ice cover through ridging (Itkin et al.,
2018; Kwok, 2015; Rampal et al., 2009).

So far, it remains challenging to quantify the net-effects of changed sea ice dynamics on sea ice thickness and volume change
because the existing redistribution theory that links deformation and thickness change is not yet well constrained by observations
(Lipscomb et al., 2007; Thorndike et al., 1975; Hibler, 1979). Two recent studies, a short-term, local-scale study based on
airborne laser scanning (Itkin et al., 2018) and a long-term, basin-wide study based on CryoSat-2 ice thickness retrievals (Kwok
and Cunningham, 2016) provide first observational evidence for a linear proportionality between deformation and dynamic
thickness change.



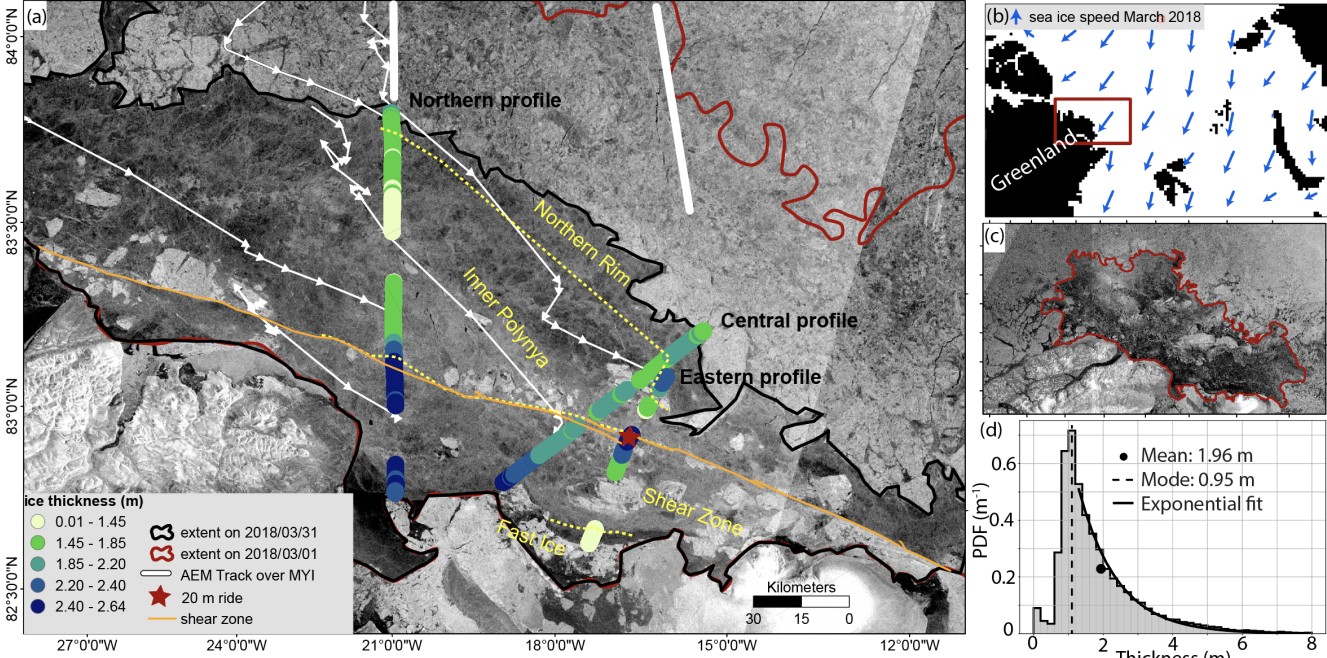

**Figure 1. Ice thickness survey of the FYI in the closing polynya off the coast of North Greenland in March 2018.** a. Sentinel–1 SAR image of the heavily deformed, first year ice (FYI; low backscatter, darker) surrounded by multi-year ice (MYI; higher backscatter) on March 31. The extents of the former polynya on March 1 and 31 are outlined by red and black lines (see also c). Sequence of white arrows illustrates four trajectories representing the typical south-easterly ice movement during the convergent closing of the polynya. Isolated, light gray MYI floes are found in the darker gray FYI. The orange line marks the location of the large-scale shear zone and the red star marks the location of a 20 m thick ridge. Colored circles show 15 km running mean FYI ice thicknesses. Stippled yellow lines show four distinct zones with different mean thickness and deformation history: Thin ice in the weakly deformed Fast Ice; Thick ice in the severely deformed Shear Zone; Moderately deformed, thin ice in the Inner Polynya; and strongly deformed, thick ice in Northern Rim. b. Overview map with monthly averaged, low resolution sea ice drift in March 2018 (www.osi-saf.org, Lavergne et al., 2010). Red box marks region shown in a. c. Sentinel–1 SAR image of the polynya region on March 1, showing initial extent of polynya which was just covered by freshly formed, consolidated ice. d. Combined ice thickness distribution of the FYI shown in a. on March 30 and 31.

Here, we present a regional case study of sea ice deformation and its impacts on dynamic ice thickness change and redistribution, using satellite synthetic-aperture radar (SAR) data and airborne electromagnetic (AEM) ice thickness observations. We

have studied the closing by refreezing and convergence of an unusual polynya that occurred along the coast of North Greenland in late winter of 2018 (Fig. 1). The polynya was located approximately 250 km north-west of the North East Water polynya, a regularly re-occuring event during summer (Schneider and Budéus, 1995). In February 2018, an unusually strong and persistent atmospheric pattern reversed the normally coastward direction of the large-scale ice drift close to Northeast Greenland and thus pushed the common, thick coastal multi-year ice north, to open up a coastal polynya in its place (Moore et al., 2018;

Ludwig et al., 2019). The polynya reached its maximum extent of approximately 65,000 km$^2$ on February 25 (Fig. 1c). The





observed sea ice concentration was unprecedented low for this area and time (Ludwig et al., 2019). While the open water area quickly refroze due to air temperatures well below the freezing point, convergent ice dynamics of the surrounding multi-year ice due to coastward directed winds decreased the area of the refreezing polynya and deformed the newly formed ice heavily, thereby strongly impacting its thickness. One month after the maximum extent of the former polynya, we carried out an AEM

ice thickness survey over the closed polynya that showed mean and modal thicknesses of 1.96 m ± 1.5 m and 0.95 m, respectively, with substantial spatial variability (Fig. 1a, d). Since modal thickness is considered a good first guess for the thickness of the thermodynamically grown, undisturbed ice, the large difference between mode and mean can be attributed to dynamic ice growth by deformation.

Here we present a detailed analysis of deformation derived from SAR imagery and relate it to the resulting ice thickness

distributions obtained from the AEM surveys. We focus on three aspects: First, we relate the large-scale area decrease of the closed polynya to the observed average thickness and show that dynamic processes contributed about 50% of the observed mean thickness. Second, we relate the regional variability of mean thickness and the shape of the ITD to differences in regional deformation observed by SAR ice drift tracking. With this we can establish relationships between key properties of the ITD like mean thickness and *e*-folding and the magnitude of deformation. Third, we demonstrate that high-resolution information

of deformation derived from SAR images can be used to calculate dynamic thickness change with realistic ITDs under some general assumptions summarized in a simple, ice-volume conserving model.

## 2   Data and Methods

Our work is based on AEM ice thickness measurements (Sect. 2.1) and SAR-derived deformation observations and we proceeded as follows:

(1) To quantify the overall, large-scale dynamic contribution to the observed, average ITD, we first estimated the thermodynamic growth of the newly formed FYI (Sect. 2.2) and then considered the simultaneous dynamic thickening of that ice due to the overall area decrease of the closing polynya (Sect. 2.3).

(2) To analyze the small-scale, spatial variability of deformation within the polynya, we derived divergence and shear from SAR-derived sea ice motion fields (Sect. 2.4, 2.5) and reconstructed the individual deformation histories of the surveyed ice

along Lagrangian backward trajectories since its initial formation (Sect. 2.6).

(3) To model the ice thickness along the surveyed ice thickness profiles, we applied a simple, volume-conserving ice thickness model along the backward trajectories, forced by time series of SAR derived, small-scale deformation experienced by the ice along those trajectories (Sect. 2.7). To evaluate the model results, we compared the obtained thicknesses with the observed ones.

### 2.1   Ice thickness measurements

On March 30 and 31, the Alfred Wegener Institute's research aircraft Polar 5 conducted two AEM survey flights over the newly formed ice of the closing Northeast Greenland Polynya and the surrounding MYI. Three profiles, a Northern, Central, and





Eastern with a total length of 230 km were flown over the FYI. Total (snow+ice) thickness was recorded with a point spacing of approximately 6 m (Fig. 1a). In addition, the surrounding MYI between 82.4 °N and 84.5 °N and 14.2 °W and 8.9 °E was
surveyed, with a total profile length of 450 km (thick, white lines in Fig. 1a).

The measurement principle of EM thickness retrievals is based on the strong conductivity difference between sea water and ice that is utilized to determine the vertical distance to the ice-water interface. A laser altimeter provides the distance to the upper snow surface, and subtraction of these two distances gives the combined snow + ice thickness (Haas et al., 2006; Pfaffling et al., 2007; Haas et al., 2009). The footprint of the measurements was approximately 40-50 m and the uncertainty is generally
estimated to be ±0.1 m over level ice (Haas et al., 2009). The footprint smoothing leads to an underestimation of the peak ridge thickness, but because the thickness of the ridge flanks is slightly overestimated, the effects compensate each other for the mean thickness that was found to be in close agreement with drill-hole measurements (Pfaffling et al., 2007; Hendricks, 2009; Haas et al., 1997). Detailed information on the data processing are provided in Haas et al. (2009).

To evaluate the contribution of snow to the observed total thickness, we analyzed snow thickness measurements by NASA's
Operation Ice Bridge surveys over the FYI in the closing polynya on March 22. Their observed modal snow thickness of 4 cm (mean 9 cm) agrees well with the expected accumulation between February and March from the Warren Climatology (Warren et al., 1999). Meteorological observations at Villum Research Station (Station Nord, 81° 36' N, 16° 40' W) indicate no further snow fall event between March 22 and the AEM surveys on March 30 and 31. Since the measurement uncertainty of the EM instrument lies above the estimated snow thickness, we refrain from correcting the total thickness for snow and consider the
thickness measured by the EM instrument as ice thickness. However, we note qualitatively that local snow thickness variability, especially close to ridges, adds an additional uncertainty to the thickness measurements.

Since our study focuses on the evolution of the ice that formed and deformed during the closing of the polynya, we used SAR images to visually identify the northern, outer boundary of the polynya. The boundary is well visible by the strong backscatter contrast between newly formed ice (low backscatter) and MYI (high backscatter, Fig. 1). Several MYI floes were also located
within the polynya. They were excluded from the polynya ice thickness profiles, but were used later for validation of the tracking algorithm. The distinction between MYI and newly formed FYI was based on visual interpretation of the SAR backscatter signature and the thickness observations. After removal of data gaps and MYI ice from the thickness profiles, the total profile length over the FYI was 180 km.

To characterize the ice thickness distributions, we used mean and modal thickness, where the latter was calculated based on
a bin width of 10 cm. For the $e$-folding $\lambda$, we performed an exponential fit of the form $f(h) = a \cdot e^{-\frac{(h-h_{mode})}{\lambda}}$ to ice thicker than the modal thickness $h_{mode}$. The Full Width at Half Maximum (FWHM) characterizes the width of the ITD where it is at 50 % of the maximum. Large $e$-folding and FWHM are taken as indicator of enhanced deformation experienced by the ridged ice.

## 2.2 Thermodynamic ice thickness growth

To separate the dynamic and thermodynamic contributions to the observed and computed ITDs we need a reliable estimate of
the thermodynamic growth. Here we estimated accumulated thermodynamic growth from the observed modal thickness and the thickness of level ice, and the temporal evolution of thermodynamic growth from a thermodynamic model run.





First, ignoring potential early rafting events, we assumed that most level ice that is characterized by extended flat areas of uniform thickness represents undisturbed, thermodynamically grown ice. Hence, we applied a modified version of the level ice filter suggested by Rabenstein et al. (2010) that identifies level ice based on two criteria. First, the vertical thickness gradient along the thickness profile is smaller than 0.006 and second, this condition is met continuously for at least 40 m of profile length, a parameter that was chosen to approximate the footprint of the AEM measurements. The approach is stricter than other identification schemes (e.g., Wadhams and Horne, 1980), but well suited to minimize the amount of deformed ice wrongly passing the filter (Rabenstein et al., 2010).

Second, instead of using a Freezing-Degree-Day model like, e.g., Ludwig et al. (2019), we carried out a dedicated thermodynamic model experiment of the refreezing of the polynya with a regional setup of the coupled ocean and sea-ice configuration of the Massachusetts Institute of Technology general circulation model (MITgcm, e.g., Losch et al., 2010). The model domain comprised the polynya region and surrounding MYI. The model was run with two-category, zero-layer thermodynamics (Menemenlis et al., 2005; Semtner, 1976) and forced with hourly re-analysis data (ERA–5) with a spatial resolution of 31 km. We started the model with an initial ice thickness of 0 m in the polynya on February 25, when the polynya had reached its maximum extent, and ran it until March 31. The model was run without sea ice dynamics in the polynya region and hence enabled us to reconstruct the time series of pure thermodynamic growth $h_{\mathrm{th}}$ without deformation from the daily, spatial means of ice thickness within the polynya.

### 2.3 Overall, large-scale dynamic thickness change due to area decrease of the FYI in the closing polynya

For a first overview of the magnitude of deformation, we quantified the area decrease of the closing polynya by visually analyzing near-daily Sentinel–1 SAR images from February 25 to March 31 (Fig. 1, Fig. 4a). We excluded all MYI floes that were located within the newly formed ice from our area calculations to assure that only the area of newly formed ice was considered.

As a first approximation we assumed ice volume conservation, i.e. that the average dynamic ice thickness increase is proportional to the average area decrease. Hence, we estimated the mean thickness $\overline{h_{t_i}}$ by adding up the thermodynamically grown ice volume up to and including $t_i$ and dividing by the area $A(t_{i+1})$ of the closing polynya on the next time step $t_{i+1}$ to include all dynamic changes up to and including $t_i$. We calculated the thermodynamically grown ice volume by multiplying the thermodynamic growth between $t_i$ and $t_{i+1}$, $\Delta h_{th}(t_i)$, with the area that was available at the beginning of the time step for ice growth $A(t_i)$:

$$\overline{h_{t_i}} = \frac{\sum_{i=1}^{n-1} A(t_i) \cdot \Delta h_{th}(t_i)}{A(t_{i+1})} \tag{1}$$

where n=30 is the the total number of days considered in this study. Note that we assumed that thermodynamic growth continued unaffected by the thickness change induced by deformation. Hence, we did not account for reduced ice growth as the mean thickness increased, but estimated the thermodynamic thickness $h_{th}(t_i)$ from the output of the pure thermodynamic model (Sect. 2.2, Fig. 4).





## 2.4 Sea ice drift from sequential SAR images

To examine spatial differences on scales of a few hundred meters in the deformation of the ice in the closing polynya, we require
165 high-resolution deformation fields. These were computed from ice drift fields retrieved with an ice tracking algorithm introduced
by Thomas et al. (2008, 2011) and modified by Hollands and Dierking (2011). The algorithm matches radar intensity patterns
in sequential SAR images and estimates the spatial displacement of the patterns between the images. The algorithm is based on
multi-scale, multi-resolution image pattern matching offering high robustness at reasonable computational costs (Hollands and
Dierking, 2011). Drift fields were obtained from Sentinel–1, HH polarization SAR images acquired in enhanced wide mode.
170 These had a pixel resolution of 50 m in Polar Stereographic North projection (latitude of true scale: 70 °N, center longitude:
45 °W). Pre-processing was carried out with the ESA SNAP software package and included thermal noise removal using the
noise vectors provided by ESA, image calibration, refined-Lee speckle filtering (7x7 pixels), and a coastline terrain correction
(DEM: GETASSE30 with bilinear interpolation). Based on satellite data coverage, the time steps between two scenes used to
derive ice drift varied between 0.9 and 2 days. The resulting drift data set was defined on a regular grid with a spatial resolution
175 of 700 m. Outliers in the velocity data were reduced by a 3x3 point running median filter covering an area of 2.1x2.1 km.

## 2.5 Sea ice deformation derived from drift fields

Deformation is quantified by strain rates that describe how an object distorts relative to a reference length-scale. For deformations
in which velocities and their gradients are small in comparison to the reference length scale, the strain rates can be linearized
and transformed into two invariants of the 2D strain rate tensor, a normal component comprising tensile and compressive strain
180 and termed divergence rate ($\dot{\epsilon}_{div}$) and a shear rate ($\dot{\epsilon}_{shear}$). Divergence and shear rate can be combined to yield total deformation
rate $|\dot{\epsilon}|$. These parameters are defined by:

$$\dot{\epsilon}_{div} = \frac{\partial u}{\partial x} + \frac{\partial v}{\partial y} \quad \text{(a)} \qquad \dot{\epsilon}_{shear} = \sqrt{\left(\frac{\partial u}{\partial x} - \frac{\partial v}{\partial y}\right)^2 + \left(\frac{\partial u}{\partial y} + \frac{\partial v}{\partial x}\right)^2} \quad \text{(b)} \qquad |\dot{\epsilon}| = \sqrt{\dot{\epsilon}_{div}^2 + \dot{\epsilon}_{shear}^2} \quad \text{(c)} \qquad (2)$$

To calculate spatial derivatives of the ice drift velocity fields, we used a linear approximation based on Green's Theorem that
relates the double integral over a plane to the line integral along a simple curve surrounding the plane. We discretized the curve
185 applying the trapezoid method that linearly interpolates velocity between the vertices ($i$) of the grid cells (e.g. Kwok et al.,
2008; Dierking et al., 2020). The spatial derivatives are thus given by:

$$\frac{\partial u}{\partial x} = \frac{1}{A}\oint_C u\, dy = \frac{1}{A}\sum_{i=1}^{N}\frac{1}{2}(u_{i+1}+u_i)(y_{i+1}-y_i) \qquad \frac{\partial v}{\partial x} = \frac{1}{A}\oint_C v\, dy = \frac{1}{2A}\sum_{i=1}^{N}(v_{i+1}+v_i)(y_{i+1}-y_i)$$

$$\frac{\partial u}{\partial y} = -\frac{1}{A}\oint_C u\, dx = -\frac{1}{2A}\sum_{i=1}^{N}(u_{i+1}+u_i)(x_{i+1}-x_i) \qquad \frac{\partial v}{\partial y} = -\frac{1}{A}\oint_C v\, dx = -\frac{1}{2A}\sum_{i=1}^{N}\frac{1}{2}(v_{i+1}+v_i)(y_{i+1}-y_i)$$

$$(3)$$

We calculated the spatial derivatives from the averaged velocity fields (see Sect. 2.4) considering for every derivative a set
of four grid cells. Hence, we used eight vertices ($N = 8$) to describe the line integral around the plane. We relate the result to





the center of the four grid cells. Moving with a step width of one grid cell, the grid spacing of the deformation fields remained
700 m. Divergence, shear and total deformation were derived using Eq. 2a–c.

## 2.6 Lagrangian analysis of deformation along ice drift trajectories

To attribute differences in the regional thickness variability to differences in the deformation history of the respective ice, we
require detailed information on the origin and drift tracks of the surveyed ice. Therefore, we performed a Lagrangian analysis of
ice deformation along drift trajectories. In order to coincide with the surveyed ice, the drift trajectories were derived backwards
in time, starting at the ice thickness profiles and using the satellite-derived sequential velocity fields described in the previous
section. The Lagrangian deformation analysis was carried out as follows:

(1) As starting positions of the backward tracking we chose 715 points on the thickness profiles surveyed on March 30/31
spaced at intervals between 250 and 350 m (Fig. 1). First we corrected the GPS positions of the AEM measurements by the
drift that took place between the time of the AEM survey and the acquisition time of the satellite images. Backward trajectories
were then initialized at the positions of the surveyed ice on the last satellite image.

(2) For each of the corrected starting positions, we calculated backwards the position of the points at the time step before. To
find the displacement exactly at the points of interest, we interpolated the velocity field and multiplied it with the time difference
between the two SAR scenes. As examples, four trajectories thus derived are displayed as thin white lines in Fig. 1a.

(3) For each time step, we extracted divergence, shear and total deformation from the corresponding deformation fields (Sect.
2.5) at the position of the trajectory at this respective moment in time. Hereby, we selected the deformation from the center
points of the nearest deformation grid cells.

(4) We performed the backward Lagrangian deformation analysis from March 30/31 until March 1. We have chosen March
1 as last day of the backtracking because before this date the new ice in the polynya was not consolidated enough to reveal
recognizable backscatter patterns on two consecutive days. Thus, the tracking algorithm did not produce reliable results before
March 1.

In short, we reconstructed for each of the 715 derived trajectories a time series of the deformation events that the surveyed
patches of ice had experienced during March 2018.

### 2.6.1 Uncertainty of deformation estimates along the ice drift trajectories

Uncertainties of the drift fields arise from a lack of recognizable radar signature variations that may be due to local ice and
weather conditions, sensor parameters, and strong changes of the signatures due to deformation. The error in the initial ice
velocity fields propagates into the deformation estimates along the trajectories in three different ways.

(1) The **tracking error** accounts for the deviation of the reconstructed trajectory from the true one due to erroneous pattern
matching. In our case, in an inhomogeneous velocity field, a deviating trajectory results in the extraction of deformation that was
not experienced by the surveyed patch of ice in reality, but by ice nearby. We assessed the tracking error comparing calculated
trajectories to a manually collected reference dataset of 12 MYI floes that were located in the polynya (see Fig. 1a). In general, the





tracking error was low at the start of the trajectory and increased with every step. After the first time step, the average difference between the reference and calculated trajectories was 51 m, with a maximum of 210 m. On March 1 (end of tracking), the difference between the reference and calculated trajectories had increased to on average 1050 m with a maximum of 2150 m. To account for the fact that the uncertainty of the tracking error sums up over the time steps, we increased the accumulated uncertainty linearly from 700 m, which corresponds to one deformation grid cell on March 30/31 to 2150 m which corresponds to the maximal observed error, on March 1. We interpret the tracking error as an increasing area (circle) around the trajectory, in which the true path of the ice is located. To account for this spatial uncertainty, we extracted divergence, shear and total deformation from all deformation cells whose center points fell into the uncertainty circle (Fig. 2 c). Those values are averaged and saved.

(2) In addition to the tracking error, random errors of the velocity field may introduce **statistical errors** in the deformation. To reduce them, we applied the concept of backmatching that originally is found in the field of photogrammetry (e.g. Schreer, 2005), and has been applied as reliability measure for sea ice drift before (Hollands et al., 2015). Backmatching implies to calculate the drift field twice, by reversing the order of image 1 and image 2 in between. It makes use of the fact that the pattern-search algorithm is initialized on different grids which is either the grid of image 1 or image 2, that will under realistic conditions result in small differences in velocity induced by slightly different pattern matches along strong velocity gradients and random errors in the correlation. A very similar procedure has been applied to determine the errors that arise from independent tracking efforts in the RGPS data by Lindsay and Stern (2003). Assuming that the statistical errors are uncorrelated on the two different grids, considering both deformation estimates will reduce the statistical and the tracking error. Hence, we calculated for every time step a forward and backward field and extracted deformation from both.

(3) A third source of errors for deformation calculations, the **boundary definition error**, is related to the discretization of an inhomogeneous velocity field into grid cells and is difficult to quantify (Griebel and Dierking, 2018; Lindsay and Stern, 2003). Griebel and Dierking (2018) showed that using a rectangular, regular grid and an eight-point ring integral reduces the boundary definition error by at least a factor of two compared to four-point ring integrals. Thus, we decided to apply both means to decrease this source of uncertainty (see Sect. 2.5).

Combining tracking and statistical errors, we calculate deformation and its uncertainty at each point by extracting all grid cells that are located in a circle defined by the tracking error on the forward and backward deformation field.

## 2.7 Ice thickness change along trajectories from a simple model

Based on the basic principles of thermodynamic and dynamic ice thickness changes described by Thorndike et al. (1975) and Hibler (1979), we apply a simple model to model dynamic thickness change from deformation that is sketched in Fig. 2. The aim of this model is to demonstrate in the most simple framework that deformation derived from high-resolution SAR images is capable of reproducing realistic dynamic ice thickness changes and ice thickness distributions. The simple model is a one-layer, volume-conserving model that models mean thickness of a grid cell due to advection and deformation of ice and thermodynamic growth. Each grid cell is defined by the eight vertices that are used to calculate the deformation value from the velocity field





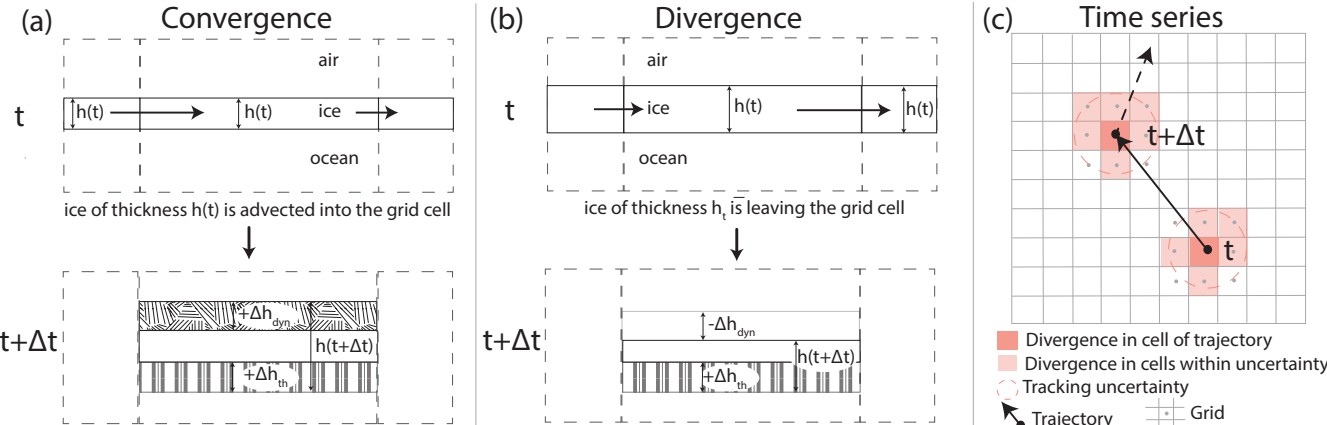

**Figure 2. Sketch of the simple ice thickness model showing vertical cross section of thickness change between time steps $t$ and $t + \Delta t$**
a. In case of convergence, ice with a mean grid cell thickness $\overline{h_t}$ is advected into the grid cell, resulting in a volume-conserving thickness increase $(+\Delta h_{dyn})$. Concurrently, thermodynamic growth $+\Delta h_{th}$ continues unabatedly. b. In case of divergence, ice with a mean thickness $\overline{h_t}$ leaves the grid cell, reducing ice thickness dynamically by $-\Delta h_{dyn}$. Thermodynamic growth $+\Delta h_{th}$ continues unabatedly. c. Trajectories on deformation grid. Grid cells as sketched in a and b are defined by the deformation grid. Divergence and convergence is extracted from the grid cell in which the trajectory is located. For the uncertainty estimate divergence is extracted in all grid cells whose center points (gray dots) are located in the uncertainty range given by the tracking uncertainty (dashed circle).

(see Sect. 2.5) and thus has a side length of 1.4 km. The locations of used grid cells were chosen based on the locations of the ice along the drift trajectories at each respective time step (Fig. 2c). The model does not include any ridging scheme.

We based the general assumptions for the model on the redistribution theory of Thorndike et al. (1975) which states that changes of the ITD $(g)$ in a certain region are a function of (1) thermodynamic growth and melt $(\frac{\partial (fg)}{\partial h})$, (2) advection of ice $(div(\boldsymbol{v}g))$, and (3) redistribution $(\psi)$ within the region:

$$\frac{\partial g}{\partial t} = \frac{\partial (fg)}{\partial h} - div(\boldsymbol{v}g) + \psi \tag{4}$$

Here, every grid cell of our model is considered to be such a region. We simplify Eq. 4 by replacing the ice thickness distribution function $g$ by the mean thickness $\overline{h}$ obtained from integrating and normalizing $g$. This neglects any spatial ice thickness variability in the region, i.e. within the grid cells of our model. Following Hibler (1979), Eq. 4 then simplifies to:

$$\frac{\partial \overline{h}}{\partial t} = \Delta h_{th} - div(\boldsymbol{v} \cdot h) \tag{5}$$

Hence, the mean ice thickness in each grid cell changes by way of thermodynamic growth $(+\Delta h_{th})$ or melt $(-\Delta h_{th})$ and by advection, i.e. convergence or divergence of ice $(-div(\boldsymbol{v} \cdot h))$. This equation eventually represents ice volume conservation and does not take into account ice density changes, for example due to the porosity of pressure ridges (Flato and Hibler, 1995).





Since our deformation analysis is based on a Lagrangian reference frame, we move with the ice and the advection term is reduced to $-div(\boldsymbol{v} \cdot h) = -div(\boldsymbol{v}) \cdot h$ (e.g. Thorndike, 1992). Note that according to Eq. 5 mean thickness change is proportional
to divergence of the velocity field.

Based on this theoretical framework, we made the following assumptions about the properties of the ice: First, we determined the thickness $h$ of the ice that is advected into the grid cell. We assumed that due to the high spatial resolution, neighboring grid cells have a common thermodynamic and dynamic growth history, which is why their mean thicknesses are similar. Hence, we approximate the thickness of the advected ice $h$ by the mean thickness of the grid cell $\bar{h}_t$ at time step $t$ (see Fig. 2). In the
case of convergence ($div(\boldsymbol{v}) < 0$), this results in a dynamic thickness increase of $+\Delta h_{dyn} = -div(\boldsymbol{v}) \cdot \bar{h}_t$ (Fig. 2a). In the case of divergence ($div(\boldsymbol{v}) > 0$) it results in a dynamic thickness decrease of $-\Delta h_{dyn}$ (Fig. 2b).

Second, we approximated the thermodynamic ice growth $\Delta h_{th}$ within the grid cell in Eq. 5 by the growth of the undisturbed, thermodynamically growing ice (see Fig. 2 a,b). We based our assumptions on the observation that deformation changes the thickness only very localized, and hence does only affect a part of the grid cell while thermodynamic growth continues unabat-
edly under the level ice. We are aware that this underestimates (overestimates) ice growth in grid cells that experienced strong divergence (convergence), because divergence results in open water where thermodynamic growth is strongly enhanced, and convergence may create such thick ice that the thermodynamic growth is reduced or even reverted to melt (see discussion Sect. 4.3).

Applying this procedure to every time step of the ice drift trajectories starting at the consolidation of the ice, we thus obtained
a Lagrangian time series of thermodynamic and dynamic thickness change from the deformation grid cells located along each trajectory from March 1 to March 31 by calculating the mean thickness at time step $t + \Delta t$ by:

$$\bar{h}_{t+\Delta t} = \bar{h}_t + \Delta h_{th} - div(\boldsymbol{v} \cdot \bar{h}_t) \tag{6}$$

To account for the tracking uncertainty, we created for each trajectory random combinations of the potentially experienced divergence that were given by the tracking and statistical uncertainty. For each time step, we randomly choose one of the observed
divergence states that were found in the uncertainty circles described in Sect. 2.6.1. We calculated thickness change along each trajectory with 10.000 combinations for the 30 time steps. Mean thickness converged to the first decimal after approximately 1000 iterations.

For each trajectory, we state modal and mean thickness and the corresponding standard deviation from the 10.000 iterations as an estimate of the spread. For a subset of approximately 5 % of the calculations, we observed that the ice thickness in a
grid cell could fall below 0 during the accumulation. To prevent that grid cells that contain only open water (zero thickness) accumulate "negative thickness" when divergence continues, we reset the accumulated thickness to zero in those cases.



## 3 Results

In this section, we first quantify the overall, large-scale dynamic contribution of the decreasing polynya size to the mean thickness in the closing polynya to establish a relationship between large-scale thickness change and deformation (Sect. 3.1). Second, we
describe the regional small-scale spatial thickness variability within the FYI of the closing polynya and demonstrate that it can be attributed to clear regional differences in deformation (Sect. 3.2). Based on those differences, we establish links between the shape of the ITDs and the magnitude of deformation (Sect. 3.2.1). Finally we apply our simple model of dynamic thickness change from deformation and evaluate our results by comparison with the observed thicknesses (Sect. 3.3).

### 3.1 Overall, large-scale dynamic thickness change due to area decrease of the closing polynya

The AEM thickness surveys showed that after only just over one month of ice growth the newly formed FYI had a mean thickness of 1.96 ±1.5 m including an open water fraction of 1.5 %, and a mode of 0.95 m (bin contains all thicknesses between 0.9 and 1.0 m; Fig 1d, 3a). The asymmetric shape of the ITD (Fig. 3a, Table 1), with most of the ice distributed in the thicker part comprising the tail, clearly documents the impact of deformation that has redistributed the thinner ice into thicker ridges. As a result there is a large difference of approximately 1 m between the mean and modal thickness. Since the thermodynamic
growth is expected to be evenly over the polynya region, it leads to rather uniform, level thicknesses of most of the surveyed ice. Therefore we consider the mode to be a good approximation of the thickness of the thermodynamically grown, undisturbed ice. Deformation has led to the presence of a long tail of the distribution up to 20 m thickness, indicating the presence of thick ridges. The tail of the ITD is well approximated by an exponential function with a large $e$-folding of 1.04 (see Table 1). And last, the distribution is rather broad as expressed by the FWHM of 0.8 m. Since the sole interpretation of mean and mode with regard
to dynamic and thermodynamic contributions may miss underlying processes, e.g. the potential contribution of deformation to the observed modal thickness, we will investigate different aspects in the following sections.

The level ice classification (see Sect. 2.2) found only 14 % level ice on the three (Northern, Central, and Eastern) profiles. This is another indication of the large amount of deformed ice in the closing polynya. The ITD of the level ice only (Fig. 3a) is very narrow and almost normally distributed with similar mode and mean of 0.95 m and 1.0 ±0.3 m, respectively. The
modal thickness of the level ice is also identical to the mode of the overall ITD, supporting our assumption that it represents best the thickness of thermodynamically grown ice. The spread around the mean accounts for undeformed ice that started to grow after February 25, potentially some early rafting events, and the spatial variability of the thermodynamic growth due to inhomogeneous snow coverage. We therefore decided to use the mode of the level ice of 0.95 m as the best observational estimate for thermodynamic growth between February 25 and March 30/31.
The thermodynamic model computed an ice thickness of 0.87 ±0.03 m on March 31, in good agreement with the observed modal level ice thickness of 0.95 m (Fig. 1d). The thermodynamic model also allows us to reconstruct the temporal development of ice growth before the observations on March 30/31. This is shown in the time series in Fig. 4a. Results indicated that on March 1 when we started to reconstruct the ice drift trajectories and deformation history, the ice had already grown thermodynamically to a thickness of 0.49 m. Thus, during our study period in March ice grew thermodynamically only by an additional 39 cm.



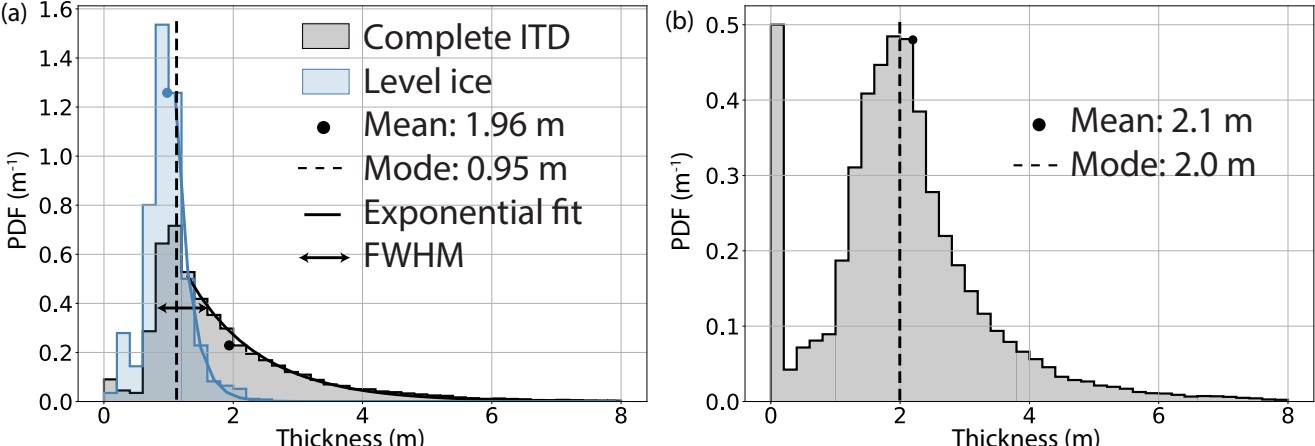

**Figure 3.** Ice Thickness distributions observed by AEM on March 30/31. a. Complete ITD of all FYI in the closing polynya (black) and ITD of the level ice only (blue). For the complete ITD, mean, mode, *e*-folding, and FWHM are indicated. b: ITD of the MYI surrounding the FYI.

For completeness, here we also summarize the thickness observations of the MYI that gave way for and then surrounded the polynya (Fig. 3b; see flight tracks in Fig. 1). The MYI had a total thickness of 2.1 ±1.4 m or 2.3 ±1.3 m, including or excluding open water along the profiles, respectively. The open water fraction was 10% (Fig. 3b) which is quite high, most likely due to divergent drift conditions of the MYI region just before the surveys. Divergence on March 30/31 and the occurrence of open water and very thin ice are visible in the divergence time series in Fig. 4 and in the ITD of the closing polynya (Fig. 1d, 2a),

respectively. Interestingly, the modal thickness of the MYI was 2.0 m, and therefore quite similar to the mean (Fig. 3 b). This is due to the presence of larger fractions of ice thinner than 2 m, but also due to a less pronounced tail of deformed ice. We speculate that this could show that large ridges have been smoothed by the previous summer's melt. We note that the mean thickness of the FYI in the closing polynya is almost as large as the one of the surrounding MYI, but that they differ strongly in their modal thicknesses.

### 3.1.1 Three phases of enhanced area decrease and deformation of the polynya and and their impact on mean ice thickness

As shown above, the general shape of the ice thickness distribution of the closing polynya showed signs of strong deformation since its formation. In the following section, we relate the overall area decrease of the polynya to the observed thickness change.

After the polynya had reached its maximum extent on February 25 (Moore et al., 2018; Ludwig et al., 2019), the usual,

large-scale coastward ice drift had reestablished and persisted through the whole month of March (Fig. 1b). During this time, the area of the closing polynya and the FYI forming in it decreased by 60 % (Fig. 4a). The overall compression of the polynya took place in three major phases, termed early, main and late phase (gray areas in Fig. 4), that were separated by quiet phases with relatively weak deformation. The area decrease and deformation observed within the polynya are closely connected to the

**Figure 4. Dynamic and thermodynamic contributions to mean thickness.** a. Time series of FYI area change (green, right y-axis) derived from satellite images. The three deformation phases (early, main, late) are marked in gray. The thickness derived from the area change is shown in black (left axis). Accumulated mean thermodynamic (red) and dynamic (gray) contribution to the thickness modeled from all trajectories (see b) is displayed in blue. b. Daily contributions from dynamics and thermodynamics to overall thickness. Error bars indicate the standard deviation of the dynamic contribution.

large scale ice drift, and to the magnitude of its coastward component (see insets Fig. 5). Despite the apparent uniformity of the

large scale forcing, deformation within the polynya was regionally variable and distinctly different in certain zones (see Sect. 3.2, Fig. 5).



With the observed time series of polynya area decrease and thermodynamic growth shown above (Fig. 4), we can now compute the resulting ice thickness increase using Eq. 1 (Sect. 2.3). Accordingly, between February 25 and March 31 this simple approach yielded a mean thickness of $\overline{h}$=1.96 m on March 31. This is identical to the observed mean thickness (Fig. 1d,

Table 1). The corresponding time series of mean ice thickness change derived from the area change is also displayed in Fig. 4a. The agreement between theory and observation, which is excellent here, shows that dynamic ice thickness changes within the closing polynya can well be derived from its area decrease. This supports our earlier hypothesis that deformation contributed on average approximately 1 m, i.e. 50 % of the mean thickness, in one month. Further, we note that this good fit is based on only very simple assumptions about thermodynamic and dynamic growth.

## 3.2 Regional differences of ice thickness and deformation within the closing polynya

The previous section was concerned with the average, large-scale dynamic thickness change in the closing polynya. However, we have also observed characteristic regional, small-scale differences of ice thickness and deformation history within the polynya. In the following, we examine potential links between different ice thickness distributions and dominant deformation processes.

Along all three ice thickness profiles (Northern, Central, and Eastern) from the coast across the former polynya we found
common patterns of thickness variability (Fig. 1, Fig. 6). Based on variations of mean ice thickness along the profiles we identified four different banded zones parallel to the coastline: Fast Ice, Shear Zone, Inner Polynya, and Northern Rim. The locations of the four zones are shown in Fig. 1, and Fig. 6 gives an example of the ice thickness in the zones along the Northern profile. The ice within each zone had similar mean thicknesses and similar ITDs. They are shown in Fig. 7. The ITDs of the four zones resemble each other in their modes in the range of 0.85 m to 0.95 m. This is further support of our assumption that
thermodynamic growth was rather uniform across the closing polynya. However, the shapes of the ITDs of the four zones differ strongly in mean thickness, *e*-folding, FWHM, and maximum ice thickness (Table 1, Fig. 7), properties sensitive to dynamic ice redistribution, and indicative of the different deformation histories of the zones. We note that the ITD of the Fast Ice zone shows the weakest signs of ice thickness redistribution, with the smallest mean thickness and highest areal fraction of level ice, while the ice in the neighboring Shear Zone shows the strongest signs of deformation with the largest mean, *e*-folding, and
FWHM (Table 1). In contrast to all other sections, in the Shear Zone there is no clearly defined peak at the thermodynamically grown thickness in agreement with the lowest areal fraction of level ice observed along the profiles. In this zone we observed ridges with a thickness of up to 20 m (Fig. 1). Ice of the Inner Polynya and the Northern Rim had properties between those two extremes, where the ITD of the Inner Polynya indicates less ice redistribution than the one of the Northern Rim. We can obtain more evidence for the inferred differences in the deformation by reconstructing the individual deformation history experienced
by small sections of the ice thickness profiles.

To do so, we derived ice drift trajectories of those 715 sections by means of the SAR imagery (Sect. 2.6). The general direction of ice movement was South-South-East and the total distance traveled by the ice along the trajectories within one month varied strongly between 0.3 and 221 km (mean: 150 km, Fig. 1, Fig. 5a). The drift was unsteady, varying between 0 to 45 km per day and exhibited a high degree of spatial variability and differential motion visible by the deviating course of the trajectories. This
becomes most obvious when comparing trajectories of ice located seaward and coastward of the large shear zone visible on







**Figure 5. Trajectories, drift and deformation during the three main deformation phases** a. Example of trajectories initialized on the Northern profile. The shear zone (see d) splits the trajectories into two groups. Their colors indicate the zone in which they end. b,c,d show snapshots of divergence (red), convergence (blue) and drift (arrows) within the FYI area during the three main deformation phases. Density and length of arrows indicate magnitude of speed. The location of the trajectories at the respective time is marked by colored dots. The insets show the average, large scale drift of a 48h-period covering the indicated time (arrows, provided by www.osi-saf.org) that is linked to the local deformation within the FYI.

March 30/31 (Fig. 5a, d). Combining the course of the trajectories with the deformation fields enables us to locate the surveyed patches of ice of the four zones Fast Ice, Shear Zone, Inner Polynya, and Northern Rim within the deformation fields during the main deformation phases (Fig. 5). This provides valuable information on different deformation histories and origins of the ice, naturally affecting the ice thickness distributions of the four zones.





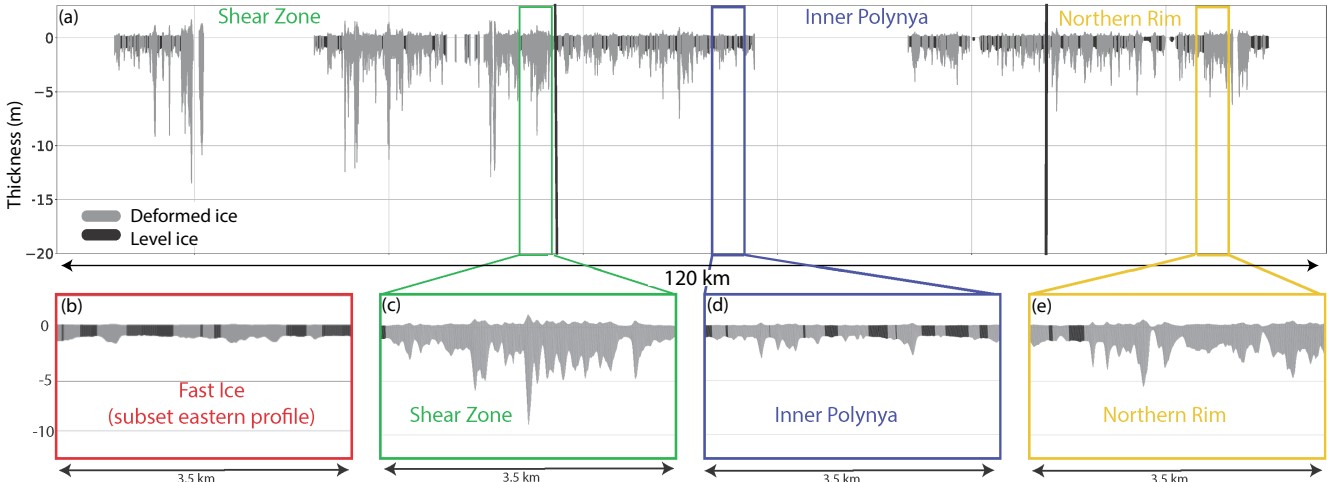

**Figure 6. Ice thickness profile of the Northern profile.** The black and gray colors distinguish between level and deformed ice. Four subsets that are representative for the four thickness zones are displayed in b–e. The locations of the subsets c–e are indicated in a. The Fast Ice zone is present in the first kilometers off the coast of all profiles, but extends only on the Eastern profile for more than 8 km (see Fig. 1). Note the different degrees of deformation in the four zones, depicted by the areal fraction of level ice and the total thickness of the ridges.

The ice that ended in the Inner Polynya and the Northern Rim formed exclusively outside of the polynya region remaining on March 31, i.e. was pushed into the region from farther north previously still covered by the polynya. The trajectories terminating in the Shear Zone and Fast Ice indicate an origin within the FYI region remaining on March 31.

During the early deformation phase, deformation within the polynya appeared locally confined to a network of intersecting deformation zones, often only 2–3 km wide. Ice of the coastward side of the major shear zone (Fast Ice, Shear Zone) was dom-

inated by divergence, while ice in the seaward zones (Inner Polynya, Northern Rim) experienced convergence (Fig. 5b, March 3–6). As deformation increased during the main deformation phase (Fig. 5c, March 16–20), the locally confined deformation features merged into two major deformation zones that formed a band of 20–25 km width along the coast in the South and another band of 25–30 km width close to the MYI edge in the North. Within the major zones, convergence dominated, but was locally accompanied by divergence. The strong compression is represented in the trajectories ending in the Northern Rim that

converged radially (Fig. 5a). In between the Northern Rim and the Shear Zone, the ice terminating in the Inner Polynya traveled with increasing speed southwards provoking divergence in the northern, upstream part of the closing polynya. Apart from this, the ice ending in the Inner Polynya experienced only little deformation (Fig. 5c). During the main deformation phase, the ice of the Shear Zone was located in the coastward, southern deformation band. The strong convergence becomes visible in the deviation of neighboring trajectories and the decreasing travel distance of the patches of ice with increasing distance to the

coast that underlines the strong, northwesterly-orientated gradient in velocity and hence shear that the ice experienced. Ice of the Fast Ice zone became immobile before the main deformation phase and experienced only little deformation.





**Table 1.** Properties of ITDs of different zones of the closing polynya and the result of the thickness model (see Sect. 3.3).

| Zone | Mean (m) | Mode (m) | $e$-folding (m) | FWHM (m) | Level ice fraction (%) | modeled mean (m) |
|---|---|---|---|---|---|---|
| All data | 1.96 ±1.5 | 0.95 | 1.04 | 0.8 | 14 | 1.7 |
| Fast Ice | 1.4 ±0.90 | 0.85 | 0.58 | 0.4 | 27 | 1.1 |
| Shear Zone | 2.4 ±0.85 | 0.90 | 1.49 | 1.5 | 7 | 2.3 |
| Inner Polynya | 1.6 ±0.95 | 0.95 | 0.73 | 0.8 | 15 | 1.0 |
| Northern Rim | 1.8 ±0.95 | 0.85 | 1.05 | 0.7 | 12 | 2.0 |
| modeled ITD | 1.7 ±0.65 | 0.85 | 1.01 | 0.8 | – | – |

Deformation in the late deformation phase (Fig. 5d, March 27–31) was mostly limited to a more than 400 km long, dextral shear zone close to the coast that was identified in the shear (not shown) and divergence fields. The shear was accompanied by convergence dominating March 29 to 30 and divergence dominating March 30 to 31. During the late deformation phase, ice in the Shear Zone became immobile and experienced strong deformation, while the ice seaward of the shear zone (Inner Polynya, Northern Rim) continued to move southwards without significant deformation.

In short, we were able to identify four zones across the closing polynya with differently shaped ITDs and clearly different deformation histories. In contrast, modal thicknesses were similar in all zones and in agreement with the result of a thermo-dynamic model, indicating that thermodynamic ice growth was uniform throughout the polynya. Therefore, we conclude that the observed spatial thickness variability is fully linked to the deformation history of the ice. In the following section we will further explore this link on a more quantitative base.

### 3.2.1 Relationships between magnitude of deformation and the shape of the ITD

In the previous section we qualitatively described the relationship between the spatially varying deformation and ice thickness properties. In this section, we quantify this relationship by linear regression of deformation parameters and ITD properties (Fig. 8). As our deformation information only start on March 1, we subtracted the modeled thermodynamic ice thickness of 0.49 m on March 1 from the mean ice thicknesses of each zone measured on March 30/31 to obtain the thickness change between March 1 and March 30/31. In addition, we averaged the deformation along all trajectories of each zone to obtain the mean deformation experienced by each zone.

Figure 8 shows that increasing convergence (negative divergence) and total deformation are proportional to increasing mean thickness, $e$-folding, and FWHM. Note, that all linear regressions between thickness change and deformation as given in Fig. 8 represent the ice thickness change obtained within 30 days. Like Itkin et al. (2018) and Kwok and Cunningham (2016) we find evidence of a linear relationship between convergence and thickness change (Fig. 8a). Small deviations from this relationship for the Inner Polynya and Fast Ice zones are well within the range of uncertainty indicated by the large standard deviation of the convergence measurements. As the Fast Ice zone is much smaller than the Inner Polynya zone, fewer data points were available





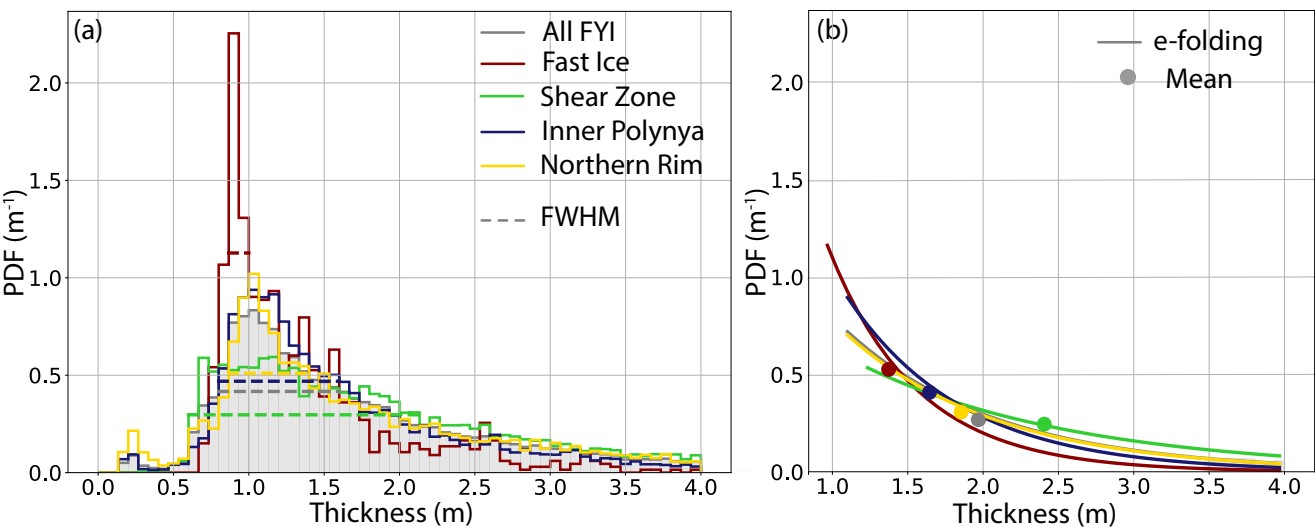

**Figure 7.** ITDs of the four FYI zones on March 30/31. The ITDs differ in a. FWHM that characterizes the dominance of the mode, b. mean and *e*-folding of the exponential tail. The ITD of the complete measurements (all FYI) is displayed in gray.

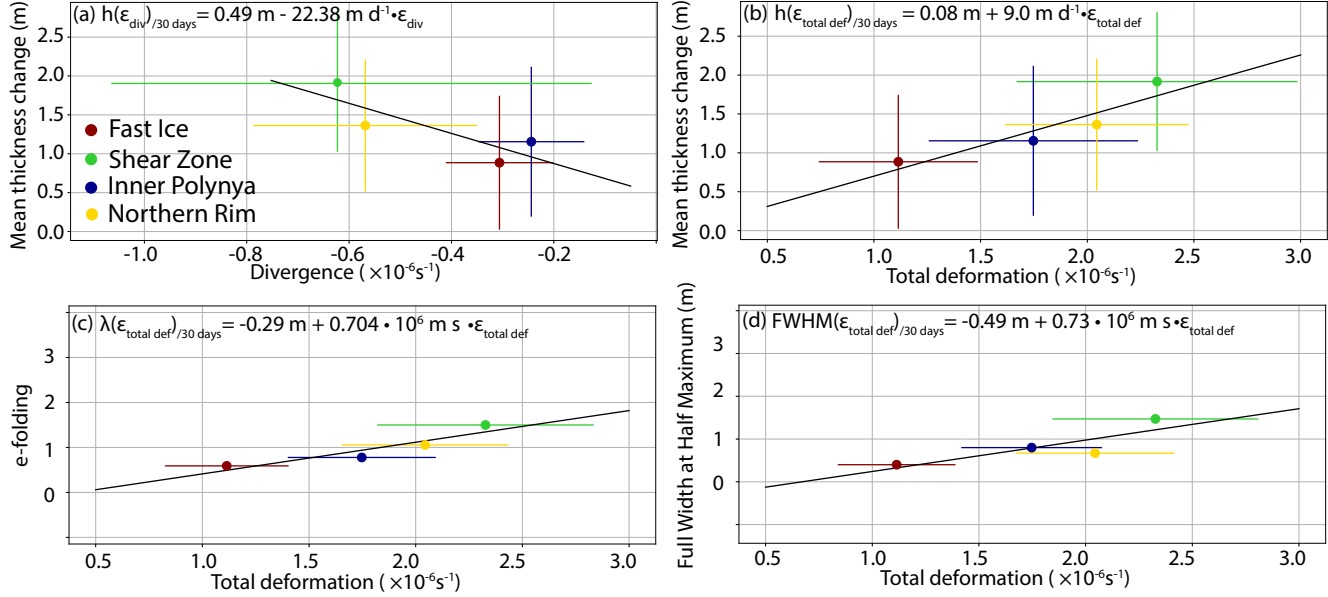

**Figure 8. Relationship between mean deformation and ITD key parameters in the four polynya zones**. The standard deviations of means and deformation are displayed as error bars. Thickness change and mean deformation is given for March 1–31. Note, convergence is negative divergence.





to compute the means and standard deviations. The small number of points do not allow for a more sophisticated fit than the one of a linear regression shown in Fig. 8.

### 3.3 Modeling small-scale thickness variations from high-resolution deformation fields

In the two previous sections, we described the impact of large-scale and regional deformation differences on the thickness distribution. We demonstrated that the amount of area decrease of the closing polynya can directly be used to accurately predict

the corresponding ice thickness increase (Sect. 3.1). In this section, we present the results of the simple volume-conserving model that allows us to compute ice thickness change from high-resolution deformation information. We will evaluate the quality of our thickness model with regard to: (1) how well they reproduce the observed average thickness change, (2) how well they reproduce the observed ITD, and (3) how well they reproduce the observed spatial thickness variability in the four different zones of the closing polynya.

**(1) Average thickness change**

We modeled thickness change along each of the 715 trajectories based on the modeled thermodynamic growth and the observed deformation between March 1 and 31 as described in Sect. 2.7. Figure 4b summarizes the relative contributions of dynamic and thermodynamic growth to the resulting mean thickness. Note that the standard deviations of the dynamic contributions are large, indicating strong spatial variability among the different trajectories. In Fig. 4a we indicate the deformation

phases derived from the polynya area decrease time series (Sect. 3.1.1). During the early deformation phase, the model only computes a weak ice thickness increase because the mean ice thickness was still small and mainly derived from thermodynamic growth. In contrast, the largest dynamic contributions were found during the main deformation phase in the middle of March. The late deformation phase consisted of both convergent and divergent motion in different regions of the trajectories. On average, divergence dominated and mean ice thickness decreased during that phase. Overall, the dynamic and thermodynamic

components resulted in a mean total thickness of 1.7 m, i.e. only 11 % smaller than the observed thickness of 1.96 m. Comparing this thickness derived from high-resolution deformation with the one derived from polynya area decrease (Sect. 3.1) we note that there is good agreement until March 21. Only after that date the area-derived ice thickness increases slightly more rapidly than the deformation-derived ice thickness, resulting in the thickness difference of 0.26 m. However, the ice thickness decrease at the end of the study period between the March 30 to 31 is present in both time series (Fig. 4a).

**(2) Comparison of modeled and observed ITDs**

The mean thicknesses of all 715 trajectories or grid cells, respectively, were combined to compute the ITD of the modeled ice thicknesses. This modeled ITD is compared with the original, observed ITD in Fig. 9. It can be seen that the modeled ITD resembles the shape of the observed ITD well, with regards to a strong, similar mode and a long tail of thick ice that dominates the mean. The modeled ITD possesses similar mean and modal thickness, as well as $e$-folding and FWHM as the observed ITD

(Table 1). However, it lacks the frequent occurrence of ice thicker than 3 m. Additionally, the modeled ITD reveals a secondary mode at 2.2–2.4 m missing in the observations.

**(3) Spatial agreement between modeled and observed thickness profiles**





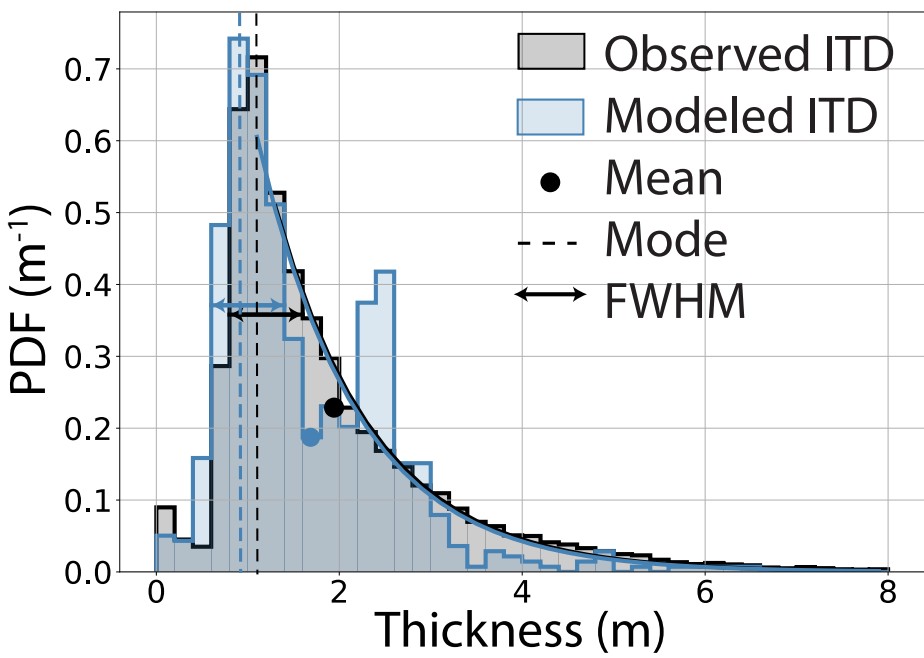

**Figure 9.** Observed and modeled ITD with mean (dot), exponential fit to the tail of the distribution, and FWHM (horizontal bar).

Lastly, we compared the modeled and observed thicknesses along the three AEM profiles (Fig. 10). The modeled thickness profiles resulted from the model grid cells of each trajectory that were all properly placed along the profiles at the end of each
trajectory's drift track, corresponding to their position at the first instance of the backward tracking. For the results shown in Fig. 10 the observed and modeled thicknesses were averaged with a running mean to a common resolution of 2.5 km along the profiles. The figure shows that the modeled thicknesses generally reproduce the characteristic variability of the four zones (Table 1). However, they underestimate the observed thickness at most points of the profiles. The mean modeled ice thickness in the Fast Ice zone is steadily increasing away from the coast as observed. The mean modeled ice thickness in the Shear Zone is
2.3 m, and both modeled and observed ice thicknesses are highly variable and reveal the largest values (Table 1). The model was even able to produce very thick ice representative of ridge zones, but they are not in the same locations as ridge clusters in the observations. In the Inner Polynya zone with a mean modeled thickness of 1.0 m there was less variability in both, modeled and observed thicknesses. The Northern Rim was characterized by thicker ice with a mean thickness of 2.0 m, which was in good agreement with observations on the Northern and Central profiles, but smaller than observed on the Eastern profile. In summary,
our results show that the modeled thicknesses are able to show general differences between the different zones, although details can differ quite much. Underestimation of observed thicknesses is larger in the less deformed Fast Ice and Inner Polynya zones.





**Figure 10.** Modeled and observed thickness profiles across the FYI from south (left) to north (right) of a. the Northern profile, b. the Central profile, c. the Eastern profile. The four zones are marked with colors. Modeled mean thickness are given with the uncertainty derived from the tracking (Sect. 2.6.1). Note different x-axis scales based on different lengths of profiles (see Fig. 1).



## 4 Discussion

### 4.1 Dynamic contribution to mean thickness

One of our key results is that after only one month of ice growth the thickness of the new FYI in the closing polynya was almost
2 m, with sea ice deformation contributing on average 50% and locally up to 90% to the mean ice thickness. This way, within
a month thermodynamics and dynamics restored a first-year ice cover that was almost as thick as the surrounding MYI. These
results provide direct observational support for notions of the importance of sea ice dynamics for predicting the future impact
of a thinner, more dynamic ice cover in the changing Arctic, and if stronger and more frequent deformation can contribute to
compensate for the expected, continuing sea ice losses. Our results obtained on regional scale and over one month bridge the
spatial and temporal gap between two recent, similar studies of ice deformation and thickness change: the short-term, local-
scale study by Itkin et al. (2018) that observed deformation and ridge formation of a single deformation event with airborne
laser scanning data; and the long-term, basin-wide study by Kwok and Cunningham (2016) that used CryoSat-2 ice thickness
retrievals and deformation from low-resolution satellite data. All these studies provided evidence for the large contribution of
deformation to thickness change, and together will contribute to improved representation of sea ice deformation and thickening
in sea ice models.

### 4.2 Magnitude of deformation shapes ITD

Our observations provide insights into two key aspects in modeling sea ice dynamics, namely the mean dynamic thickness
change and the effect of deformation on the shape of the ITD whose accurate representation in models is subject of present
research (e.g. Lipscomb et al., 2007; Ungermann and Losch, 2018).

First, our results have shown that mean dynamic thickness change can be approximated as linear function of convergence. This
is in good agreement with other observational studies (Itkin et al., 2018; Kwok and Cunningham, 2016) and the redistribution
theory (Thorndike, 1992; Hibler, 1979) that forms the basis for sea ice dynamics in most models. For Eq. 5 (Sect. 2.7) we obtain
from the least square fit between convergence and mean thickness (Sect. 3.2.1) the following coefficients:

$$\Delta \overline{h}/_t = \Delta h_{th,t} - h \cdot \dot{\epsilon}_{div} \qquad \text{after Thorndike, 1975, Hibler, 1979} \qquad (7)$$

$$\Delta \overline{h}/_{day} = 0.0163 \, \text{m} \, \text{d}^{-1} - 0.746 \, \text{m} \cdot \dot{\epsilon}_{div} \qquad \text{this study} \qquad (8)$$

The thermodynamic growth term ($\Delta h_{th,t}$) of $0.0163 \, \text{m} \, \text{d}^{-1}$ results in $0.49 \, \text{m}$ of ice growth if integrated over 30 days. This
corresponds reasonably well to the observed thermodynamic contribution of $0.39 \, \text{cm}$ between March 1 and 30/31.

The redistribution theory (Thorndike, 1992; Hibler, 1979) suggests that the slope of the dynamic growth term ($h \cdot \dot{\epsilon}_{div}$) is
given by the thickness of the ice participating in ice compression ($h$). Since we observed the integrated effect of a series of
deformation events during one month, $h$ represents the weighted average of all ice that has participated in ridging during that
time period. Taking advantage of the fact that the strongest deformation event left the largest impact on $h$, we suggest that $h$
is close to the thickness of the ice that participated during the strongest deformation phase on March 16–20. Indeed, the slope





of 0.746 m agrees well with the mean thickness of 0.75 m at the beginning of this event on March 16 (see Fig. 4). Differences between our observations and the coefficients as suggested by the redistribution theory in Eq. 8 are within the uncertainties of

the linear regression.

Second, our results suggest that the *e*-folding of the ITD is proportional to the deformation rate (Fig. 8c). The *e*-folding is defined in the redistribution function ($\psi$ in Eq. 4, Sect. 2.7) that describes how the ice participating in deformation is distributed over the different thickness categories. Previous observational studies have shown that the tail of ITDs derived from ice draft thicker than 5 m is well approximated by an exponential function with a constant, negative exponent between $\lambda=3$ and $\lambda=6$ (e.g.

Vinje et al., 1998; Amundrud et al., 2004). Sea ice models based on Lipscomb et al. (2007) use an exponential ridge redistribution function with a variable *e*-folding that depends on the ice thickness via $\lambda = \mu \cdot \sqrt{h_i}$ where $h_i$ refers to the thickness of the ice that was ridged and $\mu$ is a tunable parameter that can be used to improve the fit between model and observations.

We test whether the here observed range of *e*-foldings between 0.6 m and 1.5 m. can be explained by different ice thickness as suggested by Lipscomb et al. (2007) in contrast to different deformation rates as we found in this study. Following Lipscomb

et al. (2007) we assume that the relationship between *e*-folding and thickness in the ridge redistribution function defined for a single ridging event is passed on during a series of deformation events leading to the final ITD. Granted that the contribution shaping the tail of the ITD comes most from the undeformed, thermodynamically grown ice, i.e. from ice with a thickness between 0.49 and 0.86 m, the *e*-folding could only vary by a factor of $\frac{\sqrt{0.86}}{\sqrt{0.49}} = 1.3$, assuming that the tunable parameter $\mu$ is constant. However, our observed range of *e*-folding values correspond to a factor of 2.5, i.e the relation to thickness alone cannot

explain the variability.

Based on the good linear fit (Fig. 8c), we attribute the large range in the *e*-folding to the magnitude of the deformation rate in agreement with Rabenstein et al. (2010) who related differences in ITDs in the Arctic Trans Polar Drift to varying amount of convergence. Hence, we suggest to chose the parameter $\mu$ as a function of the deformation rate. Since Ungermann and Losch (2018) showed in a sensitivity study with the MITgcm that $\mu$ is an important parameter in shaping the modeled ITD, we expect

this to improve the fit between modeled and observed ITDs.

We identified two processes that change the *e*-folding and potentially link it to the deformation rate.

(1) Ridge formation models from Hopkins (1998) and Hopkins et al. (1991, 1999) showed that ridges first reach a maximum thickness and then continue to grow laterally. This lateral growth widens the ridge and therefore increases the relative occurrence of deformed ice with the maximum thickness, and thereby reduces the *e*-folding. When a ridge begins to form, the balance of

the force needed to push ice farther up or down and the force needed to fracture the ice is decisive for redistributing the ice. In this process, ice thickness and friction play major roles. When the maximum thickness is reached, the ridge grows laterally in proportion to the ongoing deformation. In this stage, larger deformation rates result in wider ridges with the maximum thickness and hence with smaller *e*-folding. Applying the maximum keel draft criterion of Amundrud et al. (2004), we identified several ridges in the measured thickness profiles in the Shear Zone that had reached the maximum ice thickness. However, the

relationship between *e*-folding and deformation rate might only be applicable in regions that experience strong deformation, e.g. coastal regions, because Hopkins (1998) and Amundrud et al. (2004) pointed out that ridges in the central Arctic rarely reach the maximum thickness as the critical stresses do often not last long enough to complete the ridge building process.





(2) Rafting leads to a different *e*-folding than ridging. Riding distributes more ice into a few thicker ice thickness categories, while rafting leads to deformed ice with a rather uniform thickness of only double the original one. If the occurrence of rafting

and ridging depends on the magnitude of deformation, this could establish a link between *e*-folding and deformation rate. Hopkins et al. (1999) identified that the relative likelihood of rafting increases with increasing homogeneity of the ice floes. Hence, regions like the Fast Ice Zone that only experienced little deformation and while the ice was still of relatively uniform thickness might have a higher portion of rafted ice, and thus a different *e*-folding than regions that experienced more ridging. Consequently, the *e*-folding could also depend on the initial composition of thin and thick ice and on the deformation history.

Lastly, we acknowledge other aspects, for example the creation of rubble fields, hammocks, or the ratio of shear and convergence, could influence the *e*-folding. The shear to convergence ratio varied among the four zones in the polynya, but we were not able to draw any conclusion due to too few data points. Since we do not have more frequent thickness observations during the polynya closing period we can only evaluate the impact of deformation integrated over 30 days. Therefore, we also miss information about potentially contrasting effects like, e.g., ridge consolidation and collapse.

### 4.3 Modeled vs observed thickness: Limitations of the model

Based on a simple volume-conserving model, we derived thickness change along ice drift trajectories and calculated ITDs from the final thickness at the end of each trajectory. The modeled ITD resembles the observed one in the typical, skewed shape with a dominant central mode and a long tail of thicker ice (Sect. 3.3, Fig. 8). However, the derived ITDs are composed of mean thicknesses in the 1.4 km, long grid cells of our model, which are too large to resolve individual ridges or ridge clusters. The

modeled ITDs are rather comparable to, for example, ITDs derived from strongly averaged radar altimetry data, e.g. CryoSat–2 ITDs of e.g. Kwok (2015). Those ITDs have been derived from measurements with an altimeter footprint of approximately 0.31 km by 1.67 km in along- and across-track direction, respectively. Our modeled ITD agrees well with the observations in the thinner thickness categories. However, it shows a second mode at 2.2–2.4 m that was not observed, and underestimates the amount of of ice thicker than 3.5 m. Inherently, our model smooths the thickness of ridges over one grid cell, reducing the

occurrence of very thick ice. We attribute those differences to the absence of an explicit ice redistribution scheme. For example, the unrealistic second mode formed during the main deformation phase (March 16–20) when a lot of ice with a thickness 0.75–1.2 m was advected into many grid cells, doubling their thickness to 1.5–2.4 m (Fig. 4). Here, an explicit ridging scheme could have distributed the ice volume more realistically into ridges thicker than 2.2–2.4 m.

Apart from those differences in the shape of the ITD, we have found that modeled mean ice thicknesses were generally smaller

than the observed ones. As the agreement between modeled and observed thermodynamically grown ice was quite good, we attribute the general underestimation of mean thicknesses to insufficient modeling of the dynamic contribution. There are two main shortcomings of the model:

First, our model does not account for the high macro-porosity of unconsolidated FYI ridge keels. Numerous studies have shown that mean ridge porosities amount to 11–22 % (Kharitonov and Borodkin, 2020; Kharitonov, 2019a, b; Strub-Klein and

Sudom, 2012), with the largest range between 11 % and 45 % for old FYI ridges and newly formed FYI ridges, respectively





(Ervik et al., 2018; Høyland, 2007). If we assume that the fraction of 86 % of deformed ice in all observations had a porosity of 11–22 % the mean modeled thickness would increase by 0.1–0.3 m to 1.8–2 m.

Second, in the simple model the thermodynamic growth was modeled based on the growth of an undeformed layer of ice, regardless of the actual mean thickness of each grid cell. Hence, the model overestimates thermodynamic growth in all cells that
experienced strong convergence and were therefore thicker than the thermodynamic thickness. At the same time, our approach underestimates ice growth in all cells that experienced divergence, because thermodynamic growth is stronger in leads than in adjacent consolidated ice. We carried out a sensitivity study to estimate the impact of unaccounted new ice formation in leads. If there was divergence, we replaced the ice leaving the grid cell with new ice of the thickness that could form within one day. Integrated over 30 days and all profiles, this resulted in an additional 0.3 m of ice, i.e. a mean thickness of 2 m. Since
the dominating deformation type in our study was convergence and shear, this effect is less important than it might be in a different deformation regime. For future work, we suggest to couple the SAR deformation retrievals with a fully developed sea ice model that considers those interdependencies. For example, the single-column model ICEPACKincludes full solutions for thermodynamic growth and melting and mechanical redistribution due to ridging (see CICE Consortium Icepack, 2020). SAR derived deformation rates can be used to force the mechanical redistribution of ice in the ICEPACK model.

Both those shortcomings can explain the observed differences in the mean thicknesses. However, there are additional possible reasons for deviations of observed and modeled thickness which we shortly discuss below.

(1) Due to challenging conditions for SAR tracking over very young ice we could only begin the thickness modeling on March 1 and assumed an initial, uniform thickness of 0.49 m corresponding to the thermodynamic ice growth in the first days of the closing polynya. However, early deformation before March 1 might already have created an inhomogeneous ice thickness field.
This concern is mitigated by the fact that we observed almost no polynya area decrease between February 25 to March 1.

(2) We did not consider additional opening and closing of ice due to shear on subgrid scales that can be observed in similar situations (e.g. Stern et al. (1995) and Kwok and Cunningham (2016)). However, the effects of divergence and convergence on mean thickness compensate each other on a subgrid scale in our simple model, apart from the effect of divergence on new ice formation (see above, main sources of uncertainty).

(3) The daily imaging of the polynya by SAR images cannot account for deformation caused by tides. Tides and inertial motion can cause recurrent opening and closing with associated sub-daily new ice formation and subsequent deformation. These processes can contribute 10–20 % of the Arctic wide seasonal ice growth (Kwok et al., 2003; Heil and Hibler, 2002; Hutchings and Hibler, 2008). Due to the polynya's location across the continental slope, tidal currents in this region exceed the ones in the central Arctic that are in the order of 0.5–1 cm s$^{-1}$ (Baumann et al., 2020). In the polynya region over the continental
slope (83.2° N 22.9° W) the Oregan State University tide model (Egbert and Erofeeva, 2002) states tidal currents of up to 5–6 cm s$^{-1}$ and oceanographic measurements under the Fast Ice close to Station Nord indicated semi-diurnal tidal currents in the order of 2 cm s$^{-1}$ (Kirillov et al., 2017). Assuming a contribution of tides to sea ice formation of at least a similar order as in the central Arctic, tides could have contributed in our case an additional new ice growth of 0.14–0.28 m.

(4) Uncertainties in the drift fields and the deformation rates could have introduced additional errors. The spatial uncertainties
of the trajectories shown in Fig. 10 are not able to explain all deviations between modeled and observed thickness. Due to the





highly localized nature of deformation in time and space, the true deformation rates might be larger than the calculated, averaged ones. For example, during the main deformation phase Fig. 4a shows that the area derived thickness (black line) indicates more thickness increase than the deformation derived ice thickness (blue line). A potentially underestimation of the deformation rate during this strongest deformation event could explain the underestimation of the mean thickness.

(5) When comparing the model results to the EM ice thickness measurements, uncertainties of the latter need to be considered as well. While the accuracy of EM measurements is ±0.1 m over level ice, EM measurements typically underestimate the maximum thickness of pressure ridges (Haas et al., 2009). However, despite this shortcoming most EM thicknesses obtained here were still larger than the modeled thicknesses. This provides evidence that mean EM ice thickness estimates over length scales of 1.4 km or so are quite reliable, and that footprint smoothing of ridge thickness profiles compensates the underestimation

of ridge crests by overestimating thickness over the flanks of ridges (Pfaffling et al., 2007; Hendricks, 2009).

## 5   Conclusions

An unusual latent heat polynya with a size of $> 65000\,\mathrm{km^2}$ occurred in late winter 2018 at the coast of North Greenland and provided us with a unique opportunity to observe a natural, but well-constrained, full-scale ice deformation experiment. While the open water refroze quickly due to low air temperatures, convergent ice dynamics deformed the newly formed ice. One

month after the maximum extent of the polynya was observed, the area had halved, naturally accompanied with strong impact on the ice thickness distribution. In this case study, we analyzed airborne measurements of ice thickness and their relationship to deformation obtained from high-resolution synthetic-aperture radar (SAR) satellite images. Our aim was to link the magnitude of deformation to ice thickness redistribution and to show that deformation derived from SAR images can be used to derive dynamic thickness change of the region.

This study provides evidence of the high relevance of deformation dynamics in creating and maintaining a thick ice coverage. Sea ice deformation contributed on average 50% or 1 m and locally up to 90% to the mean thickness. Within one month the dynamic processes re-established an ice coverage with a mean thickness of 1.96 m, almost as thick as the surrounding multi-year ice with a mean thickness of 2.1 m.

In the view of a changing Arctic with increasing fractions of thin ice, increased ice drift speed, and a higher frequency of

deformation events, accurate representation of sea ice deformation in models is crucial for predictions of future sea ice thickness and extent. Our observations reveal new insights into the link between deformation and the redistribution of ice shaping the ice thickness distribution (ITD). We provide quantitative evidence that the *e*-folding of the ITD is a function of the deformation rate. These findings can be used for further improving the representation of ITDs in sea ice models, e.g. by constraining the parametrization of the ridge redistribution function. Further, we found that mean dynamic thickness change is a linear function of

convergence in close agreement with the redistribution theory (Thorndike et al., 1975; Hibler, 1979), and previous observational studies (Itkin et al., 2018; Kwok and Cunningham, 2016).

We developed a simple volume-conserving model to derive dynamic thickness change from high-resolution SAR deformation tracking. Modeled mean thicknesses were smaller than observed ones, but they agree within the limits of the main uncertainties due to ridge porosity and not considered new ice formation in leads formed by divergence.

The model based on input from high-spatial resolution deformation fields of 1.4 km allowed us to reconstruct an ITD that resembled the measured one in the typical, skewed shape with a dominant central mode and a long tail of thicker ice. However, we conclude that without a redistribution scheme the tail of the thickest ice cannot be realistically modeled.

For future work, we suggest to couple the deformation retrievals with a fully developed sea ice model that takes drift and deformation as forcing and calculates dynamics and thermodynamics for several thickness categories, e.g. ICEPACK (CICE

Consortium Icepack, 2020). Considering the good availability of SAR data in the polar regions, this opens up the possibility to derive dynamic thickness change and ITDs for many regions of the Arctic and Antarctic sea ice cover.

*Data availability.* Sentinel-1 scenes are publically available from the Copernicus Open Access Hub (https://scihub.copernicus.eu/dhus/home) and may be processed with the open source software SNAP (https://step.esa.int/main/toolboxes/snap/). AEM thickness data and fully processed deformation fields will be made available at the World Data Center PANGAEA https://www.pangaea.de/

*Video supplement.* The Video supplement shows a time series of divergence and shear in the closing polynya from March 1 to 31. Dots display location of selected trajectories on the respective dates specified in the title. Lines show the traveled distance within the last time step. Arrows indicate sea ice drift. The colors show magnitude of divergence (left) and shear (right). The video supplement is made available via TIB AV-Portal.

*Author contributions.* LvA carried out the analysis, processed the deformation data and wrote the manuscript. All authors contributed to the
discussion and provided input during the concept phase and the writing process.

*Competing interests.* CH is a member of the editorial board of the The Cryosphere. All other authors declare that they have no conflict of interest.

*Acknowledgements.* We thank Martin Losch for fruitful discussions on the content of this paper and for helping setting up the MITgcm model runs. Thomas Hollands supported us with the tracking algorithm. Nils Hutter provided initial oceanographic data sets for the model runs.
Stefan Hendricks and Jan Rohde were involved in the field campaign and processed the AEM data. Thomas Krumpen and Florent Birning provided guidance on the thermodynamic growth in the polynya. This work contains modified Copernicus Sentinel data [2019] and snow thickness data used in this study was acquired by NASA's Operation IceBridge. We acknowledge EUMETSAT Ocean and Sea Ice Satellite



Application Facility (OSI SAF, www.osi-saf.org) for the low resolution sea ice drift products. The Department of Environmental Science, Aarhus University is acknowledged for providing data from the Villum Research Station in North Greenland. This study was supported by the

Institutional Strategy of the University of Bremen, funded by the German Excellence Initiative, and by the Deutsche Forschungsgemeinschaft (DFG) through the International Research Training Group (IRTG) ArcTrain.





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
