# Peer review of "Linking sea ice deformation to ice thickness redistribution using high-resolution satellite and airborne observations"

_The Cryosphere, 2020_

## Referee Comment (RC1) · Anonymous Referee #1 · 21 Nov 2020

General comments:

This manuscript deals with an unusual ice deformation event that occurred off the northeastern coast of Greenland in early 2018. The authors make use of SAR data and ice thickness derived from an airborne survey to understand the ice deformation history within a polynya that opened rapidly and refroze over the course of one month. The authors address the contributions of thermodynamic and dynamic processes to the geometric shape of the ice thickness distribution within the polynya. The data and analysis presented are of interest to the community, and the study may warrant publication after revisions to address the concerns listed below. I found the manuscript text

itself to be extremely long, and shortening some sections could really improve things. Thorough copyediting throughout is also required.

Specific comments:

Polynya:

What type of polynya is this (latent or sensible heat), and how did it form? I think a brief discussion about the geophysical characteristics of this polynya, and specific details of its formation, are warranted in the introduction/background information. On L63 the authors refer to an "unusually strong and persistent atmospheric pattern" that drove formation of the polynya. But I think this statement is too vague to understand the forcing mechanism/s. What type of atmospheric pattern? Was the polynya formation wind driven, temperature driven, or both? They say that the atmospheric pattern reversed the "normally coastward direction of the large-scale ice drift close to northeast Greenland". Is there a reference (publication or data set/analysis) to corroborate this statement about the typical ice drift direction? Is ice drift normally coastal in this area in winter, or is it actually parallel to the coastline? Knowledge of the processes driving daily thickness change is also needed to interpret the data shown in Fig 4. If the polynya opened as a result of a wind-driven event, rather than a local heat event, then the thickness change that occurred at the beginning of the time series was dynamic (i.e. thicker MYI replaced by open water). This point does not appear to be fully captured.

Ice type:

The survey area (shown in Figure 1 and discussed in the text, L98) is described as a region of first year ice (FYI), but I don't think this is accurate. (1) The airborne survey was conducted over both FYI and multi-year ice (MYI) as shown in 1(a) – AEM tracks shown by both coloured dots, and thick white lines, traverse areas of high and low SAR backscatter indicating a mixture of ice types. (2) There seem to be many MYI floes within the polynya itself that were either advected into the polynya (as the authors

assume) or that remained in the polynya after the temporally abrupt opening event. I'm curious to know how the thickness of these thicker MYI floes within the polynya (that were traversed by the airborne survey) were excluded (L120) from the ice thickness distributions shown in 1(d) and 3(a). Since there is MYI within the FYI area (Fig 1a), and new ice forming in leads within the MYI region (Fig 3b, L332) a qualifier regarding ice type should be included throughout. It would be more accurate to describe a region as having a "dominant" ice type rather than only one ice type.

SAR analysis:

The description of how the SAR data are analyzed is vague with no specific details about how ice type was derived, nor how ice parcels were identified for drift tracking, or the derivation of deformation (divergence and shear rates). This means the analysis presented is irreproducible, and does not lend itself to the future work proposed on L650-651. For example, rather than quantitively referring to the SAR data presented in Figure 1 (a) and (c) as "light" and "dark to distinguish areas in the study region I recommend something more quantitative. Consider providing threshold values, or ranges, of backscatter (in dB) that can be used alongside a colour scale with appropriate units, to define ice type. How were the data used to define the "rim" of the polynya? i.e. was a specific back-scatter threshold used here? If so, please include the details of this in the text. Similarly, how was the sheer zone within the polynya identified and defined? How were regions of fast ice defined? (Fig 1a) Was SAR backscatter utilized to define these zones?

Modal ice thickness:

I found that there was an error of 0.05 – 0.1 m throughout the study in the reporting of modal ice thickness. If the bin width of the thickness distributions is 20 cm, as shown in the figures, then the modes reported in the text are not consistent with the data shown in the figures. Isn't the mode of the thickness distribution shown in Fig 1d 1.1 m +/- 0.1 m (not 0.95 m as written on L70, L306-307)? Likewise, the mode of the level ice shown

in Fig 3a is 0.9 m, (not 0.95 m, as stated on L319). The primary mode shown in Fig 3b appears to be 0.1 m +/- 0.1 m, and the secondary mode is 1.9 m +/- 0.1 m (and not 2.0 m, as stated on L335).

Data:

The majority of the data used in this study lack citations, or are not public, meaning the analysis is not reproducible. For example, the citation for ice drift data (a web URL to a generic data page) is not sufficient, and is not indicative of the particular data set used. Similarly for the IceBridge data, no product is referenced. Are the AEM data publicly available?

Language:

Overall, the manuscript needs to be revised to improve English usage and grammar, since there are many functional mistakes in the formulation of key sentences. Substantial copyediting is required to aid understanding. I've suggested some specific edits within the technical list below.

Technical corrections:

L11: Change "by drift tracking along Lagrangian backward trajectories" –> by tracking ice drift along reverse Lagrangian trajectories to go back in time

L16-17: Change "The computed ice thickness distribution resembles main characteristics like mode, e-folding, and width of the observed distribution" –> The computed ice thickness distribution resembles the main characteristics of the observed distribution including mode, e-folding, and width.

L17: clarify what is mean by width, do you mean "full width half maximum"?

L19-20: Change "The similar first- and multi-year ice mean thicknesses elude to the large amount of deformation experienced by the closing polynya" –> the similarity between the mean thickness of first- and multi-year ice suggests the scale of deformation

that occurred within the closing polynya.

L30-32: the use of "mean peak depth" to describe keel depths does not sound right. I suggest rewording this using a term such as modal depth. Also, I appreciate that there's a lot of statistics presented in Strub-Klein and Sudom (2012) and so I would suggest here that you clarify which observational statistics you are quoting. I believe they relate to "maximum" sail heights and keel depths. Here you might also refer to Duncan et al. (2020) which updates and expands upon the Strub-Klein and Sudom (2012) work, and found that the average maximum sail height was 2.01 m for >17,000 ridges formed in FYI.

L45-46: Provide a suitable reference to support this statement.

L52: Remove "so far" since it is not needed in this sentence. Also change "changed" –> changes in

L65: the text refers to the date of the maximum extent of the polynya as February 25th, but the figure (Fig 1c) shows a SAR image from March 1st.

L66: change "unprecedented" –> unusually

L68: revise "coastward directed winds" with a vector direction

L69-73: Is this a statement of work described elsewhere (i.e., previously published) or a summary of the work we are about to read about in this manuscript? If the former, provide a suitable reference; if the latter, move this statement from the introduction to the conclusion section.

L96: for completion I suggest you include the year to any dates provided within the text, in addition to month and day.

L105: by "peak" do you mean modal, or maximum?

L109-111: What Operation IceBridge data product for snow thickness is used here? What is the uncertainty associated with an airborne snow thickness observation of 4

cm?

L109-111: How much does uncertainty in snow thickness contribute to errors in the attribution of thermodynamic processes to the overall ITD?

L113-115: how does this assumption impact the uncertainty associated with the AEM thickness estimates, relative to that stated on L105?

L118-119: unfortunately this is not possible for the reader since there is no colour scale provided with the SAR data shown in Fig 1, nor is it clear what the units are.

L121-122, L149-151: I'm curious to understand what is meant by "visual interpretation of the SAR backscatter signature". Is this done using linear pixel greyness values, and therefore subjective, or by applying thresholds to the SAR data in dB? How are MYI floes defined and excluded?

L125: according to the figures, the bin width is 20 cm.

L166: can you explain what is meant by "radar intensity"?

L127: change "Large e-folding and FWHM" –> to large values of e-folding and FWHM

L142: can you briefly describe what is meant by "two-category, zero-layer thermodynamics"?

L153: change "adding up" –> summing

L153-154: does 't' represent thickness, or time?

L159-160: Snow depth on thin ice has a large control on thermodynamic ice growth. How was thermodynamic growth impacted by snow thickness changes (and/or snow redistribution) over 30 days? Does imprecise knowledge of this impact the conclusions drawn?

L174: Provide an example of the derived ice drift data so that the reader may evaluate the results for reasonableness

L182: what are the variables u and v in equation 2?

L171: Provide an example of the derived ice deformation data so that the reader may evaluate the results for reasonableness

L191: Did the authors compute uncertainty in the derived divergence, shear and deformation fields?

L195: "In order to coincide with the surveyed ice" – this is quite awkward, consider rewording.

L200: what is the delta time between the airborne survey and image acquisition?

L202: Change "at the time step before" –> at the previous time step

L210: how is the reliability of the tracking algorithm quantified?

L212-213: another very awkward sentence that is hard to follow - consider rewording

L220-224: Can you show this assessment?

L225: change "sums up" –> accumulates

L291, L293: is there a reason why the number of combinations and iterations are reported to three significant figures? Are combinations and iterations the same thing?

L306 and L332: how is open water fraction computed?

L307: "with most of the ice" – state the % here

L310" Change "evenly" –> even, or is there a word missing here?

L310: is uniform thermodynamic growth "expected" over such a large area? I think it is assumed (but not necessarily expected) since we do not know about snow distribution.

L312-313: "Deformation has led to the presence of a long tail of the distribution up to 20 m thickness" – But the scale in in Fig 3a only shows data to 8 m. What % of samples in the tail span 8 m to 20 m? Consider adding AEM profile data here to substantiate

this statement (similar to the data shown in Fig 6).

L320: Are the modes identical? Looking at Fig 3a, it appears the mode of the level ice thickness distribution is less than that of the 'complete' ice thickness distribution.

L335: Isn't the modal thickness 1.95 m (Fig 3b)?

L560: Did the authors consider the ice thickness distribution from CryoSat-2 for this region so as to substantiate their statement?

L564: double "of"

L586: change "we suggest to couple the SAR deformation retrievals with..." –> we suggest coupling deformation statistics retrieved from SAR analysis with ....

L630, 632: change "coverage" –> cover

L645-655: this sentence is quite hard to follow - consider rewording

Figure 1:

In 1(a) the drift trajectories (thin white lines with arrows) are not defined in the legend. What is the reasoning behind the uneven increments used in the color scale for ice thickness? Why, for example, is the majority of ice (according to figure 1d) combined, and represented by only one colour increment (light green) while thicker ice is divided into four increments ranging between 0.15 and 0.24 m in thickness? A scale bar for the blue arrows in 1 (b) is required, showing drift magnitude. Change "stippled" to "dashed" in the figure caption. Do the data in 1(a) and (d) show ice thickness (as stated in the figure caption) or ice thickness + snow depth (as stated on L98-99)?

Figure 3:

Indicate in the axis labels for (a) and (b) whether you show ice thickness + snow depth or ice thickness only. From my reading of the text I think (a) is the distribution of ice thickness, but (b) is the distribution of ice + snow thickness. Is the "complete" thickness

distribution shown in Fig 3a repetition of the data shown in Fig 1d? If so, remove one of these duplicate figures.

---

## Author Comment (AC1) · 18 Jan 2021

**Response to Anonymous Referee #1**

Dear Referee #1:

Thank you very much for your time and effort you put into the detailed comments on our manuscript with the title "Linking sea ice deformation to ice thickness redistribution using high-resolution satellite and airborne observations". We believe that your suggestions will help us to improve the readability of our manuscript significantly. Please see below our answers (blue) to your specific and the technical comments (black) that do not address language edits. We will address the remaining technical comments with the throughout copyediting you asked us for in the final revised manuscript.

You asked for shortening the manuscript, but also suggested to add more information on several aspects. To find a balance between those competing demands, we decided to remove information where we think it is not strictly necessary for the main message of our manuscript, as described below. We will remove:
- L. 176 to L. 186: The description of the deformation calculation (as it only repeats existing literature)
  L.300 to L. 339: The description of the multi-year ice (MYI) surrounding the polynya and Figure 3 b

We also add more references in the manuscript to studies that provide the information you were asking for.

**Specific comments:**
  (1) Type of polynya:
- Thanks for pointing out where essential information are still missing. We agree that it is very important to make clear that we studied the closing of a polynya that was created primarily dynamically (Moore et al. 2018). While air temperatures were rising above 0°C, Moore et al. 2018 showed that the polynya was a latent heat polynya, created by the divergent ice motion, and the warmer surface air temperatures contributed only by reducing the sea ice production. Hence, we will add a sentence on the type of the polynya in the introduction (l. 2 "an unusual, large, latent heat polynya", l 60: "of an unusual, latent heat polynya that…")
- However, we are aware that our manuscript is already long, which is why we suggest to add more and clearer reference to the preceding studies (Ludwig et al. 2019, Moore et al. 2018) that dealt with the formation history of the polynya instead of describing it in more detail in our manuscript.
- We replace the description of the most likely origin, a "unusually strong and persistent atmospheric pattern", by its effect which where "unusually strong and persist northward winds over the Greenland Sea" (l. 62)
- The large-scale drift patterns associated with the opening and closing of the polynya are presented in detail by Ludwig et al. 2019, e.g. Figure 9 a, b. Here, the authors compared the unusual drift direction end of February with the long-term mean. We have referenced this publication at the end of the sentence (l. 65).

(2) Ice Type:

Thanks for asking for a clarification on the ice type. (1) We follow your advice and differentiate between the ice surveyed by the campaign that comprised of both, young ice and MYI, and the ice for which we calculated deformation and modeled thickness, that was only young ice. (2) To differentiate between the MYI floes and the young ice, we predominantly based our assumptions on their formation history which we could reconstruct by tracking the ice backwards in time. This way, we could distinguish between ice that had formed beginning of March and MYI that drifted into the open water /was located within the open water before. We combined this information with the thickness profiles and the backscatter of the SAR images on March 31/30.

(3) SAR analysis:

Thanks for your comments that helped us to identify unclear points. We add short statements based on our explanations in the manuscript. Regarding …

(1) start of drift tracking: For the start point of the tracking, we down-sampled the GPS coordinates of the airborne flight campaigns to 250 m. Gaps in the thickness observations made it necessary to increase the distance between the starting points which lead occasionally to distances of 350 m. No additional selection process based on ice type or similarly was done here. The tracking started at the down-sampled GPS coordinates.

(2) Derivation of deformation: To calculate deformation from drift, we followed the approach widely used in literature, described in details by e.g. Kwok et al. 2003, Kwok et al. 2008, for a review: Dierking et al. 2020. As you pointed out, the manuscript is long which is why we tried to remain as concise and short as possible. More information can be found in the references provided. We add in l. 184-190 that the reader can find more detailed information in the cited literature. Indeed, for the sake of keeping the manuscript as short as possible, we are considering to move the complete description of the deformation derivation into a supplement.

(3) SAR backscatter values and the classification of the ice type:

The radar backscattering coefficients depend on frequency, polarization, incidence angle, and season (freezing, melting, and effects of melt-freeze cycles), hence also the thresholds between ice types vary. Also, the influence speckle and instrument at low backscattering levels noise has to be considered. In recent work, automated sea ice segmentation and classification is therefore carried out e.g. using statistical methods such as maximum likelihood decisions, or machine-learning methods such as neural networks. This is far beyond the scope of our study here. Grey tone variations are good proxies for separating various ice classes visually (a practice common also in operational ice charting), in particular if complementary information is available, as in our case thickness properties and deformation history as described in lines 365-366. In this context, the qualitative description of "light" MYI and "dark" young ice in the caption of Fig. 1 was only used deliberatively to give the reader a quick guide for where to look for.

In respect to the naming convention of the zones, the names (Fast Ice, Shear Zone, Inner Polynya, Northern Rim) were chosen to distinguish between the four

zones. They only reflect one aspect of the deformation history. For example, ice in the Fast Ice zone became quickly immobile (see L. 405-406, red trajectories in Fig. 5a). The ice in the Shear Zone experienced strong shear during March 29-31 (L. 407-410). For display of the shear fields, please see the video supplement (http://doi.org/10.5446/49540).

(4) Modal thickness:
Thanks for pointing this inconsistency out to us. We make sure that figures and text agree upon this point in the revised manuscript.

(5) Data:
We have submitted the airborne electromagnetic (AEM) ice thickness, high resolution drift and deformation data to the data repository Pangaea and add the reference as soon as we receive it. We also add details on the specific products we used for the large-scale drift and operation ice bridge data in the revised manuscript.

**Technical corrections:**
We address all of the technical corrections with the revised manuscript. Please see our suggestions and answers to the ones that do not address language edits below.

L109-111: How much does uncertainty in snow thickness contribute to errors in the attribution of thermodynamic processes to the overall ITD?
In this paragraph we only describe the contribution of the snow cover to the observed total thickness, since the laser signal is from the snow surface.
We are aware that snow has a strong effect on the thermodynamic growth of thin ice and have attributed the variability of the level ice thickness partly to this effect (see L. 321-324). For a more detailed answer on how uncertainty in snow redistribution affects our results, please see our answer to your question related to L. 159-160.

L113-115: how does this assumption impact the uncertainty associated with the AEM thickness estimates, relative to that stated on L105?
The overall uncertainty of the airborne electromagnetic (AEM) ice thickness survey increases to the sum of both, i.e. 14 cm. We add a sentence in l. 115: "Hence, the combined uncertainty of the AEM thickness is +- 14 cm. "

L118-119: unfortunately this is not possible for the reader since there is no colour scale provided with the SAR data shown in Fig 1, nor is it clear what the units are.
The boundary of the young ice – MYI is identified visually based on the grey tone contrast. We found that the edge of the polynya, marked by the sharp transition of darker and brighter grey tones, was easy to identify in almost all images. We worked on backscatter data given in dB-scale, where we applied a histogram stretch for an improved visual interpretation. The knowledge of grey scale and related units is not required in this context. We add a half sentence about the (stretched) backscatter values in dB-scale in the caption of Fig. 1. We provide an additional video supplement here (https://doi.org/10.5446/50650) to let the reader evaluate the manually created outlines.

L142: can you briefly describe what is meant by "two-category, zero-layer thermodynamics"?

A two-category, zero-layer thermodynamics refers to a model set-up that simulates only ice thickness and concentration, i.e. its thickness categories only consist of zero thickness (open water, given by the concentration) and mean thickness. Although there are also multi-category thickness distribution sea-ice models, the 2-category model based on Hibler (1980) is still most widely used and has proven to result in realistic simulations. The "zero-layer thermodynamics" refer to the fact that the model does not consider storage of heat in the ice. This two-category, zero-layer thermodynamics model set-up complies with a standard version of the MITgcm. Therefore, we provided several references that describe the thermodynamics of the MITgcm. We believe that adding more details in the text would unnecessarily prolong the manuscript.

L159-160: Snow depth on thin ice has a large control on thermodynamic ice growth. How was thermodynamic growth impacted by snow thickness changes (and/or snow redistribution) over 30 days? Does imprecise knowledge of this impact the conclusions drawn?

The timing of snow fall events was considered in the thermodynamic modelling by forcing with precipitation from the ERA-5 reanalysis data. However, the local snow redistribution due to the wind is dependent on the ice surface topography and cannot be considered explicitly. Hence, individual trajectories (Fig. 10) include an uncertainty in the thermodynamic growth due to unknown snow cover variations, which contributes to the deviations between observed and modelled thickness. However, we based our conclusions on regionally averaged trajectories. On those larger spatial scales, we are confident that our thermodynamic estimates are valid thanks to 1) the agreement of the estimated overall thickness from the area change and the observed thickness (section 3.1) and 2) the agreement between the modal thickness of the ice and the modelled thermodynamic ice thickness in the four subregions (Tab. 1). Thus, we think that the imprecise knowledge of the snow redistribution does not impact our conclusions.

L174: Provide an example of the derived ice drift data so that the reader may evaluate the results for reasonableness

Three examples of ice drift data are displayed in Fig. 5 (arrows). We can now also provide the link to the video supplement (http://doi.org/10.5446/49540) where arrows indicate drift speed and direction. We have submitted all drift + deformation data to the data repository Pangaea where the reader may download and evaluate them as soon as it is published there.

L171: Provide an example of the derived ice deformation data so that the reader may evaluate the results for reasonableness

Three examples of ice deformation data are displayed in Fig. 5 b-d (colours) and in the video supplement (http://doi.org/10.5446/49540).

L191: Did the authors compute uncertainty in the derived divergence, shear and deformation fields?

We are aware of the different sources of uncertainty of deformation parameters, which we describe in section 2.6.1, where we explain how those propagate into our final modelled ice thickness. We did not compute uncertainty of the single deformation estimates since in particular the estimation of the tracking error requires an effort beyond the scope of this study, and directly applicable equations for the boundary definition errors have not been published yet. The uncertainty of the *drift* depends on the local conditions, and is difficult to judge for thinner, easily deformable ice. Therefore, we decided to provide a reference value based on the manual tracking of the MYI floes (described in l. 220). As major point, however, we assume that the uncertainty in thickness changes is more strongly influenced by the position errors of the reconstructed paths of ice drift than by the uncertainty of the deformation parameters.

L210: how is the reliability of the tracking algorithm quantified?

We based our decision regarding the use or rejection of results on the criteria described in Hollands et al. 2015, that are the difference in backmatching and the confidence factor (CFA). The CFA consists of several quality criteria in respect to the texture of the SAR image and the correlation itself. For details we refer to their publication.

L220-224: Can you show this assessment?

We have provided an additional figure (Figure RV1) attached to this answer that presents the analysis of the reference tracks. The figure shows the difference between reference track and the calculated trajectory for each time step. Also indicated are the mean of the differences at the first and the last time step. The dashed black line gives the assumed uncertainty for each time step as described in l. 225.

L291, L293: is there a reason why the number of combinations and iterations are reported to three significant figures? Are combinations and iterations the same thing?

Yes, they are the same. We will reword this to make it clear.

L312-313: "Deformation has led to the presence of a long tail of the distribution up to 20 m thickness" – But the scale in in Fig 3a only shows data to 8 m. What % of samples in the tail span 8 m to 20 m? Consider adding AEM profile data here to substantiate this statement (similar to the data shown in Fig 6).

We consider to provide additional figures of the Eastern and Central profile line in the supplements.

L560: Did the authors consider the ice thickness distribution from CryoSat-2 for this region so as to substantiate their statement?

We believe that there was a misunderstanding in how we intended this reference to the CryoSat-2 ice thickness distributions. We did not mean to say that Kwok (2015) analysed CryoSat-2 data from the former polynya. Rather we wanted to express that both in our approach, as well as in Kwok's (2015) radar altimetry, ITDs are compiled from highly averaged data with a comparable averaging length of 300 to 1400 m. We will reformulate this sentence to make this clearer.

Figure 1:

What is the reasoning behind the uneven increments used in the color scale for ice thickness? Why, for example, is the majority of ice (according to figure 1d) combined, and represented by only one colour increment (light green) while thicker ice is divided into four increments ranging between 0.15 and 0.24 m in thickness?
We have chosen the colour scale to stress the differences in the four zones (Fast Ice, Shear Zone, Inner Polynya, Northern Rim). As described in Tab. 1 the mean of the four zones varies between 1.4 and 2.4 m. This is why we have chosen this non-linear colour scale. We can add a half sentence about this to the caption.

Figure 3: Indicate in the axis labels for (a) and (b) whether you show ice thickness + snow depth or ice thickness only. From my reading of the text I think (a) is the distribution of ice thickness, but (b) is the distribution of ice + snow thickness. Is the "complete" thickness distribution shown in Fig 3a repetition of the data shown in Fig 1d? If so, remove one of these duplicate figures.
Thanks for pointing out that this caused confusion. We will mention in the caption that we both times present total thickness, i.e. snow + ice thickness. Fig. 3a contains information from Fig. 1d, but provides additional information on the level ice thickness distribution.

**References used in this reply:**

Dierking, W., Stern, H. L., and Hutchings, J. K.: Estimating statistical errors in retrievals of ice velocity and deformation parameters from satellite images and buoy arrays, The Cryosphere, 14, 2999–3016, https://doi.org/10.5194/tc-14-2999-2020, 2020.

Hollands, T. and Dierking, W.: Performance of a multiscale correlation algorithm for the estimation of sea-ice drift from SAR images: initial700results, Annals of Glaciology, 52, 311–317, https://doi.org/10.3189/172756411795931462, 2011

Hollands, T., Linow, S., and Dierking, W.: Reliability Measures for Sea Ice Motion Retrieval From Synthetic Aperture Radar Images, IEEE Journal of Selected Topics in Applied Earth Observations and Remote Sensing, 8, 67–75, https://doi.org/10.1109/jstars.2014.2340572, 2015.

Kwok, R.: Sea ice convergence along the Arctic coasts of Greenland and the Canadian Arctic Archipelago: Variability and extremes (1992-2014), Geophysical Research Letters, 42, 7598–7605, https://doi.org/10.1002/2015gl065462, 2015.

Kwok, R., Cunningham, G. F., and Hibler, W. D.: Sub-daily sea ice motion and deformation from RADARSAT observations, Geophysical Research Letters, 30, 2218, https://doi.org/10.1029/2003gl018723, 2003.

Kwok, R., Hunke, E. C., Maslowski, W., Menemenlis, D., and Zhang, J.: Variability of sea ice simulations assessed with RGPS kinematics, Journal of Geophysical Research, 113, https://doi.org/10.1029/2008jc004783, 2008.

Ludwig, V., Spreen, G., Haas, C., Istomina, L., Kauker, F., and Murashkin, D.: The 2018 North Greenland polynya observed by a newly introduced merged optical and passive microwave sea-ice concentration dataset, The Cryosphere, 13, 2051–2073, https://doi.org/10.5194/tc-13-2051-2019, 2019.

Moore, G. W. K., Schweiger, A., Zhang, J., and Steele, M.: What Caused the Remarkable February 2018 North Greenland Polynya?, Geophysical Research Letters, 45, 13,342–13,350, https://doi.org/10.1029/2018gl080902, 2018.

---

## Referee Comment (RC2) · Amelie Bouchat (Referee) · 26 Jan 2021

In this paper, the authors present the results of an airborne electromagnetic thickness survey over the anomalous polynya that opened north of Greenland in 2018. They link the observed ice thickness distribution of first-year ice that froze within the polynya after one month of its opening to (a) the area change of the polynya (assuming volume conservation) and (b) the reconstructed small-scale deformation history of the ice along the observed profiles. They show that both methods agree really well with each other, and also agree well with the observed mean thickness and ITD. By inspecting the three AEM thickness profiles and the reconstructed deformation fields from SAR im-

agery, four zones presenting different ITDs and deformation history are identified. The authors show that the main ITD parameters (i.e. e-folding, FWHM, and mean thickness) depend linearly on the deformation magnitude experienced on average within each zone. While it is commonly assumed in the modelling community that the ITD shape should depend on the thickness of the ridging ice, this observational case-study shows that a dependence on the deformation rate should also be introduced. This offers the possibility for further tuning of ITD parameterizations, which is known to affect the ability of sea-ice models to accurately represent the small-scale, sub-grid, dynamics.

The paper is very rich in new data and analysis/conclusions that will be useful to the sea-ice community. However, I find that it needs major editing, restructuring, and perhaps shortening, to improve its readability. I therefore recommend it for publication in The Cryosphere, after the following comments are addressed in the revisions.

Main comments:
————————————
1) Sections 2.4-2.6 should be clarified with the help of additional details about how the drift and deformation fields are obtained. Specifically, I did not understand how the deformations are obtained from the gridded drift data set described in Section 2.4, which resulted in further difficulty understanding the description of the deformation errors in Section 2.6.1. What I understood is that you have a tracking algorithm that you use to obtain ice velocity fields from SAR images, which you gridded on a regular 700m x 700m grid. Then, using this gridded velocity field, you obtain the deformations, but using a Lagrangian formulation (i.e. the contour integral in Eq. 3), which confuses me because I thought you were dealing with a gridded drift data set, not a Lagrangian one. Then it looks like you assume that you have a gridded deformation field (e.g. Fig. 2c), correct? And do you then use a gridded velocity field to integrate Lagrangian

trajectories backwards starting on the AEM profiles?

2) In the results section, the authors sometimes refer to results that were not already presented in statements like "We have seen this..." and then proceed to show what these results are. Whereas it should be reversed: present the evidence/observations first before you can say that we have seen it or what you conclude from it. In the "other comments" below, I have identified some of these places where I find that restructuring how the results are presented would help improve the manuscript.

3) The manuscript would benefit from text/grammar editing to improve its clarity. I have made some "editing suggestions" below to help with this.

Other comments:
—————————————-

p3 L71: "Since modal thickness is considered a good first guess for the thickness of the thermodynamically grown" Reference? If this comes from your results, then mention it.

p3 Figure 1:
a) Mention how the white trajectories were acquired. Insert: "20m ride" –> "20m ridge"
b) The general low-resolution ice drift does not match the trajectories seen in (a). Maybe mention something about this?
c) This panel does not seem useful. Could be removed and keep only the March 1st contour in red in (a)?
d) "Combined ice thickness distribution of the FYI shown"... add along the AEM tracks?

p5 L125: "a bin width of 10 cm." Not sure if this will affect the results greatly, but the bin size should probably be larger than the instrument uncertainty. Also, the bin width looks like 20 cm in the figures. Can you clarify?

p6. Eq (1):

Shouldn't this be $h(t_{i+1}) = ...$? Or at least the index of the left-hand side should be the same as the one for the denominator on the right-hand side. The summation index should also be replaced with something else than i, or instead you can replace $A(t_{i+1})$, $h(t_{i+1})$ with $A(t_n)$, $h(t_n)$ if $n = 30$ as mentioned in text.

My reasoning is as follows:
If we assume ice volume conservation over the whole 30 day period, then we have:

$$A(0)h(0) + \sum_k A(t_k)\Delta h_{th}(t_k) = A(t_{30})h(t_{30}) \text{ ; with } k = [0:29]$$

where $A(t_{30})$, $h(t_{30})$ is the final area and thickness of ice after 30 days, $\Delta h_{th}(t_k)$ = thermodynamical growth between $t_k$ and $t_{k+1}$ (k is an index representing the days of integration) and $A(0)$,$h(0)$ is the initial ice thickness and concentration which are in fact both zero. So we have:

$$\sum_k A(t_k)\Delta h_{th}(t_k) = A(t_{30})h(t_{30}) \text{ ; with } k = [0:29]$$

or, if we pose $n = 30$ as mentioned in the text, then:

$$h(t_n) = \sum_k (A(t_k)\Delta h_{th}(t_k))/A(t_n) \text{ ; with } k = [0:n-1]$$

We can also stop the integration before the end of the 30 day period. In this case, we have: $h(t_{i+1}) = \sum_k (A(t_k)\Delta h_{th}(t_k))/A(t_{i+1}); with k = [0:t_i]$

p7 L175: "Outliers in the velocity data were reduced by a 3x3 point running median filter covering an area of 2.1x2.1 km."
I am not super familiar with signal pre-processing filters, but does this filter simply smooth the drift to "reduce" the outliers, or does it remove the outliers? If the drift is smoothed, then it will also affect deformation calculations later, which will then have an effect on the reconstructed thickness.

p7 L187: "For deformations in which velocities and their gradients are small in comparison to the reference length scale, the strain rates can be linearized and transformed into two invariants of the 2D strain rate tensor"
I a not sure why this specification is needed. The shear rate and divergence can be written in terms of the trace and determinant of the 2D strain rate tensor (i.e. its invariants) making them also invariants. This is true regardless of their magnitude compared to the scale of measurements.

p7 L188: "We calculated the spatial derivatives from the averaged velocity fields"
Averaged temporally and/or spatially? I don't think this is specified in Section 2.4. Also, for consistency, you should make sure that these integrals are calculated only if the positions/drift values are obtained at similar times in your data set. It is also not clear what trajectories you are using for the positions (x,y) in the integral formulation of the strain rates. Are you seeding drifters at the cell's corner and integrating their trajectories in time using the gridded drift data set described in Section 2.4? Maybe I did not understand the format of the drift data set of Section 2.4... is it a list of trajectories with positions and velocity or is the resulting product a gridded velocity field? So far, I understood that your final drift data set is a gridded product.

p8 L205: You should mention what is the typical time interval used to obtain the Lagrangian trajectories/deformations here.

p8 L218: "deviation of the reconstructed trajectory"
I am confused now which trajectories we are talking about. The ones used to obtain the drift field that is then used to derive the deformation field, or the Lagrangian trajectories that were reconstructed backwards from the AEM tracks?

p8 L218: I am used to the term "tracking error" for the error resulting from incorrect pattern matching between two satellite images used for deriving the drift field (which will then affect the deformation estimates). But I think here you are talking about the Lagrangian position uncertainty that results from uncertainty in the drift field you used to integrate the Lagrangian trajectories backwards, and not a mis-match of patterns at

the pixel-level in the satellite images. Correct? Maybe you could add this distinction here. And maybe discuss the "other" tracking error (i.e. the one resulting from incorrect pattern matching in your algorithm for obtaining the drift data set)?

p9 L229: "Those values are averaged and saved."
So you use the averaged deformation within this uncertainty range as the deformation history along the Lagrangian paths?

p9 L239: "Hence, we calculated for every time step a forward and backward field and extracted deformation from both."
Add "a forward and backward DEFORMATION field..." Do you average both the backward and forward deformation estimates and use that as an averaged deformation field from which you extract the deformation history along the Lagrangian trajectories (which is then also averaged in the "tracking uncertainty circle")?

Section 2.7: This section could be shortened by going straight to Eq (5) which gives the continuity equation for the mean ice thickness as done in many dynamic-thermodynamic sea-ice models (e.g. Hibler 1979, Tremblay and Mysak 1997).

p10 Eq (5): Is the "dot" necessary in $div(u\dot{h})$ since u is a vector and h is a scalar?

p10 L266: "thermodynamic growth or melt..."
This should be a thermodynamic growth/melt RATE (i.e. $\Delta h/\Delta t$) to have units matching that of $dh/dt$ in Eq (5).

p11 L277: "Second, we approximated the thermodynamic ice growth within the grid cell in Eq. 5 by the growth of the undisturbed, thermodynamically growing ice (see Fig. 2 a,b)."
Which you estimate from the thermodynamical simulation described in Section 2.2?

p11 Eq(6): Again, the units do not match. The divergence term should be multiplied by $\Delta t$ (assuming $\bar{h}$ and $\Delta h_{th}$ are given in meters).

p11 L291: "Mean thickness converged to the first decimal after approximately 1000

iterations."
It is not clear what iterations are here. I thought you have 10 000 different represen-
tations of the possible ice thickness evolution along the trajectories. Why would these
"converge" to something?

p12 L310: "Since the thermodynamic growth is expected to be evenly over the polynya
region, it leads to rather uniform, level thicknesses of most of the surveyed ice."
This sentence is not clear and needs to be rephrased. It also seems to contradict the
sentence just above stating that most of the ice is in the thicker bins of the ITD due to
significant deformation over the whole polynya.

p12 L314: "Since the sole interpretation of mean and mode with regard to dynamic and
thermodynamic contributions may miss underlying processes, e.g. the potential con-
tribution of deformation to the observed modal thickness, we will investigate different
aspects in the following sections."
Is it necessary to discuss the above paragraph then?

p12 L320: "The modal thickness of the level ice is also identical to the mode of the
overall ITD, supporting our assumption that it represents best the thickness of thermo-
dynamically grown ice."
The order should be reversed: you don't need to assume anything if you show this
result first. This is really what allows you to speak of the overall modal thickness as
representative of the thermodynamical growth of level ice. This should be presented
first in the manuscript, or at least you can mention that "AEM results show that the
modal thickness is representative of the thermodynamical growth of level ice in the
polynya."

p12 Section 3.1: This section is titled "3.1 Overall, large-scale dynamic thickness
change due to area decrease of the closing polynya" and therefore hints at a link
with Eq (1) presented earlier, but it starts by describing mostly the thermodynamical
growth... The link with the area change and dynamical growth comes only later in a

subsection (3.1.1). Maybe you could have two subsections instead to separate the discussion around large-scale thermodynamical (3.1.1) and dynamical (3.1.2) growth and rename this section "3.1 Large-scale thermodynamical growth and dynamical thickness change due to area decrease of the closing polynya"?

p13 L333: "Divergence on March 30/31 and the occurrence of open water and very thin ice are visible in the divergence time series in Fig. 4 and in the ITD of the closing polynya (Fig. 1d, 2a), respectively."
There is no time series of divergence in Fig 4. If you refer to the time series of "Area extent FYI", then please introduce the relationship between the Area change and the divergence.

p14 Figure 4: In the label, please mention that the thermodynamic contribution (red) is obtained from a simulation, and not observations. The title could also be changed to "Dynamic and thermodynamic contributions to mean thickness from model and observations" or " Observed dynamical and simulated thermodynamical contributions to the mean ice thickness", or something like that. Are the contributions presented in (b) calculated for the trajectories only? At least, from the text in section 3.3 (p.20), it seems like the error bars are derived from the trajectories. Please specify.

15 L362: "we have also observed"
This has not been shown yet in the figures, so we don't know what this means. Change to "we also observe" and then refer to the figure where we can see these differences? Or present the observations for the different zones first, and then conclude about their regional variability.

p15 L365: "Based on variations of mean ice thickness along the profiles"
Can you give more details about how you separated the regions? i.e. an increase/decrease in the mean thickness along a moving average, or was it heuristic? Please specify.

p16 Figure 5: This figure should come after the current Figure 7 since it is discussed

mostly after Figures 6-7 are discussed.

p19 Figure 7: Are all 3 profiles included in these ITDs? Please specify.

p20 L456: "The mean thicknesses of all 715 trajectories or grid cells, respectively, were combined to compute the ITD of the modeled ice thicknesses."
I imagine that you only compiled the simulated thicknesses after the full integration of the trajectories was complete. Correct? Please specify it.

p21 L476: "Underestimation of observed thicknesses is larger in the less deformed Fast Ice and Inner Polynya zones."
I don't see this in FIg. 10. The modeled ice thickness is almost right on top of the observation sin the Fast Ice region in (c).

p22 Figure 10: For clarity, the dashed line for the modeled uncertainty should be the same colour as the data is belongs to (i.e. blue instead of black).

p25 L569: "Apart from those differences in the shape of the ITD, we have found that modeled mean ice thicknesses were generally smaller than the observed ones."
But the reported simulated mean thicknesses in Table 1 always fall within the uncertainty of the observations.

Editing suggestions:
————————————-

p1 L9: "characteristic" –> significant?

p1 L18: MYI was not previously defined.

p2 L32: "results in the presence of very variable thickness" weird formulation... maybe write "results in large ice thickness variations"?

p4 90:"along Lagrangian backward trajectories..." –> using Lagrangian trajectories integrated backwards until its initial formation?

p4 L92: "forced by time series of SAR derived, small-scale deformation" –> forced by the time series of SAR-derived small-scale deformation history...

p7-8 L189-90: "We relate the result to the center of the four grid cells." It is not clear what this means.

p9 L239: "considering both deformation estimates.." –> "considering both deformation estimates calculated with the forward and backward drifts"

p12 L310: "to be evenly" –> "to be evenly distributed"? or change to "to be the same"?

p13 L343: "we relate the overall area decrease of the polynya to the observed thickness change." using Eq. (1)?

p14 L350: "deformation within the polynya was regionally variable and distinctly different in certain zones" This means the same thing twice. Change to "deformation within the polynya showed significant regional variability"?

p15 L354: "the observed mean thickness" –> "the observed mean thickness along the AEM tracks"?

p15 L368: "The ice within each zone had similar mean thicknesses and similar ITDs." I think you mean "The ice within each zone had similar mean thicknesses and similar ITDs across all 3 profiles." or something like that, otherwise it sounds like the different zones have the same characteristics, which defeats the purpose of defining them.

p15 L381: "To do so, we derived ice drift trajectories of those 715 sections by means of the SAR imagery (Sect. 2.6)." The wording isn't clear. Remove and say in the next sentence: "The general motion of the 715 reconstructed trajectories (see Sect. 2.6) was South-South-East... " ?

p18 L416: "base" –> "basis"

p18 L419: "deformation parameters" add what they are in ()?

p20 L435: "of the simple volume-conserving model" –> add "(Sect. 2.7)" or Equation no.

p20 L437: "our thickness model" –> "this thickness model"

p20 L437-438: "they reproduce" –> "it reproduces"

p23 L480: "This way, within a month thermodynamics and dynamics restored a first-year ice cover that was almost as thick as the surrounding MYI." The wording is not clear. Please rephrase.

p23 L491: "Magnitude of deformation shapes ITD" –> "The magnitude of deformation shapes the ITD"

p23 L502: "0.39 cm" –> 39 cm or 0.39 m

p23 L506: "Taking advantage of the fact that the strongest deformation event left the largest impact on h" Not clear. Please rephrase.

p24 L518: "We test whether the here observed..." –> "We test whether the range of e-foldings observed here..."

p25 L558: "However, the derived ITDs are composed of mean thicknesses in the 1.4 km, long grid cells of our model, which are too large to resolve individual ridges or ridge clusters." Not clear.. Change to "However, the simulated ITDs are obtained with a spatial resolution of 1.4 km..." or something like that.
* * *

---

## Author Comment (AC2) · 4 Feb 2021

**Response to Amélie Bouchat (referee #2)**

Dear Amélie Bouchat,

Thank you for your very detailed and thorough review. We highly appreciate the effort you made to follow our thoughts and results. Your specific questions and suggestions will help us to improve the manuscript significantly. Please see our answers to your main and specific comments below. We will address your editing suggestions in the revised document.

Main comments:
1) Derivation of (1) drift, (2) deformation, and (3) trajectories:

    Thanks for expressing your questions so clearly. It helped us a lot to identify where our description in the manuscript lacks further details. (1) The output of the pattern matching algorithm is a regularly spaced velocity field with a spatial resolution of 700x700 m. (2) We calculate deformation from this velocity field. In the manuscript we chose the description of Green's theorem because this is a commonly known approach in the community, but we have realized now that this choice led to confusion, probably because it is normally used for buoys. We can simplify this approach for our gridded fields (see below). For example, the 8pt ring-integral (Eq. 3) for a subset of the u-component of the velocity field is:

    $$\mathbf{u} = \begin{pmatrix} u_1 & u_8 & u_7 \\ u_2 & u_9 & u_6 \\ u_3 & u_4 & u_5 \end{pmatrix}$$

    $\frac{\partial u}{\partial x} = \frac{1}{2A} [(u_2 + u_1)(y_2 - y_1) + (u_3 + u_2)(y_3 - y_2) + (u_4 + u_3)(y_4 - y_3) + (u_5 + u_4)(y_5 - y_4) + (u_6 + u_5)(y_6 - y_5) + (u_7 + u_6)(y_7 - y_6) + (u_8 + u_7)(y_8 - y_7) + (u_8 + u_1)(y_8 - y_1)]$

    Due to the regular grid, we can simplify.
    $0 = (y_4 - y_3) = (y_5 - y_4) = (y_8 - y_7) = (y_1 - y_8)$
    $\Delta y = -(y_2 - y_1) = -(y_3 - y_2) = (y_6 - y_5) = (y_7 - y_6)$
    $\Delta x = \Delta y$
    $A = (2\Delta y) \times (2\Delta x) = 4\Delta y \Delta y$
    with $\Delta y = 700$ m.

    Then, the derivate is:
    $$\frac{\partial u}{\partial x} = \frac{1}{8\Delta y \Delta y} \Delta y[(2u_6 + u_5 + u_7) - (2u_2 + u_1 + u_3)]$$
    $$= \frac{(2u_6 + u_5 + u_7) - (2u_2 + u_1 + u_3)}{8\Delta y}$$

    This is equivalent to calculating the convolution of u with a 3x3 Sobel kernel
    $$\mathbf{k} = \begin{pmatrix} -1 & 0 & 1 \\ -2 & 0 & 2 \\ -1 & 0 & 1 \end{pmatrix}$$
    and normalizing it by a factor of $8\Delta y$.

    We will clarify in the manuscript that deformation was calculated on the regularly spaced grid and not from the Lagrangian trajectories.

(3) For the trajectories, we do exactly as you suggested, we use the gridded velocity fields to integrate Lagrangian trajectories backward starting on the AEM profiles. We reconstruct the position of the trajectory for each time step by interpolating the regularly spaced velocity field to the location of the trajectory and adding the respective displacement (velocity*time). We will add a sentence at the beginning of Sections 2.4 to make this clearer. The trajectories were only used to identify the position of the ice within the deformation field, but not for calculating deformation.

2) Restructuring and text/grammar editing.
Thanks for your detailed notes on where we could restructure and also edit the manuscript for clarity. We will follow your suggestions and address them in the final revised manuscript.

Specific comments:

p3 L71: "Since modal thickness is considered a good first guess for the thickness of the thermodynamically grown" Reference? If this comes from your results, then mention it.
We add a reference to studies that provide details on this interpretation of the ITD (Wadhams 1994, Throndike, Parkison, Rothrock, 1992) and add a sentence in the introduction that explains why the modal thickness is a good first estimate for the thermodynamic growth. We remove this information previously found in the results (L309-310).

"The modal thickness of an ITD represents the thickness of the undeformed, level ice (Wadhams, 1994). Level ice forms by thermodynamic growth and melt whose atmospheric and oceanic forcing varies only on large spatial scales (Throndike, Parkison, Rothrock, 1992, Haas et al. 2008). In contrast, deformation results in strongly varying ice thickness and ice which is thicker than can be attained by thermodynamic growth. Consequently, we attribute the large difference between mode and mean to dynamic ice growth by deformation."

p3 Figure 1:
  a) Mention how the white trajectories were acquired. Insert: "20m ride" –> "20m ridge"
We modified the caption: "Sequence of white arrows illustrates four ice drift trajectories derived from daily velocity fields (section 2.6) representing the typical south-easterly ice movement during the convergent closing of the polynya.

b) The general low-resolution ice drift does not match the trajectories seen in (a). Maybe mention something about this?
For display reasons, we have chosen a coarse spacing of the arrows displaying the low-resolution drift in (b). That is why the change in drift direction from a coastward-directed (b) to a partly coast-parallel motion (a) cannot be identified in the low-resolution drift product. We extended the caption of b. "Overview map with monthly averaged, low-resolution sea ice drift in March 2018 (not showing the small-scale drift variability in the polynya)"

c) This panel does not seem useful. Could be removed and keep only the March 1st contour in red in (a)?
We agree and consider restructuring Figure 1 completely by increasing (a) and removing (c) and (d). Readers interested in the SAR images on March 1-March 30 are directed to the video supplement (https://doi.org/10.5446/50650).

d) "Combined ice thickness distribution of the FYI shown"... add along the AEM tracks?

Ok

p5 L125: "a bin width of 10 cm." Not sure if this will affect the results greatly, but the bin size should probably be larger than the instrument uncertainty. Also, the bin width looks like 20 cm in the figures. Can you clarify?

Thanks for pointing the inconsistency out to us. We make sure that figures and text agree upon this point in the revised manuscript. We will use 20 cm as bin width for the numbers stated in the text.

p6. Eq (1): Shouldn't this be h(ti+1) =...? Or at least the index of the left-hand side should be the same as the one for the denominator on the right-hand side. The summation index should also be replaced with something else than i, or instead you can replace A(ti+1), h(ti+1) with A(tn), h(tn) if n = 30 as mentioned in text.

My reasoning is as follows:
If we assume ice volume conservation over the whole 30 day period, then we have:
A(0)h(0) +Pk A(tk)_hth(tk) = A(t30)h(t30) ; with k = [0 : 29]
where A(t30), h(t30) is the final area and thickness of ice after 30 days, _hth(tk) = thermodynamical growth between tk and tk+1 (k is an index representing the days of integration) and A(0),h(0) is the initial ice thickness and concentration which are in fact both zero. So we have:
P
k A(tk)_hth(tk) = A(t30)h(t30) ; with k = [0 : 29]
or, if we pose n = 30 as mentioned in the text, then:
h(tn) =
P
k(A(tk)_hth(tk))=A(tn) ; with k = [0 : n ⍰ 1]
We can also stop the integration before the end of the 30 day period. In this case, we have: h(ti+1) =
P
k(A(tk)_hth(tk))=A(ti+1); withk = [0 : ti]

We went back to our code and agree with you that the index of the left-hand side should be the same as the one for the denominator on the right-hand side. We will modify Equation 1 to:

$$\overline{h(t_k)} = \frac{\sum_{k=1}^{n} A(t_{k-1}) \Delta h_{th}(t_{k-1}, t_k)}{A(t_k)}$$

with n=34. k=0 refers to the 25th of February 2018 and k=34 to the 31st of March 2018. Dynamic thickness changes between e.g., the 25[th] and 26[th] of February are counted at time step k=1 and so on.

p7 L175: "Outliers in the velocity data were reduced by a 3x3 point running median filter covering an area of 2.1x2.1 km."
I am not super familiar with signal pre-processing filters, but does this filter simply smooth the drift to "reduce" the outliers, or does it remove the outliers? If the drift is smoothed, then it will also affect deformation calculations later, which will then have an effect on the reconstructed thickness.

The median filter removes outlier but also smooths the velocity field. We have chosen a median filter because it is better than e.g., Gaussian smoothing at removing noise whilst preserving sharp gradients in the velocity field that represent the physical deformation zones. Physically incorrect

outliers in unfiltered data would bias the results even more. We plan to add a sentence in the revised manuscript.

p7 L187: "For deformations in which velocities and their gradients are small in comparison to the reference length scale, the strain rates can be linearized and transformed into two invariants of the 2D strain rate tensor"
I a not sure why this specification is needed. The shear rate and divergence can be written in terms of the trace and determinant of the 2D strain rate tensor (i.e. its invariants) making them also invariants. This is true regardless of their magnitude compared to the scale of measurements.
We agree and omit the statement.

p7 L188: "We calculated the spatial derivatives from the averaged velocity fields"
Averaged temporally and/or spatially? I don't think this is specified in Section 2.4.
Here we refer to the spatially averaged (3pt median filter) velocity fields. We will clarify this in the text.

Also, for consistency, you should make sure that these integrals are calculated only if the positions/drift values are obtained at similar times in your data set.
They are all calculated for the same period in time defined by the two satellite scenes that were used to derive the drift fields.

It is also not clear what trajectories you are using for the positions (x,y) in the integral formulation of the strain rates. Are you seeding drifters at the cell's corner and integrating their trajectories in time using the gridded drift data set described in Section 2.4? Maybe I did not understand the format of the drift data set of Section 2.4... is it a list of trajectories with positions and velocity or is the resulting product a gridded velocity field? So far, I understood that your final drift data set is a gridded product.
Thanks for pointing out here that further clarification is needed. Our drift fields are gridded fields that result from the pattern matching using the two SAR scenes. For the positions (x,y) in the integral formulation, we use the grid points of our gridded velocity fields. This results in a regularly gridded deformation field. Please see also our detailed answer to your general comments on the derivation of drift, deformation, and trajectories.

p8 L205: You should mention what is the typical time interval used to obtain the Lagrangian trajectories/deformations here.
We add "For each time step, which was typically one day, we extracted …"

p8 L218: "deviation of the reconstructed trajectory"
I am confused now which trajectories we are talking about. The ones used to obtain the drift field that is then used to derive the deformation field, or the Lagrangian trajectories that were reconstructed backwards from the AEM tracks?
Thanks for pointing out that further clarification is needed. Please see our answer to your general comments on the derivation of drift, deformation, and trajectories. Deformation was calculated from the gridded drift fields. The term trajectory always refers to the Lagrangian trajectories that were reconstructed backward from the AEM tracks.

p8 L218: I am used to the term "tracking error" for the error resulting from incorrect pattern matching between two satellite images used for deriving the drift field (which

will then affect the deformation estimates). But I think here you are talking about the Lagrangian position uncertainty that results from uncertainty in the drift field you used to integrate the Lagrangian trajectories backwards, and not a mis-match of patterns at the pixel-level in the satellite images. Correct? Maybe you could add this distinction here. And maybe discuss the "other" tracking error (i.e. the one resulting from incorrect pattern matching in your algorithm for obtaining the drift data set)?

Thanks for pointing out this confusion and for suggesting a better term. Yes, we describe what you termed "Lagrangian position uncertainty". We will modify L. 218:

"The tracking error accounts for the deviation of the reconstructed trajectory from the true one due to erroneous pattern matching. Hollands and Dierking (2011), e.g., found tracking errors between 0.8 and 1.6 pixels (their Tables 3 and 4, standard deviations), for pixel size of 50 m this corresponded to 40-80 m.
In the case of trajectories in an inhomogeneous velocity field, there is an accumulated trajectory position error that describes that a deviating trajectory results in the extraction of deformation that was not experienced by the surveyed patch of ice in reality, but by ice nearby. We estimated the accumulated position error from manual tracking of MY ice floes that were located in the polynya (see Fig. 1a). After the first step, the position error was of magnitudes between 51 and 210 m, which at some places is already larger than the expected tracking error. At the end of the tracking (March 1), the magnitudes of the accumulated trajectory position error are significantly larger (1050-2150m). Hence, the tracking error can be neglected."

p9 L229: "Those values are averaged and saved."
So you use the averaged deformation within this uncertainty range as the deformation history along the Lagrangian paths?
We took all deformation values from the forward and backward deformation field and average them for the deformation history along the Lagrangian paths. We will change this in the manuscript and write "To account for this spatial uncertainty, we extracted divergence, shear and total deformation from all deformation cells with their center points falling into the uncertainty circle (Fig. 2 c). The averaging included the individual deformation magnitudes both for the uncertainty circles along the forward and backward trajectory."

p9 L239: "Hence, we calculated for every time step a forward and backward field and extracted deformation from both."
Add "a forward and backward DEFORMATION field..."
Ok.

Do you average both the backward and forward deformation estimates and use that as an averaged deformation field from which you extract the deformation history along the Lagrangian trajectories (which is then also averaged in the "tracking uncertainty circle")?
First, we extract all values (from the backward and forward deformation field). Second, we averaged the extracted values to get a value for the deformation history. Please see our answer to Question Line 229

Section 2.7: This section could be shortened by going straight to Eq (5) which gives the continuity equation for the mean ice thickness as done in many dynamic thermodynamic sea-ice models (e.g. Hibler 1979, Tremblay and Mysak 1997).
p10 Eq (5): Is the "dot" necessary in div(u_h) since u is a vector and h is a scalar?
Thanks for suggesting how to shorten the methods. We considered this but would like to keep it because otherwise, we need to re-introduce the redistribution function in the discussion (L. 512). We can omit the "dot".

p10 L266: "thermodynamic growth or melt..."
This should be a thermodynamic growth/melt RATE (i.e. $\partial h/\partial t$) to have units matching
that of dh=dt in Eq (5).
Thanks for pointing out this inconsistency. We will unify the equations, units, and text, also related
to your remark on Eq (6).

p11 L277: "Second, we approximated the thermodynamic ice growth within the grid
cell in Eq. 5 by the growth of the undisturbed, thermodynamically growing ice (see Fig.
2 a,b)." Which you estimate from the thermodynamical simulation described in Section 2.2?
Yes. We will add half a sentence there.

p11 Eq(6): Again, the units do not match. The divergence term should be multiplied by
$\partial t$ (assuming $\partial hand$ $\partial hth$ are given in meters).
Yes, this is exactly what we did. We are sorry, that we have lost the factor Δt during our writing
process.

p11 L291: "Mean thickness converged to the first decimal after approximately 1000 iterations."
It is not clear what iterations are here. I thought you have 10 000 different representations
of the possible ice thickness evolution along the trajectories. Why would these
"converge" to something?
With this sentence, we wanted to express that 10 000 are a sufficiently large subset of all the
potential realizations. Already after 1000 realizations, the mean thickness calculated from all the
individual realizations is not changing any longer to the first decimal. We rephrase:

"To account for the tracking uncertainty, we created for each of the 715 trajectory random
combinations of the potentially experienced deformation within the uncertainty circles described in
Sect. 2.6.1. For each time step, we randomly choose one of the observed divergence states and
calculated mean thickness change along each trajectory. We repeated this 10.000 times and
calculated mean thickness and standard deviation as uncertainty from the resulting 10.000 thickness
estimates. For almost all of the 715 trajectories mean thickness changed little already after the first
1000 computations."

p12 L310: "Since the thermodynamic growth is expected to be evenly over the polynya
region, it leads to rather uniform, level thicknesses of most of the surveyed ice."
This sentence is not clear and needs to be rephrased. It also seems to contradict the
sentence just above stating that most of the ice is in the thicker bins of the ITD due to
significant deformation over the whole polynya.
Thanks for letting us know that this is confusing. We only refer here to the level ice thickness and
would like to make the point that the thickness of the level ice is rather uniform. We will rephrase
this, e.g.: "Since the thermodynamic growth of level ice is expected to be equal everywhere in the
polynya region, it leads to rather uniform, level thicknesses of most of the surveyed ice."

p12 L314: "Since the sole interpretation of mean and mode with regard to dynamic and
thermodynamic contributions may miss underlying processes, e.g., the potential contribution
of deformation to the observed modal thickness, we will investigate different
aspects in the following sections."
Is it necessary to discuss the above paragraph then?
The sentence only limits the reliability of the interpretation of the modal value, but not the reliability
of the other parameters describing the ITD (maximum thickness, e-folding, open water fraction, ...).

We understand that this sentence is misleading at this location, which is why we move it to the next paragraph and rephrase it.

p12 L320: "The modal thickness of the level ice is also identical to the mode of the overall ITD, supporting our assumption that it represents best the thickness of thermodynamically grown ice."
The order should be reversed: you don't need to assume anything if you show this result first. This is really what allows you to speak of the overall modal thickness as representative of the thermodynamical growth of level ice. This should be presented first in the manuscript, or at least you can mention that "AEM results show that the modal thickness is representative of the thermodynamical growth of level ice in the polynya."
We agree that "assumption" is not the right choice of words here. As indicated in our answer to L. 71, we build here on previous results and thus rephrase this part to: "The modal thickness of the level ice is also identical to the mode of the overall ITD, confirming the results of previous studies (e.g., Haas et al. 2008) that the modal thickness represents the thickness of thermodynamically grown ice well."

p12 Section 3.1: This section is titled "3.1 Overall, large-scale dynamic thickness change due to area decrease of the closing polynya" and therefore hints at a link with Eq (1) presented earlier, but it starts by describing mostly the thermodynamical growth... The link with the area change and dynamical growth comes only later in a subsection (3.1.1). Maybe you could have two subsections instead to separate the discussion around large-scale thermodynamical (3.1.1) and dynamical (3.1.2) growth and rename this section "3.1 Large-scale thermodynamical growth and dynamical thickness change due to area decrease of the closing polynya"?
Thanks for suggesting how to restructure this part. We will follow your advice.
p13 L333: "Divergence on March 30/31 and the occurrence of open water and very thin ice are visible in the divergence time series in Fig. 4 and in the ITD of the closing polynya (Fig. 1d, 2a), respectively."
There is no time series of divergence in Fig 4. If you refer to the time series of "Area extent FYI", then please introduce the relationship between the Area change and the divergence.
Thanks for pointing this out. You are right, we referred to the time series of "area extent FYI" and will rephrase L33.

p14 Figure 4: In the label, please mention that the thermodynamic contribution (red) is obtained from a simulation, and not observations. The title could also be changed to "Dynamic and thermodynamic contributions to mean thickness from model and observations" or " Observed dynamical and simulated thermodynamical contributions to the mean ice thickness", or something like that.
We will follow your suggestions.

Are the contributions presented in (b) calculated for the trajectories only? At least, from the text in section 3.3 (p.20), it seems like the error bars are derived from the trajectories. Please specify.
The thermodynamic contributions are the output of the MITgcm runs. The dynamic contributions (starting from the 2nd of March) are from the trajectories only. We will specify this in the caption. We will also mark in Fig. 4b that the first calculation of dynamic thickness change is for the period March 1 to March 2.

15 L362: "we have also observed"
This has not been shown yet in the figures, so we don't know what this means. Change to "we also observe" and then refer to the figure where we can see these differences?
We will change the sentence as you suggested.
Or present the observations for the different zones first, and then conclude about their regional variability.

p15 L365: "Based on variations of mean ice thickness along the profiles"
Can you give more details about how you separated the regions? i.e. an increase/ decrease in the mean thickness along a moving average, or was it heuristic?
Please specify.
Thanks for asking. We realized that our text lacks some information here. Our decision was based on several aspects of thickness and deformation history. 1: thickness: the running mean of the ice thickness (see Fig. 1), the occurrence of level ice, and the frequency and thickness of the deformed ice (Fig. 6, Tab. 1). 2: deformation history of the ice: path length and origin of the trajectories (Fig. 5a) and the timing, magnitude, and type of deformation that the ice experienced (see Fig. 5, video supplement). For example, ice in the Fast Ice zone is the thinnest and has the largest percentage of level ice. It had short travel paths and experienced weak deformation only during the early deformation phase (Fig. 5b-d). All four zones with their characteristic thickness and deformation properties are described in the text. We rephrase in the text: "Based on the degree of deformation, as well as mean and variation of ice thickness along profiles, we separated four regions with clearly different deformation histories. To be more specific, the criteria for separation were: (1) the running mean of the ice thickness (see Fig. 1), the areal fraction of level ice, and the frequency and thickness of deformed ice (Fig. 6, Tab. 1). (2) the deformation history of the ice, i.e. path length and origin of the trajectories (Fig. 5a), and timing, magnitude, and type of deformation that the ice experienced (see Fig. 5b-d, video supplement)."

p16 Figure 5: This figure should come after the current Figure 7 since it is discussed mostly after Figures 6-7 are discussed.
We follow your advice but this results in referencing the original Fig. 5 (p.14, large-scale drift in the insets) before Figures 6-7.

p19 Figure 7: Are all 3 profiles included in these ITDs? Please specify.
Yes, all 3 profile lines are displayed in this figure. We will specify in the caption "ITDs of all four FYI zones of all three AEM lines on March 30/31."

p20 L456: "The mean thicknesses of all 715 trajectories or grid cells, respectively, were combined to compute the ITD of the modeled ice thicknesses." I imagine that you only compiled the simulated thicknesses after the full integration of the trajectories was complete. Correct? Please specify it.
Thanks for pointing out the unclear wording. We rather mean the "integrated thickness of all 715 trajectories on March 30/31 were combined to compute the ITD of the modeled ice thicknesses."

p21 L476: "Underestimation of observed thicknesses is larger in the less deformed Fast Ice and Inner Polynya zones."
I don't see this in FIg. 10. The modeled ice thickness is almost right on top of the observation sin the Fast Ice region in (c).
We agree that in Fig. 10 c this is not visible. We refer here to the mean (see Tab. 1, last column). Since we do not conclude from this observation, we will omit the sentence to avoid confusion.

p22 Figure 10: For clarity, the dashed line for the modeled uncertainty should be the same colour as the data is belongs to (i.e. blue instead of black).
Thanks for this good hint! We will follow your advice.

p25 L569: "Apart from those differences in the shape of the ITD, we have found that modeled mean ice thicknesses were generally smaller than the observed ones."
But the reported simulated mean thicknesses in Table 1 always fall within the uncertainty of the observations.
Thanks for mentioning this, because we realized that more information is needed in the caption. In Tab. 1, the mean is given with its standard deviation which describes the spatial variability. The uncertainty of the thickness observations is assumed to be +/- 10 cm (L. 105), plus an additional 4 cm from the snow cover. We will add this information to the table.

Editing suggestions
Thank you for your effort in listing your editing suggestions that we will implement in the revised manuscript.

**References:**

Thorndike, A.S., Parkinson, C., Rothrock, D.A. (Eds.) (1992) Report of the sea ice thickness workshop, 19-21 November 1991, New Carrollton, Maryland. Polar Science Center, Applied Physics Laboratory, University of Washington, Seattle
(http://citeseerx.ist.psu.edu/viewdoc/download?doi=10.1.1.1018.8242&rep=rep1&type=pdf)

Wadhams, Peter (1994), Sea Ice Thickness Changes and Their Relation to Climate, In: The Polar Oceans and Their Role in Shaping the Global Environment, Geophysical Monograph, American Geophysical Union

Haas, C., A. Pfaffling, S. Hendricks, L. Rabenstein, J.-L. Etienne, and I. Rigor (2008), Reduced ice thickness in Arctic Transpolar Drift favors rapid ice retreat, Geophys. Res. Lett., 35, L17501, doi:10.1029/2008GL034457.

---

## Author Response (AR1)

**Response to Editor**

Dear Jenny,

Please find below our answers to the two reviewers. In our answers, we have included now line numbers referring to the revised manuscript for better orientation. Further changes in the manuscript not explicitly mentioned by the reviewers were motived by shortening and editing the manuscript to improve its readability. We have extended the "data availability" section substantially and included links to the AEM thickness and drift and deformation fields published now with Pangaea.

Best regards,

Luisa (on behalf of the authors)

**Response to Anonymous Referee #1**

Dear Referee #1:

Thank you very much for your time and effort you put into the detailed comments on our manuscript with the title "Linking sea ice deformation to ice thickness redistribution using high-resolution satellite and airborne observations". We believe that your suggestions will help us to improve the readability of our manuscript significantly. Please see below our answers (blue) to your comments (black) and changes in the text (green).

You asked for shortening the manuscript, but also suggested to add more information on several aspects. To find a balance between those competing demands, we decided to remove information where we think it is not strictly necessary for the main message of our manuscript, as described below. We removed:
- - The detailed description of the deformation calculation (as it only repeats existing literature). It is now found in the supplement.
  - The description of the multi-year ice (MYI) surrounding the polynya and Figure 3 b
  - Introduction: We linked the description of thermodynamic and dynamic thickness change with the one of the ITD to remove doubling of content in the introduction.
  - detailed description of the deformation history of the different zones that can be also seen on Fig.5 and the video supplement 2 (former L. 390-411)
  Detailed description of Fig. 10 (former 468-474).

At the same time, we also added more references in the manuscript to studies that provide the information you were asking for.

**Specific comments:**
  (1) Type of polynya:
  - Thanks for pointing out where essential information are still missing. We agree that it is very important to make clear that we studied the closing of a polynya that was created primarily dynamically (Moore et al. 2018). While air temperatures were rising above 0°C, Moore et al. 2018 showed that the polynya was a latent heat polynya, created by the divergent ice

motion, and the warmer surface air temperatures contributed only by reducing the sea ice production. Hence, we added a sentence on the type of the polynya in the introduction (l. 1 "an unusual, large, latent heat polynya", l 61: "of an unusual, latent heat polynya that…")

- However, we are aware that our manuscript is already long, which is why we added more and clearer reference to the preceding studies (Ludwig et al. 2019, Moore et al. 2018) that dealt with the formation history of the polynya instead of describing it in more detail in our manuscript.
- We replaced the description of the most likely origin, a "unusually strong and persistent atmospheric pattern", by its effect which where "unusually strong and persist northward winds over the Greenland Sea" (l. 63)
- The large-scale drift patterns associated with the opening and closing of the polynya are presented in detail by Ludwig et al. 2019, e.g. Figure 9 a, b. Here, the authors compared the unusual drift direction end of February with the long-term mean. We have referenced this publication at the end of the sentence (l. 65).

(2) Ice Type:

Thanks for asking for a clarification on the ice type. We follow your advice and differentiate between the ice surveyed by the campaign that comprised of both, young ice and MYI, and the ice for which we calculated deformation and modeled thickness, that was only young ice. First, we removed all information on the MYI outside the refrozen polynya. Second, to differentiate between the MYI floes and the young ice, we predominantly based our assumptions on their formation history which we could reconstruct by tracking the ice backwards in time. This way, we could distinguish between ice that had formed beginning of March and MYI that drifted into the open water /was located within the open water before. We combined this information with the thickness profiles and the backscatter of the SAR images on March 31/30.  We have rephrased this in see e.g., L 117 – L 124.:

(3) SAR analysis:

Thanks for your comments that helped us to identify unclear points. We add short statements based on our explanations in the manuscript. Regarding …

(1) start of drift tracking: For the start point of the tracking, we down-sampled the GPS coordinates of the airborne flight campaigns to 250 m. Gaps in the thickness observations made it necessary to increase the distance between the starting points which lead occasionally to distances of 350 m. No additional selection process based on ice type or similarly was done here. The tracking started at the down-sampled GPS coordinates. See L183-185 in the manuscript.

(2) Derivation of deformation: To calculate deformation from drift, we followed the approach widely used in literature, described in details by e.g., Kwok et al. 2003, Kwok et al. 2008, for a review: Dierking et al. 2020. As you pointed out, the manuscript is long which is why we tried to remain as concise and short as possible. We have rephrased Sect. 2.4 (drift and deformation from SAR) and moved additional derivations to the supplement (see L.160-179). Further, we added more references so that the reader can find more detailed information in the cited literature (L. 175).

(3) SAR backscatter values and the classification of the ice type:

The radar backscattering coefficients depend on frequency, polarization, incidence angle, and season (freezing, melting, and effects of melt-freeze cycles), hence also the thresholds between ice types vary. Also, the influence speckle and instrument at low backscattering levels noise has to be considered. In recent work, automated sea ice segmentation and classification is therefore carried out e.g., using statistical methods such as maximum likelihood decisions, or machine-learning methods such as neural networks. This is far beyond the scope of our study here. Grey tone variations are good proxies for separating various ice classes visually (a practice common also in operational

ice charting), in particular if complementary information is available, as in our case thickness properties and deformation history as described in lines L117-124. In this context, the qualitative description of "light" MYI and "dark" young ice in the caption of Fig. 1 was only used deliberatively to give the reader a quick guide for where to look for. In respect to the naming convention of the zones, the names (Fast Ice, Shear Zone, Inner Polynya, Northern Rim) were chosen to distinguish between the four zones. They only reflect one aspect of the deformation history. For example, ice in the Fast Ice zone became quickly immobile (red trajectories in Fig. 7a). The ice in the Shear Zone experienced strong shear during March 29-31 (Fig. 7d, L.339-341). For display of the shear fields, please see the video supplement (http://doi.org/10.5446/49540).

(4) Modal thickness:
Thanks for pointing this inconsistency out to us. We made sure that figures and text agree upon this point in the revised manuscript. We recalculated the modal thickness for a bin width of 20 cm.

(5) Data:
We have published the AEM ice thickness and high-resolution drift and deformation data in the data repository Pangaea and added the links to the data availability section at the end of the text (L585-L595). There, we also provided more details on the large-scale ice drift and Operation Icebridge data that we used.

(6) Language:
We carefully revised the use of Language in our manuscript and hope that we have improved its readability.

**Technical corrections:**

L11: Change "by drift tracking along Lagrangian backward trajectories" –> by tracking ice drift along reverse Lagrangian trajectories to go back in time Lagrangian ice drift trajectories backward in time
L9: We extended here a bit on the topic and rephrased this to "These closely corresponded to different deformation histories of the surveyed ice that we derived from Lagrangian ice drift trajectories backward in time. We constructed the ice drift trajectories from regularly gridded, high-resolution drift fields calculated from SAR imagery and extracted deformation derived from the drift-fields along the trajectories.

L16-17: Change "The computed ice thickness distribution resembles main characteristics like mode, e-folding, and width of the observed distribution" –> The computed ice thickness distribution resembles the main characteristics of the observed distribution including mode, e-folding, and width.
Done, L17

L17: clarify what is mean by width, do you mean "full width half maximum"?
Yes, done, L18

L19-20: Change "The similar first- and multi-year ice mean thicknesses elude to the large amount of deformation experienced by the closing polynya" –> the similarity between

the mean thickness of first- and multi-year ice suggests the scale of deformation that occurred within the closing polynya.
We removed this since we do not discuss MYI any longer.

L30-32: the use of "mean peak depth" to describe keel depths does not sound right. I suggest rewording this using a term such as modal depth. Also, I appreciate that there's a lot of statistics presented in Strub-Klein and Sudom (2012) and so I would suggest here that you clarify which observational statistics you are quoting. I believe they relate to "maximum" sail heights and keel depths. Here you might also refer to Duncan et al. (2020) which updates and expands upon the Strub-Klein and Sudom (2012) work, and found that the average maximum sail height was 2.01 m for >17,000 ridges formed in FYI.

Thanks for the additional reference. From Strub-Klein and Sudom (2012) we cited the Arctic maximum sail heights and keel depths and referred to the largest value measured and the average (their Table 2). Since we do not want to extend the length of the manuscript, we shortened this paragraph and directed the reader to the two publications. We rephrased L34: For example, two studies reported Arctic mean maximum sail heights of 2.01-2.1m (Strub-Klein and Sudom, 2012; Duncan et al., 2020).

L45-46: Provide a suitable reference to support this statement.
We added another reference to Rampal et al., 2009.

L52: Remove "so far" since it is not needed in this sentence. Also change "changed" –> changes in
Done

L65: the text refers to the date of the maximum extent of the polynya as February 25th, but the figure (Fig 1c) shows a SAR image from March 1st.
We largest extent was on Feb 25, but we displayed March 1 in Fig. 1, because this was the date at which our drift and deformation calculations started. We have removed now this panel of Fig. 1.

L66: change "unprecedented" –> unusually
Done

L68: revise "coastward directed winds" with a vector direction
L.68: We added a vector direction: "coastward-directed, i.e., southward winds",

L69-73: Is this a statement of work described elsewhere (i.e., previously published) or a summary of the work we are about to read about in this manuscript? If the former, provide a suitable reference; if the latter, move this statement from the introduction to the conclusion section.
We rephrased this paragraph in the following. We removed the mean and modal thickness from the introduction. We provided references for the observation that modal thickness is a good approximation of the thermodynamic thickness and moved it to L29.

L96: for completion I suggest you include the year to any dates provided within the text, in addition to month and day.

We considered this, but decided now to mention the year of the data acquisition in the introduction, data description, and figure captions. In the following text, however, we avoid repeating the year every time since it appears to be very formalistic.

L105: by "peak" do you mean modal, or maximum?
We rephrased: "The footprint smoothing underestimates the maximum ridge thickness but overestimates the ridge flanks." L100-101

L109-111: What Operation IceBridge data product for snow thickness is used here?
We use the Operation IceBridge (OIB) Sea Ice Freeboard, Snow Depth, and Thickness Quick Look data and indicated this now in L104 and in the data availability section.

What is the uncertainty associated with an airborne snow thickness observation of 4 cm?
Since there are no ground observations for this particular study site, we cannot present uncertainty estimates specifically tailored to our study case. However, King et al. (2015) found that OIB Sea Ice Freeboard, Snow Depth, and Thickness Quick Look data underestimate snow thickness by 5.3 cm in their study. We have quoted their results in our manuscript (L109).

L109-111: How much does uncertainty in snow thickness contribute to errors in the attribution of thermodynamic processes to the overall ITD?
In this paragraph we only described the contribution of the snow cover to the observed total thickness, since the laser signal is from the snow surface.
We are aware that snow has a strong effect on the thermodynamic growth of thin ice and have attributed the variability of the level ice thickness partly to this effect (see L. 280-281). For a more detailed answer on how uncertainty in snow redistribution affects our results, please see our answer to your question related to your question about L. 159-160.

L113-115: how does this assumption impact the uncertainty associated with the AEM thickness estimates, relative to that stated on L105?
We add a sentence in L.113-114: "The uncertainty of the AEM principle (0.1 m) and the snow thickness (0.04 m) add up to ,0.14m uncertainty of the AEM ice thickness measurements. "

L118-119: unfortunately this is not possible for the reader since there is no colour scale provided with the SAR data shown in Fig 1, nor is it clear what the units are.
The boundary of the young ice – MYI is identified visually based on the grey tone contrast. We found that the edge of the polynya, marked by the sharp transition of darker and brighter grey tones, was easy to identify in almost all images. We worked on backscatter data given in dB-scale, where we applied a histogram stretch for an improved visual interpretation. The knowledge of grey scale and related units is not required in this context.
We added a half sentence about the (stretched) backscatter values in dB-scale in the caption of Fig. 1 ("shown in dB-scale"). We provided an additional video supplement here (https://doi.org/10.5446/50650) to let the reader evaluate the manually created outlines.

L121-122, L149-151: I'm curious to understand what is meant by "visual interpretation of the SAR backscatter signature". Is this done using linear pixel greyness values, and therefore subjective, or by applying thresholds to the SAR data in dB? How are MYI floes defined and excluded?
Please see our answers to your general comments to the SAR analysis and the ice type, as well as the answer to your question related to L118-119. Please also refer to the manuscript in L 120-124.

L125: according to the figures, the bin width is 20 cm.

Thanks for pointing this out. We have changed this in the text to 20 cm.

L166: can you explain what is meant by "radar intensity"?
We rephrased to "radar backscatter coefficients" (L162)

L127: change "Large e-folding and FWHM" –> to large values of e-folding and FWHM

Done

L142: can you briefly describe what is meant by "two-category, zero-layer thermodynamics"?

A two-category, zero-layer thermodynamics refers to a model set-up that simulates only ice thickness and concentration, i.e. its thickness categories only consist of zero thickness (open water, given by the concentration) and mean thickness. Although there are also multi-category thickness distribution sea-ice models, the 2-category model based on Hibler (1980) is still most widely used and has proven to result in realistic simulations. The "zero-layer thermodynamics" refer to the fact that the model does not consider storage of heat in the ice. This two-category, zero-layer thermodynamics model set-up complies with a standard version of the MITgcm. Therefore, we provided several references that describe the thermodynamics of the MITgcm. We believe that adding more details in the text would unnecessarily prolong the manuscript.

L153: change "adding up" –> summing

We rephrased this paragraph and removed this part.

L153-154: does 't' represent thickness, or time?

It represents time. We clarified in L243.

L159-160: Snow depth on thin ice has a large control on thermodynamic ice growth. How was thermodynamic growth impacted by snow thickness changes (and/or snow redistribution) over 30 days? Does imprecise knowledge of this impact the conclusions drawn?
The timing of snow fall events was considered in the thermodynamic modelling by forcing with precipitation from the ERA-5 reanalysis data. However, the local snow redistribution due to the wind is dependent on the ice surface topography and cannot be considered explicitly. Hence, individual trajectories (Fig. 10) include an uncertainty in the thermodynamic growth due to unknown snow cover variations, which contributes to the deviations between observed and modelled thickness. However, we based our conclusions on regionally averaged trajectories. On those larger spatial scales, we are confident that our thermodynamic estimates are valid thanks to 1) the agreement of the estimated overall thickness from the area change and the observed thickness (section 3.1) and 2) the agreement between the modal thickness of the ice and the modelled thermodynamic ice thickness in the four subregions (Tab. 1). Thus, we think that the imprecise knowledge of the snow redistribution does not impact our conclusions.

L174: Provide an example of the derived ice drift data so that the reader may evaluate the results for reasonableness

Three examples of ice drift data are displayed in Fig. 5 (arrows). We can now also provide the link to the video supplement (http://doi.org/10.5446/49540) where arrows indicate drift speed and direction. We have submitted all drift + deformation data to the data repository Pangaea where the reader may download and evaluate them as soon as it is published there.

L182: what are the variables u and v in equation 2?
They represent the x and y components of the velocity. We clarified in L.171-172.

L171: Provide an example of the derived ice deformation data so that the reader may evaluate the results for reasonableness
Three examples of ice deformation data are displayed in Fig. 5 b-d (colours) and in the video supplement (http://doi.org/10.5446/49540).

L191: Did the authors compute uncertainty in the derived divergence, shear and deformation fields?
We are aware of the different sources of uncertainty of deformation parameters, which we describe in section 2.6.1, where we explain how those propagate into our final modelled ice thickness. We did not compute uncertainty of the single deformation estimates since in particular the estimation of the tracking error requires an effort beyond the scope of this study, and directly applicable equations for the boundary definition errors have not been published yet. The uncertainty of the *drift* depends on the local conditions, and is difficult to judge for thinner, easily deformable ice. Therefore, we decided to provide a reference value based on the manual tracking of the MYI floes (described in l. 220). As major point, however, we assume that the uncertainty in thickness changes is more strongly influenced by the position errors of the reconstructed paths of ice drift than by the uncertainty of the deformation parameters.

L195: "In order to coincide with the surveyed ice" – this is quite awkward, consider rewording.
We rephrased this paragraph, see L. 181-182

L200: what is the delta time between the airborne survey and image acquisition?
The time difference was between 2 and 6 hours and we corrected for the respective drift. Also, we evaluated our correction by visually comparing the location of leads (from AEM) with the SAR images. We added in the text (L186).

L202: Change "at the time step before" –> at the previous time step
We rephrased this sentence, see 188-189

L210: how is the reliability of the tracking algorithm quantified?
We based our decision regarding the use or rejection of results on the criteria described in Hollands et al. 2015, that are the difference in backmatching and the confidence factor (CFA). The CFA consists of several quality criteria in respect to the texture of the SAR image and the correlation itself. For details we refer to their publication.

L212-213: another very awkward sentence that is hard to follow - consider rewording
Since this was only a summary, we removed the sentence.

L220-224: Can you show this assessment?
We have provided an additional figure in the supplement that presents the analysis of the reference tracks. The figure shows the difference between reference track and the calculated trajectory for

each time step. Also indicated are the mean of the differences at the first and the last time step. The dashed black line gives the assumed uncertainty for each time step as described in L. 206

L225: change "sums up" –> accumulates
We rephrased this sentence, see L. 208.

L291, L293: is there a reason why the number of combinations and iterations are reported to three significant figures? Are combinations and iterations the same thing?
Yes, combinations and iterations are the same. We rephrased this and moved it to Sect. 2.5. We are now only referring to 10 000 combinations, L. 223-224.

L306 and L332: how is open water fraction computed?
We considered all thickness observations that were thinner than the instrument thickness of 10 cm as open water. This is stated in L. 126

L307: "with most of the ice" – state the % here
Done, L. 271

L310" Change "evenly" –> even, or is there a word missing here?
Rephrased, see L. 272-273

L310: is uniform thermodynamic growth "expected" over such a large area? I think it is assumed (but not necessarily expected) since we do not know about snow distribution.
We rephrased this to "approximation", see L. 271.

L312-313: "Deformation has led to the presence of a long tail of the distribution up to 20 m thickness" – But the scale in in Fig 3a only shows data to 8 m. What % of samples in the tail span 8 m to 20 m? Consider adding AEM profile data here to substantiate this statement (similar to the data shown in Fig 6).
We provide both profiles in the supplement. Please also note that the filtered thickness data was presented in Figure 10.

L320: Are the modes identical? Looking at Fig 3a, it appears the mode of the level ice thickness distribution is less than that of the 'complete' ice thickness distribution.
We have recalculated the modes and changed this in the text.

L335: Isn't the modal thickness 1.95 m (Fig 3b)?
This value was calculated based on a smaller bin size than displayed in Fig. 3b. However, we have removed the MYI from our publication.

L560: Did the authors consider the ice thickness distribution from CryoSat-2 for this region so as to substantiate their statement?
We believe that there was a misunderstanding in how we intended this reference to the CryoSat-2 ice thickness distributions. We did not mean to say that Kwok (2015) analysed CryoSat-2 data from the former polynya. Rather we wanted to express that both in our approach, as well as in Kwok's (2015) radar altimetry, ITDs are compiled from highly averaged data with a comparable averaging length of 300 to 1400 m. We have reformulated this sentence to make this clearer (L489-493)

L564: double "of"
Done

L586: change "we suggest to couple the SAR deformation retrievals with: : :" –> we suggest coupling deformation statistics retrieved from SAR analysis with : : :.
Done

L630, 632: change "coverage" –> cover
Done

L645-655: this sentence is quite hard to follow - consider rewording
We slightly rephrased the sentence.

Figure 1:

In 1(a) the drift trajectories (thin white lines with arrows) are not defined in the legend.
We added them.
What is the reasoning behind the uneven increments used in the color scale for ice thickness? Why, for example, is the majority of ice (according to figure 1d) combined, and represented by only one colour increment (light green) while thicker ice is divided into four increments ranging between 0.15 and 0.24 m in thickness?
We have chosen the colour scale to stress the differences in the four zones (Fast Ice, Shear Zone, Inner Polynya, Northern Rim). As described in Tab. 1 the mean of the four zones varies between 1.4 and 2.4 m. This is why we have chosen this non-linear colour scale. We added a half sentence about this in the caption.
A scale bar for the blue arrows in 1 (b) is required, showing drift magnitude.
Done
Change "stippled" to "dashed" in the figure caption.
Done
Do the data in 1(a) and (d) show ice thickness (as stated in the figure caption) or ice thickness + snow depth (as stated on L98-99)?
They state snow + ice thickness. We have changed this in the caption and the legend. Please note also L. 113

Figure 3: Indicate in the axis labels for (a) and (b) whether you show ice thickness + snow depth or ice thickness only. From my reading of the text I think (a) is the distribution of ice thickness, but (b) is the distribution of ice + snow thickness. Is the "complete" thickness distribution shown in Fig 3a repetition of the data shown in Fig 1d? If so, remove one of these duplicate figures.
Thanks for pointing out that this caused confusion. We changed the label to make clear that it is snow + ice thickness. We also removed Fig. 1d.

**References used in this reply:**

Dierking, W., Stern, H. L., and Hutchings, J. K.: Estimating statistical errors in retrievals of ice velocity and deformation parameters from satellite images and buoy arrays, The Cryosphere, 14, 2999–3016, https://doi.org/10.5194/tc-14-2999-2020, 2020.

Hollands, T. and Dierking, W.: Performance of a multiscale correlation algorithm for the estimation of sea-ice drift from SAR images: initial700results, Annals of Glaciology, 52, 311–317, https://doi.org/10.3189/17275641795931462, 2011

Hollands, T., Linow, S., and Dierking, W.: Reliability Measures for Sea Ice Motion Retrieval From Synthetic Aperture Radar Images, IEEE Journal of Selected Topics in Applied Earth Observations and Remote Sensing, 8, 67–75, https://doi.org/10.1109/jstars.2014.2340572, 2015.

Kwok, R.: Sea ice convergence along the Arctic coasts of Greenland and the Canadian Arctic Archipelago: Variability and extremes (1992-2014), Geophysical Research Letters, 42, 7598–7605, https://doi.org/10.1002/2015gl065462, 2015.

Kwok, R., Cunningham, G. F., and Hibler, W. D.: Sub-daily sea ice motion and deformation from RADARSAT observations, Geophysical Research Letters, 30, 2218, https://doi.org/10.1029/2003gl018723, 2003.

Kwok, R., Hunke, E. C., Maslowski, W., Menemenlis, D., and Zhang, J.: Variability of sea ice simulations assessed with RGPS kinematics, Journal of Geophysical Research, 113, https://doi.org/10.1029/2008jc004783, 2008.
Ludwig, V., Spreen, G., Haas, C., Istomina, L., Kauker, F., and Murashkin, D.: The 2018 North Greenland polynya observed by a newly introduced merged optical and passive microwave sea-ice concentration dataset, The Cryosphere, 13, 2051–2073, https://doi.org/10.5194/tc-13-2051-2019, 2019.

Moore, G. W. K., Schweiger, A., Zhang, J., and Steele, M.: What Caused the Remarkable February 2018 North Greenland Polynya?, Geophysical Research Letters, 45, 13,342–13,350, https://doi.org/10.1029/2018gl080902, 2018.

**Response to Amélie Bouchat (referee #2)**

Dear Amélie Bouchat,

Thank you for your very detailed and thorough review. We highly appreciate the effort you made to follow our thoughts and results. Your specific questions and suggestions will help us to improve the manuscript significantly. Please see our answers to your main and specific comments below.

Main comments:

1) Derivation of (1) drift, (2) deformation, and (3) trajectories:

Thanks for expressing your questions so clearly. It helped us a lot to identify where our description in the manuscript lacks further details. (1) The output of the pattern matching algorithm is a regularly spaced velocity field with a spatial resolution of 700x700 m. (2) We calculate deformation from this velocity field. In the manuscript we used the description of Green's theorem because this is a commonly known approach in the community, but we have realized now that this choice led to confusion, probably because it is normally used for buoys. We can simplify the Green's approach for our gridded fields (see below). For example, the 8pt ring-integral (Eq. 3) for a subset of the u-component of the velocity field is:

$$\mathbf{u} = \begin{pmatrix} u_1 & u_8 & u_7 \\ u_2 & u_9 & u_6 \\ u_3 & u_4 & u_5 \end{pmatrix}$$

$$\frac{\partial u}{\partial x} = \frac{1}{2A} \left[ (u_2 + u_1)(y_2 - y_1) + (u_3 + u_2)(y_3 - y_2) + (u_4 + u_3)(y_4 - y_3) + (u_5 + u_4)(y_5 - y_4) + (u_6 + u_5)(y_6 - y_5) + (u_7 + u_6)(y_7 - y_6) + (u_8 + u_7)(y_8 - y_7) + (u_8 + u_1)(y_8 - y_1) \right]$$

Due to the regular grid, we can simplify.

$$0 = (y_4 - y_3) = (y_5 - y_4) = (y_8 - y_7) = (y_1 - y_8)$$
$$\Delta y = -(y_2 - y_1) = -(y_3 - y_2) = (y_6 - y_5) = (y_7 - y_6)$$
$$\Delta x = \Delta y$$

$A=(2\Delta y) \times (2\Delta x) = 4\Delta y\Delta y$
with $\Delta y = 700$ m.

Then, the derivate is:
$$\frac{\partial u}{\partial x} = \frac{1}{8\Delta y\Delta y}\,\Delta y[(2u_6 + u_5 + u_7) - (2u_2 + u_1 + u_3)]$$
$$= \frac{(2u_6 + u_5 + u_7) - (2u_2 + u_1 + u_3)}{8\Delta y}$$

This is equivalent to calculating the convolution of u with a 3x3 Sobel kernel
$$\mathbf{k} = \begin{pmatrix} -1 & 0 & 1 \\ -2 & 0 & 2 \\ -1 & 0 & 1 \end{pmatrix}$$
and normalizing it by a factor of $8\Delta y$.

To clarify, we have:
- restructured the section 2.4 and 2.5. By combining them into one section (now section 2.4) we strengthen the point that deformation was calculated from the drift fields.
- used not the Green's theorem any longer, but the description with the convolution with the Sobel kernel. We moved the long explanation into the supplement.
- We moved the sentence "We calculated sea ice deformation from the spatially filtered velocity fields" to a more prominent location at the start of the paragraph.
- We used "Lagrangian" only with ice drift and not any longer with the term "deformation" to avoid any confusion

(3) To derive the trajectories, we do exactly as you suggested, we use the gridded velocity fields to integrate Lagrangian trajectories backward starting on the AEM profiles. We reconstruct the position of the trajectory for each time step by interpolating the regularly spaced velocity field to the location of the trajectory and adding the respective displacement (velocity*time). The trajectories were only used to identify the position of the ice within the deformation field, but not for calculating deformation. We rephrased as you suggested the Section "Lagrangian ice drift trajectories" and added a sentence at L193 "The trajectories were only used to identify the position of the ice within the deformation field, but not for calculating deformation."

2) Restructuring and text/grammar editing.
   Thanks for your detailed notes on where we could restructure and also edit the manuscript for clarity. We have restructured the following:
   o Information on ITD: We rephrased the introduction and cited literature (L. 27). We also shortened the description in the results because it is not a new information and only used to support the modelled thermodynamic thicknesses (L. 271)
   o Spatial variability in the deformation: We rephrased this as you suggested.
   o Methods: We combined the Section on Drift and Deformation to underline that deformation was calculated from the drift fields (L. 160).
   o Results: We split Section 3.1 in two subsections, one on the thermodynamic and one on the dynamic thickness change.

Specific comments:

p3 L71: "Since modal thickness is considered a good first guess for the thickness of the thermodynamically grown" Reference? If this comes from your results, then mention it.

We add a reference to studies that provide details on this interpretation of the ITD (Wadhams 1994, Throndike, Parkison, Rothrock, 1992, and add a sentence in the introduction that explains why the modal thickness is a good first estimate for the thermodynamic growth (L. 27)). We shortened the information in the results (L. 271).

p3 Figure 1:
   a)   Mention how the white trajectories were acquired.
We modified the caption: "Sequence of white arrows illustrates four ice drift trajectories derived from daily velocity fields (section 2.6) representing the typical south-easterly ice movement during the convergent closing of the polynya.
Insert: "20m ride" –> "20m ridge"
Done

b) The general low-resolution ice drift does not match the trajectories seen in (a).
Maybe mention something about this?
For display reasons, we have chosen a coarse spacing of the arrows displaying the low-resolution drift in (b). That is why the change in drift direction from a coastward-directed (b) to a partly coast-parallel motion (a) cannot be identified in the low-resolution drift product. We extended the caption of b. "Overview map with monthly averaged, low-resolution sea ice drift in March 2018 (not showing the small-scale drift variability in the polynya)"

c) This panel does not seem useful. Could be removed and keep only the March 1st
contour in red in (a)?
We have restructured Figure 1 by increasing (a) and removing (c) and (d).
Readers interested in the SAR images on March 1-March 30 are directed to the video supplement (https://doi.org/10.5446/50650).

d) "Combined ice thickness distribution of the FYI shown"... add along the AEM tracks?
Ok

p5 L125: "a bin width of 10 cm." Not sure if this will affect the results greatly, but the
bin size should probably be larger than the instrument uncertainty. Also, the bin width
looks like 20 cm in the figures. Can you clarify?
Thanks for pointing the inconsistency out to us. We recalculated all modal values with a bin width of 20 cm.

p6. Eq (1): Shouldn't this be h(ti+1) =...? Or at least the index of the left-hand side should be
the same as the one for the denominator on the right-hand side. The summation
index should also be replaced with something else than i, or instead you can replace
A(ti+1), h(ti+1) with A(tn), h(tn) if n = 30 as mentioned in text.

My reasoning is as follows:
If we assume ice volume conservation over the whole 30 day period, then we have:
A(0)h(0) +Pk A(tk)_hth(tk) = A(t30)h(t30) ; with k = [0 : 29]
where A(t30), h(t30) is the final area and thickness of ice after 30 days, _hth(tk) = thermodynamical growth between tk and tk+1 (k is an index representing the days of integration) and A(0),h(0) is the initial ice thickness and concentration which are in fact both zero. So we have:
P

k A(tk)_hth(tk) = A(t30)h(t30) ; with k = [0 : 29]
or, if we pose n = 30 as mentioned in the text, then:
h(tn) =
P
k(A(tk)_hth(tk))=A(tn) ; with k = [0 : n ⬜ 1]
We can also stop the integration before the end of the 30 day period. In this case, we
have: h(ti+1) =
P
k(A(tk)_hth(tk))=A(ti+1); withk = [0 : ti]

We went back to our code and agree with you that the index of the left-hand side should be
the same as the one for the denominator on the right-hand side. We will modify Equation 1 to:

$$\overline{h(t_k)} = \frac{\sum_{k=1}^{n} A(t_{k-1})\Delta h_{th}(t_{k-1}, t_k)}{A(t_k)}$$

with n=34. k=0 refers to the 25th of February 2018 and k=34 to the 31st of March 2018.
Dynamic thickness changes between e.g., the 25$^{th}$ and 26$^{th}$ of February are counted at time step k=1
and so on. We have also shortened this Section substantially.

p7 L175: "Outliers in the velocity data were reduced by a 3x3 point running median
filter covering an area of 2.1x2.1 km."
I am not super familiar with signal pre-processing filters, but does this filter simply
smooth the drift to "reduce" the outliers, or does it remove the outliers? If the drift is
smoothed, then it will also affect deformation calculations later, which will then have an
effect on the reconstructed thickness.

The median filter removes outlier but also smooths the velocity field. We have chosen a median
filter because it is better than e.g., Gaussian smoothing at removing noise whilst preserving sharp
gradients in the velocity field that represent the physical deformation zones. Physically incorrect
outliers in unfiltered data would bias the results even more. We added this information in the
manuscript, L. 169 "We filtered the data with a 3x3 point running median filter covering an area of
2.1x2.1 km, which efficiently reduces outliers, whilst preserving sharp gradients in the velocity field
that extend over more than two pixels."

p7 L187: "For deformations in which velocities and their gradients are small in comparison to the
reference length scale, the strain rates can be linearized and transformed
into two invariants of the 2D strain rate tensor"
I a not sure why this specification is needed. The shear rate and divergence can be
written in terms of the trace and determinant of the 2D strain rate tensor (i.e. its invariants)
making them also invariants. This is true regardless of their magnitude compared
to the scale of measurements.
We agree and omit the statement. We have moved the remainder of this paragraph to the
supplement.

p7 L188: "We calculated the spatial derivatives from the averaged velocity fields"
Averaged temporally and/or spatially? I don't think this is specified in Section 2.4.
Here we refer to the spatially averaged (3pt median filter) velocity fields. We clarified this in the text
(L. 171)

Also, for consistency, you should make sure that these integrals are calculated only if

the positions/drift values are obtained at similar times in your data set.

They are all calculated for the same period in time given by the two satellite scenes that were used to derive the drift fields.

It is also not clear what trajectories you are using for the positions (x,y) in the integral formulation of the strain rates. Are you seeding drifters at the cell's corner and integrating their trajectories in time using the gridded drift data set described in Section 2.4? Maybe I did not understand the format of the drift data set of Section 2.4... is it a list of trajectories with positions and velocity or is the resulting product a gridded velocity field? So far, I understood that your final drift data set is a gridded product.

Thanks for pointing out here that further clarification is needed. Our drift fields are gridded fields that result from the pattern matching using the two SAR scenes. For the positions (x,y) in the integral formulation, we use the grid points of our gridded velocity fields. This results in a regularly gridded deformation field. Please see also our detailed answer to your general comments on the derivation of drift, deformation, and trajectories.

p8 L205: You should mention what is the typical time interval used to obtain the Lagrangian trajectories/deformations here.

We add "For each time step, which was typically one day, we extracted …" (L. 191)

p8 L218: "deviation of the reconstructed trajectory"
I am confused now which trajectories we are talking about. The ones used to obtain the drift field that is then used to derive the deformation field, or the Lagrangian trajectories that were reconstructed backwards from the AEM tracks?

Thanks for pointing out that further clarification is needed. Please see our answer to your general comments on the derivation of drift, deformation, and trajectories. Deformation was calculated from the gridded drift fields. The term trajectory always refers to the Lagrangian trajectories that were reconstructed backward from the AEM tracks.

p8 L218: I am used to the term "tracking error" for the error resulting from incorrect pattern matching between two satellite images used for deriving the drift field (which will then affect the deformation estimates). But I think here you are talking about the Lagrangian position uncertainty that results from uncertainty in the drift field you used to integrate the Lagrangian trajectories backwards, and not a mis-match of patterns at the pixel-level in the satellite images. Correct? Maybe you could add this distinction here. And maybe discuss the "other" tracking error (i.e. the one resulting from incorrect pattern matching in your algorithm for obtaining the drift data set)?

Thanks for pointing out this confusion and for suggesting a better term. Yes, we describe what you termed "Lagrangian position uncertainty". We modified the text to clarify this point, see L. 200-210

p9 L229: "Those values are averaged and saved."
So you use the averaged deformation within this uncertainty range as the deformation history along the Lagrangian paths?

We have rephrased this part to clarify our procedure, see L. 205-209, 214, 221-224

p9 L239: "Hence, we calculated for every time step a forward and backward field and extracted deformation from both."
Add "a forward and backward DEFORMATION field..."

Done.

Do you average both the backward and forward deformation estimates and use that as an averaged deformation field from which you extract the deformation history along the Lagrangian trajectories (which is then also averaged in the "tracking uncertainty circle")?

(1) we extract all values (from the backward and forward deformation field).
(2) we randomly picked 10 000 combinations from the extracted values.
(3) we calculated ice thickness for all 10 000 combinations
(4) we averaged the thickness to obtain a mean thickness.
We have rephrased this part to clarify our procedure, see L. 221-224 and 254-259

Section 2.7: This section could be shortened by going straight to Eq (5) which
gives the continuity equation for the mean ice thickness as done in many dynamic thermodynamic
sea-ice models (e.g. Hibler 1979, Tremblay and Mysak 1997).
p10 Eq (5): Is the "dot" necessary in div(u_h) since u is a vector and h is a scalar?
Thanks for suggesting how to shorten the methods. We have followed your advice. We removed the "dot".

p10 L266: "thermodynamic growth or melt..."
This should be a thermodynamic growth/melt RATE (i.e. _h/_t) to have units matching
that of dh=dt in Eq (5).
Thanks for pointing out this inconsistency. We rephrased this (L.235).

p11 L277: "Second, we approximated the thermodynamic ice growth within the grid
cell in Eq. 5 by the growth of the undisturbed, thermodynamically growing ice (see Fig.
2 a,b)." Which you estimate from the thermodynamical simulation described in Section 2.2?
Yes. We added half a sentence there, "obtained from the thermodynamic MITgcm run" (L. 245).

p11 Eq(6): Again, the units do not match. The divergence term should be multiplied by
_t (assuming_hand _hth are given in meters).
Yes, this is exactly what we did. We are sorry, that we have lost the factor Δt during our writing process. We have added the term.

p11 L291: "Mean thickness converged to the first decimal after approximately 1000 iterations."
It is not clear what iterations are here. I thought you have 10 000 different representations
of the possible ice thickness evolution along the trajectories. Why would these
"converge" to something?
With this sentence, we wanted to express that 10 000 are a sufficiently large subset of all the potential realizations. Already after 1000 realizations, the mean thickness calculated from all the individual realizations is not changing any longer to the first decimal. We omit the statement in the text as it is not strictly necessary for the reader to know (L. 254-259)

p12 L310: "Since the thermodynamic growth is expected to be evenly over the polynya
region, it leads to rather uniform, level thicknesses of most of the surveyed ice."
This sentence is not clear and needs to be rephrased. It also seems to contradict the
sentence just above stating that most of the ice is in the thicker bins of the ITD due to
significant deformation over the whole polynya.
Thanks for letting us know that this is confusing. We only refer here to the level ice thickness and would like to make the point that the thickness of the level ice is rather uniform. We have moved this sentence to the introduction (L. 27).

p12 L314: "Since the sole interpretation of mean and mode with regard to dynamic and

thermodynamic contributions may miss underlying processes, e.g., the potential contribution of deformation to the observed modal thickness, we will investigate different aspects in the following sections."
Is it necessary to discuss the above paragraph then?
The sentence only limits the reliability of the interpretation of the modal value, but not the reliability of the other parameters describing the ITD (maximum thickness, e-folding, open water fraction, …). We understand that this sentence is misleading at this location, which is why we restructured the whole paragraph. Please see L. 266-274.

p12 L320: "The modal thickness of the level ice is also identical to the mode of the overall ITD, supporting our assumption that it represents best the thickness of thermodynamically grown ice."
The order should be reversed: you don't need to assume anything if you show this result first. This is really what allows you to speak of the overall modal thickness as representative of the thermodynamical growth of level ice. This should be presented first in the manuscript, or at least you can mention that "AEM results show that the modal thickness is representative of the thermodynamical growth of level ice in the polynya."
We agree that "assumption" is not the right choice of words here. As indicated in our answer to L. 71, we build here on previous results and thus rephrase this part, see L 271.

p12 Section 3.1: This section is titled "3.1 Overall, large-scale dynamic thickness change due to area decrease of the closing polynya" and therefore hints at a link with Eq (1) presented earlier, but it starts by describing mostly the thermodynamical growth... The link with the area change and dynamical growth comes only later in a subsection (3.1.1). Maybe you could have two subsections instead to separate the discussion around large-scale thermodynamical (3.1.1) and dynamical (3.1.2) growth and rename this section "3.1 Large-scale thermodynamical growth and dynamical thickness change due to area decrease of the closing polynya"?
Thanks for suggesting how to restructure this part. We followed your advice.

p13 L333: "Divergence on March 30/31 and the occurrence of open water and very thin ice are visible in the divergence time series in Fig. 4 and in the ITD of the closing polynya (Fig. 1d, 2a), respectively."
There is no time series of divergence in Fig 4. If you refer to the time series of "Area extent FYI", then please introduce the relationship between the Area change and the divergence.
Thanks for pointing this out. You are right, we referred to the time series of "area extent FYI". We removed this sentence as we are not covering the MYI any longer.

p14 Figure 4: In the label, please mention that the thermodynamic contribution (red) is obtained from a simulation, and not observations. The title could also be changed to "Dynamic and thermodynamic contributions to mean thickness from model and observations" or " Observed dynamical and simulated thermodynamical contributions to the mean ice thickness", or something like that.
We followed your suggestions and added "modelled" where applicable.

Are the contributions presented in (b) calculated for the trajectories only? At least, from the text in section 3.3 (p.20), it seems like the error bars are derived from the trajectories. Please specify.

The thermodynamic contributions are the output of the MITgcm runs. The dynamic contributions (starting from the 2nd of March) are from the trajectories only. We specified this in the caption. We also marked in Fig. 4b that the dynamic thickness change is only for the period March 1 to March 31.

15 L362: "we have also observed"
This has not been shown yet in the figures, so we don't know what this means. Change to "we also observe" and then refer to the figure where we can see these differences? Or present the observations for the different zones first, and then conclude about their regional variability.
We have removed this sentence as it was only repeating information that is explained further down in the text in more details

p15 L365: "Based on variations of mean ice thickness along the profiles"
Can you give more details about how you separated the regions? i.e. an increase/ decrease in the mean thickness along a moving average, or was it heuristic? Please specify.
Thanks for asking. We realized that our text lacks some information here. Our decision was based on several aspects of thickness and deformation history. 1: thickness: the running mean of the ice thickness (see Fig. 1), the occurrence of level ice, and the frequency and thickness of the deformed ice (Fig. 6, Tab. 1). 2: deformation history of the ice: path length and origin of the trajectories (Fig. 5a) and the timing, magnitude, and type of deformation that the ice experienced (see Fig. 5, video supplement). For example, ice in the Fast Ice zone is the thinnest and has the largest percentage of level ice. It had short travel paths and experienced weak deformation only during the early deformation phase (Fig. 5b-d). All four zones with their characteristic thickness and deformation properties are described in the text. We rephrase this in the text, L.310-316

p16 Figure 5: This figure should come after the current Figure 7 since it is discussed mostly after Figures 6-7 are discussed.
We followed your advice but this resulted in referencing the original Fig. 5 (p.14, large-scale drift in the insets) before Figures 6-7.

p19 Figure 7: Are all 3 profiles included in these ITDs? Please specify.
Yes, all 3 profile lines are displayed in this figure. We will specify in the caption "ITDs of all four FYI zones of all three AEM lines on March 30/31."

p20 L456: "The mean thicknesses of all 715 trajectories or grid cells, respectively, were combined to compute the ITD of the modeled ice thicknesses." I imagine that you only compiled the simulated thicknesses after the full integration of the trajectories was complete. Correct? Please specify it.
Thanks for pointing out the unclear wording. We rather mean the "integrated thickness of all 715 trajectories on March 30/31 were combined to compute the ITD of the modeled ice thicknesses."

p21 L476: "Underestimation of observed thicknesses is larger in the less deformed Fast Ice and Inner Polynya zones."
I don't see this in FIg. 10. The modeled ice thickness is almost right on top of the observation sin the Fast Ice region in (c).
We agree that in Fig. 10 c this is not visible. We refer here to the mean (see Tab. 1, last column). Since we do not conclude from this observation, we omitted the sentence to avoid confusion.

p22 Figure 10: For clarity, the dashed line for the modeled uncertainty should be the

same colour as the data is belongs to (i.e. blue instead of black).
*Thanks for this good hint! We followed your advice.*

p25 L569: "Apart from those differences in the shape of the ITD, we have found that modeled mean ice thicknesses were generally smaller than the observed ones."
But the reported simulated mean thicknesses in Table 1 always fall within the uncertainty of the observations.
*Thanks for mentioning this, because we realized that more information is needed in the caption. In Tab. 1, the mean is given with its standard deviation which describes the spatial variability. The uncertainty of the thickness observations is assumed to be +/- 10 cm (L. 105), plus an additional 4 cm from the snow cover. We added this information in the caption of the table.*

**Editing suggestions**

p1 L9: "characteristic" –> significant?
*Done*

p1 L18: MYI was not previously defined.
*We removed this sentence. MYI is now first mentioned in L.49*

p2 L32: "results in the presence of very variable thickness" weird formulation... maybe write "results in large ice thickness variations"?
*Changed, see L. 25*

p4 90:"along Lagrangian backward trajectories..." –> using Lagrangian trajectories integrated backwards until its initial formation?
*Done, see L. 86-87*

p4 L92: "forced by time series of SAR derived, small-scale deformation" –> forced by the time series of SAR-derived small-scale deformation history...
*We rephrased this, see L. 89*

p7-8 L189-90: "We relate the result to the center of the four grid cells." It is not clear what this means.
*We rephrased this, see supplement L. 27.*

p9 L239: "considering both deformation estimates.." –> "considering both deformation estimates calculated with the forward and backward drifts"
*done*

p12 L310: "to be evenly" –> "to be evenly distributed"? or change to "to be the same"?
*We rephrased this, see L.27*

p13 L343: "we relate the overall area decrease of the polynya to the observed thickness change." using Eq. (1)?
*Done, see L.298*

p14 L350: "deformation within the polynya was regionally variable and distinctly different in certain zones" This means the same thing twice. Change to "deformation within the polynya showed significant regional variability"?
*Done, See L. 296*

p15 L354: "the observed mean thickness" –> "the observed mean thickness along the AEM tracks"?
Done, see L. 300-301

p15 L368: "The ice within each zone had similar mean thicknesses and similar ITDs."
I think you mean "The ice within each zone had similar mean thicknesses and similar ITDs across all 3 profiles." or something like that, otherwise it sounds like the different zones have the same characteristics, which defeats the purpose of defining them.
Done, see L. 318/319

p15 L381: "To do so, we derived ice drift trajectories of those 715 sections by means of the SAR imagery (Sect. 2.6)." The wording isn't clear. Remove and say in the next sentence: "The general motion of the 715 reconstructed trajectories (see Sect. 2.6) was South-South-East... " ?
Done, see L. 332

p18 L416: "base" –> "basis"
Done

p18 L419: "deformation parameters" add what they are in ()?
We rephrased the sentence, see L. 348.

p20 L435: "of the simple volume-conserving model" –> add "(Sect. 2.7)" or Equation no.
done, see L. 367

p20 L437: "our thickness model" –> "this thickness model"
Done

p20 L437-438: "they reproduce" –> "it reproduces"
Done

p23 L480: "This way, within a month thermodynamics and dynamics restored a first year ice cover that was almost as thick as the surrounding MYI." The wording is not clear. Please rephrase.
We removed this sentence.

p23 L491: "Magnitude of deformation shapes ITD" –> "The magnitude of deformation shapes the ITD"
Done

p23 L502: "0.39 cm" –> 39 cm or 0.39 m
Done

p23 L506: "Taking advantage of the fact that the strongest deformation event left the largest impact on h" Not clear. Please rephrase.
We rephrased this, see L. 431-432

p24 L518: "We test whether the here observed..." –> "We test whether the range of

e-foldings observed here..."
We rephrased this, see L. 445-446

p25 L558: "However, the derived ITDs are composed of mean thicknesses in the 1.4 km, long grid cells of our model, which are too large to resolve individual ridges or ridge clusters." Not clear.. Change to "However, the simulated ITDs are obtained with a spatial resolution of 1.4 km..." or something like that.
We rephrased this, see L. 489-491.

**References:**

Thorndike, A.S., Parkinson, C., Rothrock, D.A. (Eds.) (1992) Report of the sea ice thickness workshop, 19-21 November 1991, New Carrollton, Maryland. Polar Science Center, Applied Physics Laboratory, University of Washington, Seattle (http://citeseerx.ist.psu.edu/viewdoc/download?doi=10.1.1.1018.8242&rep=rep1&type=pdf)

Wadhams, Peter (1994), Sea Ice Thickness Changes and Their Relation to Climate, In: The Polar Oceans and Their Role in Shaping the Global Environment, Geophysical Monograph, American Geophysical Union

Haas, C., A. Pfaffling, S. Hendricks, L. Rabenstein, J.-L. Etienne, and I. Rigor (2008), Reduced ice thickness in Arctic Transpolar Drift favors rapid ice retreat, Geophys. Res. Lett., 35, L17501, doi:10.1029/2008GL034457.

---

## Author Response (AR2)

Dear Jenny,

Thank you very much for your suggestions that were really helpful. We implemented them as described below. In addition, we asked a native speaker for his comments on grammar and style, which has hopefully improved the clarity of expression.

Line 3: "The closing corresponded to". As a native English speaker I would not choose to use "corresponded to" here. While this word is technically okay I would choose "presented" as this word invokes that you used a natural event as an opportunity to investigate the ice dynamics.
Done

Line 40: Do you mean to include precision of 2.01m? Also, what is mean maximum sail heights? The length scale over which you average is important. Sails can be much taller than this. I am off the opinion that this information is not actually needed here, and even a little hard to unpack.
We have removed it and only cited the papers where more information can be found.

Line 69: Kwok (2002) is singular, so change "have shown" to "showed" or "has shown".
done

Section 2
"We quantified the large-scale thermodynamic growth and thickness changes in the FYI zones, the latter from the decrease of FYI area during the closing of the polynya"
... this is not completely sensical. Do you need to replace thickness changes with "thickness redistribution"? It would also help to specify if this is quantified from observations or a model.
We rephrased this to "We quantified the thermodynamic growth from a model simulation and the large-scale dynamic thickness increase in the refrozen polynya from the decrease of the area covered by the refrozen polynya."(L85)

line 125: "is base on the strong conductivity difference between seawater and ice used to determine the vertical distance" Grammar is off a bit here, perhaps simplify the sentence. " AEM thickness retrievals find the distance to the strongly conducting seawater under the ice.
We rephrased this to "AEM thickness retrievals find the distance to the strongly conducting seawater under the ice."(L99)

line 130: I do not understand what you mean by "the effects compensate for the mean thickness". Rephrase. I think you mean "The two effects compensate each other such that the mean thickness is found to be in close agreement with drill-hole measurements." Do you need to specify this is for ridges, or are the effects only compensated with you include level ice?
We rephrased this and clarified it in the text (L104): "When averaging over longer distances of ridged ice, the two effects compensate each other such that the mean thickness is found to be in close agreement with drill-hole measurements."

line 149: We seperated MYI from the newly formed FYI
Done

line 162: You have not defined the varibles in the Equation. Also, I think you want to specify you perform the fit to the ice thickness distribution.
We rephrased this to (L132) "We characterized the tail of the ITD by the e-folding lambda of an exponential fit to all the ice categories of the ITD thicker than the modal thickness h_mode. The exponential fit has the form f(h)=a \cdot e^{-\frac{(\textrm{h}-

\textrm{h}_{\textrm{mode}})}{\lambda}} where h is the thickness of the different bins and a is a fitted parameter.

line 172: along which profiles? Specify Polar-5 or icebridge.
We specified this in the text (L137).

Section 2.2: How different are the answers for the growth rate based on the model you used and freezing degree days? The reason I ask, is it is important to understand the magnitude of the volume error that thermodynamics growth might present.
The thermodynamic thickness stated by Ludwig et al. (2019) on March 31 differs by 23-28 cm from ours (see their Fig. 10). Ludwig et al. (2019) discussed the difference between their modeled results and the modal ice thickness and attributed the differences to insufficient heat flux assumptions in the models (esp. ocean heat flux) or rafted level ice. We refrained from discussing the difference between their and our estimates in our paper because our thermodynamic growth was in very good agreement with the observations of the level ice thickness. Even though we cannot rule out that our level ice filter sometimes erroneously detects rafted ice as level ice, we still think that the majority of the level ice was undeformed and hence representative of the thermodynamic growth.

line 209: I would remove the sentence about not accounting for the reduced ice growth as the mean thickness increases, because the model does this for the level ice and is reducing the ice growth rate as the mean ice increases dynamically. Unless you are using the model to only track level ice thickness, and have turned off the dynamics.
The estimates of the thermodynamic growth (from the MITcgm) is uncoupled from the dynamic changes, because dynamics were switched off. Therefore, we regard this sentence as important information to our reader.

In general you need to define varibles in your equations. Equation 2 define y for example.
Thanks for the advice. We added the missing definitions (e.g. L. 178, 180, 246, 262)

line 349: -div(vh) = -h div(v)
We were not sure whether you comment referred to Equation (4) (L244) or the inline Equation (L248) that follows shortly after when we explain that we work in a Lagrangian framework. We have changed the inline question (L248) according to your suggestion. In case of Eq 4, we would like to quote the "basic" Equation that could still be used in an Eulerian framework as well.

line 365: only very locally
done

line 390: "quantify the thickness change from the large scale" does not make sense to me. Do you mean "for the large scale"?
We rephrased this to (L270) "In this section, we first quantify the large-scale dynamic thickness change, that is linked to the decrease of the refrozen polynya area"

Table 1: Check formatting. It is overrunning a margin.
Luckily, that was only a problem of the track-changes. In the revised version, the table fits again.

Line 546: I am not sure quoting the mean trajectory length means anything here.
We removed it.

line 551: I think you need a "that" in front of dominated. This sentence is quite long, consider spliting it to easy readability.
We have rephrased this to (L349) "For example, the ice parcels of the Shear Zone experienced divergence in the early deformation phase (March 3-6). During the main deformation phase, convergence along the coast dominated their deformation history (March 16-20)."

Line 619: Specify what "it" is in "it lacks the frequent occurance ..."
done

Line 647: "notions of the importance of the ice dynamics" What notions? Personally I would either not choose to use this word or I would clarify "notions of the ice dynamics being significant to the thickness distribution". Notions is really just a wishy-washy word. You can quote Rothrock, Thorndike or Wadhams on the importance of redistribution in the thickness distribution.
Thanks for the hint on the word "notions". We have rephrased this whole section (see also below) to (L414s): "This large contribution of sea ice dynamics is consistent with Kwok and Cunningham (2016) who attributed approximately 42–56% of the seasonal changes in mean regional ice thickness to dynamics in the Canadian Arctic Archipelago. In the light of a future Arctic ice cover which is expected to be thinner and more dynamic than now, our results may improve predictions on the impact of sea ice dynamics on future ice thickness changes, and if stronger and more frequent deformation could partially compensate for the expected increase in sea ice loss. For example, Itkin et al. 2018 concluded that divergence in winter followed by new ice formation is currently responsible for an ice volume increase of 7% in the sea ice north of Svalbard."

Line 650: " if stronger and more frequent deformation can contribute to could partially compensate for the expected, continuing sea ice losses. " If you are going to claim this as motivation for your study you do need to give some indication as to how unusual your results are that 50% of the thickness increase is due to dynamics. I think you just need to be more quantitative in the comparison of the three studies.
We added more information and rephrased this (see above).

line 655: flight -> flights
Done

line 665: as a linear function
Done

Section 4.2: This testing of theory is actually some of the more intresting outcomes of your work, in my mind. I wonder if it deserves some mention in the abstract.
We hoped to have summarized the results of this section in the abstract by "Results show a linear proportionality between convergence and thickness change that agrees well with the ice thickness redistribution theory. We found a proportionality between the e-folding of the ITDs' tails and the total deformation experienced by the ice."

Figure 3: You show the exponential fit, not the efolding value. Change caption to reflect this.
Done

Figure 4: "By March 1, 2018, the thermodynamic thickness amounted to 0.38 m" Perhaps a better sentence is "The new ice that grew in the polynia was 0.38m thick by March 1, 2018.".
Done

---

## Author Response (AR3)

Dear Jenny,

Thanks for your positive feedback. In addition to L144, we have now clarified in the text (L157): "Note that the thermodynamic growth (from the MITgcm) is uncoupled from the dynamic changes because dynamics were switched off in the model run."

We would like to thank you and the reviewers whose comments really helped improve the clarity of the paper.

On behalf of the authors,
Luisa